# Modern Extraction and Purification Techniques for Obtaining High Purity Food-Grade Bioactive Compounds and Value-Added Co-Products from Citrus Wastes

**DOI:** 10.3390/foods8110523

**Published:** 2019-10-23

**Authors:** Neelima Mahato, Mukty Sinha, Kavita Sharma, Rakoti Koteswararao, Moo Hwan Cho

**Affiliations:** 1School of Chemical Engineering, Yeungnam University, Gyeongsan 38541, Korea; mhcho@ynu.ac.kr; 2Department of Medical Devices, National Institute of Pharmaceutical Education and Research, Ahmedabad, Palej, Gandhinagar 382 355, India; muktys@gmail.com (M.S.); Kingpharmatech@gmail.com (R.K.); 3Department of Chemistry, Idaho State University, Pocatello, ID 83209, USA; sharkum2@isu.edu

**Keywords:** citrus waste, citrus byproducts, essential oils, waste management, limonene, phenolics, flavonoids, citric acid, environment friendly extraction, phytochemical extraction and purification

## Abstract

Citrus contains a range of highly beneficial bioactive compounds, such as polyphenols, carotenoids, and vitamins that show antimicrobial and antioxidant properties and help in building the body’s immune system. On consumption or processing, approximately 50% of the fruit remains as inedible waste, which includes peels, seeds, pulp, and segment residues. This waste still consists of substantial quantities of bioactive compounds that cause environmental pollution and are harmful to the ecosystem because of their high biological oxygen demand. In recent years, citrus cultivation and the production of processed foods have become a major agricultural industry. In addition to being a substantial source of economy, it is an ideal and sustainable and renewable resource for obtaining bioactive compounds and co-products for food and pharmaceutical industries. In the present article, the various methods of extraction, conventional and modern, as well as separation and isolation of individual bioactive compounds from the extraction mixture and their determination have been reviewed. This article presents both aspects of extraction methods, i.e., on a small laboratory scale and on an industrial mass scale. These methods and techniques have been extensively and critically reviewed with anticipated future perspectives towards the maximum utilization of the citrus waste.

## 1. Introduction

Citrus is cultivated as one of the largest fruit crops across the globe and has been known to humans for thousands of years due to its health benefits. The center of origin of citrus species has been discovered to be the southeast foothills of the Himalayas, Meghalaya, eastern areas of Assam, India, Northern Myanmar, and the Western Yunnan province, China, on the basis of genomic, phylogenetic, and biogeographic research on citrus fruits [1,2]. Citrus belongs to the Rutaceae family and comprises 140 genera and 1300 species. The origin, geographical spread, and popular varieties of citrus fruits cultivated across the globe are shown in Appendix A, and the main citrus growing regions in the world map along with their annual productions are shown in Figure 1 [3,4]. The cross between the native varieties and the evolution of popular hybrid variants in citrus fruits and the list of main citrus varieties cultivated widely across the globe has been presented in Appendix A [5]. The global citrus production in the year 2016 was ~124.3 million tons [3], with oranges being the largest among all citrus crops. The global orange production of oranges for the year 2018/2019 has been estimated to increase by ~6.3 million to 54.3 million tons [4]. The important varieties cultivated commercially are oranges (61% of total), mandarin (22% of total), lime and lemon (11% of total), and grapefruit (6% of total) [3]. 

The citrus processing industries have been focusing on the production of juices and essential oils for many years. It is estimated that 33% of the total harvest in the world is used for juice production [6]. A high percentage of orange production (70%) is consumed for the production of commercial derivative products, such as fresh juice, dehydrated citrus products or marmalades, jams, and flavoring agents for beverages [7]. Approximately 50–60% of the fruit parts remaining after processing are converted to citrus wastes (peels, seeds, and membrane residue) [8]. Accordingly, the built-up amount of semisolid and solid citrus waste is alarmingly huge. Annually, the citrus waste created by processing industries is estimated to be over 60 million tons worldwide [9]. The bioactive molecules obtained from citrus waste have been reported to exhibit antimicrobial, antiallergic, anticancer, and antidiabetic properties and hence have been promoted as dietary supplements for nutrition and health. The various health benefits and applications of bioactive molecules in neutraceutical/pharmaceutical/therapeutic applications and food preservation have been extensively reviewed in our previous publications [10,11,12,13,14].

### 1.1. Anatomy and Waste Composition

A typical citrus waste is composed of semisolid residue composed of endocarp residual membranes, vesicles, pith residue, and, to a large extent, albedo and exocarp or flavedo. A typical composition of citrus wastes acquired from juice-producing industries is presented in Table 1. The overall composition also depends on the citrus species, variety, and the harvesting season. The compositions of typical citrus waste (peel and rag) and dried citrus pulp are shown in Figure 2 and Figure 3.

The waste is rich in sugars, fibers, organic acids, amino acids and proteins, minerals, essential oils (mainly *d*-limonene), lipids, and large amounts of polyphenolic compounds and vitamins [28]. The physical and chemical composition of citrus waste varies in different fractions of the fruit, such as juice, albedo, flavedo, rag and pulp, or seeds [29]. This, in turn, depends on different methods/techniques employed for the juice and pulp extraction [30]. The citrus waste constitutes ~5–70% of the fruit—of which, ~60–65% is peel, ~30–35% is internal tissues, and up to 10% is seeds by weight [31]. The composition of dried citrus pulp waste is different according to the relative proportions of skins and seeds. 

### 1.2. Why Is Dumping Untreated Citrus Waste Risky?

Mostly, organic wastes originating from the fruit processing and food industries are dumped on barren grounds because it is believed that they add organic humus and minerals to the deficient soil and are converted into fertile soil. An important aspect regarding this fact is that if the nitrogen content in the waste material is below a certain level (<0.14%), it cannot support the decomposition process by microbes, e.g., bacteria. This has serious implications, as this condition creates a further challenging situation by using the existing nitrogen in the soil, rendering it deficient. Citrus canning plant wastewaters contain high concentrations of detergents and alkalis or bases and fermentable sugars, such as glucose, fructose, and sucrose, accounting for approximately 70% of the biological oxygen demand (BOD). The BOD content in different sections of industrial citrus wastes created during different operations is displayed in Table 2 [32]. 

This situation is often seen in the case of citrus waste dumping into the ground. Moreover, the sugars present in the waste attract flies, and there is a bad smell (odor) at the dumping site. It is only possible to improve this situation by supplementing the dumping area with nitrogenous supplements, i.e., chemical fertilizers containing cyanamide to the ground. This can be done systematically by adding 100–200 kg of calcium cyanamide to every ton of ground waste, mixing thoroughly and allowing it to dry. Additionally, nitrates, ammonium sulphate, and superphosphates can also be added to facilitate the bacterial decomposition of waste and humus addition to the soil [27]. Alternately, these wastes can be utilized in the fermentation process to yield organic acids. The main challenge is the removal of oils (*d*-limonene) from the waste which possesses antimicrobial properties. With the increasing risks towards the environment and restrictions imposed by government laws, the citrus waste processing industries have to consider waste management as a total financial liability. In this direction, developing a zero-waste citrus industry concept appears to be an economically profitable as well as environmentally-friendly resolution. On the one hand, the wastes can be converted into useful salable products and, on the other hand, an appreciable part of the BOD in citrus wastes can be removed, easing the disposal problem [32]. A schematic diagram illustrating the risks and potential threat of pollution to soil and aquatic ecosystems and the overall environment because of untreated waste disposal is presented in Figure 4. Alternately, the citrus waste can be utilized for obtaining valuable bioactive compounds, such as phenolic compounds, pigments, oils (limonene), lipids, etc., employing extraction and purification techniques, and the remaining residue may be subjected to anaerobic digestion to produce biomethane and alcohol. The resultant digestate post fermentation can be utilized for the production of soil compost [33].

## 2. Extraction Methods

Citrus has been a most extensively researched subject in the category of fruit crops in the last hundred years. Research began during the post-industrial revolution. In the last few decades, there has been a quantum leap in the research after the inventions of advanced techniques for the extraction, identification, characterization, and authentication of extracted compounds to spot adulteration and modern equipment for designing and manufacturing various commercial products. In recent years, extraction techniques for the chemical, food and pharmaceutical industries have received a lot of attention because of the increase in energy prices, CO_2_ emissions, and other environment-related problems. Furthermore, tremendous growth in the production, processing, and consumption of citrus fruits has created a lot of challenges for researchers. One of the most crucial topics in this regard is the development of methods and techniques to achieve the maximum extraction of the valuable compounds and byproducts at a low cost [10]. This section gives a comprehensive outlook and collection of scientific reports on the progress of the development and modernization of equipment and extraction methods, techniques, and their relative merits. Some of the popular methods for the extraction of industrially important valuable compounds from citrus peels include reflux distillation, shaking, stirring, microwave, ultrasonic extraction, and so on [19,34,35,36,37,38,39,40,41,42]. Supercritical fluid extraction, ultrasound extraction, the controlled pressure drop process, and subcritical water extraction are modern and rapid methods that consume less solvent and energy. Supercritical fluids have several advantages, such as nontoxicity, nonflammability, no chemical residue, and low/moderate operating temperatures and pressures [43]. Microwave hydrodiffusion and gravity (MHG) [44] is advantageous in comparison to conventional methods, as it requires shorter extraction periods and consumes smaller amounts of solvent and works under the effect of microwaves and the earth’s gravity at atmospheric pressure without using any solvents [45]. In this process, a mass of 500 g of fresh plant waste is heated under microwave at a certain temperature. No solvent or water is used. The direct interaction of microwaves with the cells causes rupture of the cell wall and heating the water in cytoplasmic contents sets off their release. These compounds then diffuse out and settle down naturally under the effect of gravity on a spiral condenser outside the microwave cavity. The crude juice is collected continuously in a graduated cylinder. The extraction process is continued until no more juice is extracted or overheating is detected [44]. Essential oils are one of the main byproducts of citrus peel wastes and are used extensively in several commercial products due to its characteristic pleasant aroma. Citrus essential oils possess antimicrobial, antibacterial, and anti-insect properties. Due to adulteration by other essential oils, the quality degrades. Commercial manufacturers utilizing essential oils need quality control checks in this regard in order to spot adulteration and to ensure authentic products. Mahato et al. reviewed different methods for the extraction of citrus essential oils and modern authentication techniques and the application of essential oils in the preservation of fruits and vegetables as well as processed foods [12]. A list of various extraction methods, conventional and modern, and separation, isolation, and purification techniques have been summarized in Figure 5. The conventional and modern (nonconventional) extraction methods and their working principles have been summarized in Appendix A. Furthermore, schematic diagrams illustrating the experimental apparatus set-up are presented in Appendix A. 

The extraction of compounds results in obtaining a mixture of many compounds. It is then processed for the isolation of individual compounds by purification methods. This is followed by the identification of compounds, and characterization and authentication. A summary of the methods employed for the estimation of isolated and purified compounds obtained from extractions is listed in Appendix A. Chromatographic and spectroscopic techniques are discussed later in this article. The classification of major and important bioactive compounds extracted from different parts of citrus wastes has been listed in Appendix A.

### 2.1. Polyphenols

Polyphenolic compounds are generally present in the outer surface of citrus fruits, such as peels, shells, and hulls, and protect the inner tissues from harmful UV and IR rays of the sun as well as microbial infections. Their extraction is dependent on the kind of solvent selected and the type of treatment during the extraction process chosen. For example, phenolic compounds, such as *p*-cinnamic acid, ferulic acid, isoferulic acid, 5-hydroxyvaleric acid, vanillic acid, and 2-oxybenzoic acid, are generally detected in citrus peel extracts, but a difference has been found when heat treatment is employed during the extraction process. In ethanolic extraction (with 70% ethanol) along with heat treatment for 30 min at 150 °C, 2,3-diacetyl-1-phenylnaphthalene, ferulic acid and *p*-hydroxybenzaldoxime were detected. On the other hand, in a similar process without heat treatment, 2-oxybenzoic acid and 2,4-*bis*-hydroxybenzaldehyde were detected. Alternately, in water extracts, with no heat treatment, *p*-cinnamic acid and isoferulic acid were detected, whereas post-heat treatment, 5-hydroxyvaleric acid, 2,3-diacetyl-1-phenylnaphthalene, vanillic acid, and ferulic acid were detected. Furthermore, it was also found that a simple heating process can release several bound phenolics into the extraction solution and increase the overall resultant antioxidant activity of citrus peel extract [35]. Polyphenolic compounds are extracted as total polyphenolic content. The extract is a mixture of several compounds in different proportions. A comparison of solvent type-yield amounts and the effect of the extraction method on the yields and antioxidant activity exhibited have been listed in Table 3.

The steps involved in the different extraction methods reported on total polyphenolic content from citrus peels are shown in Appendix A [55,56,57]. The composition of total polyphenol content, antioxidant properties in different citrus varieties across the globe, and the effect of the treatment and different parameters involved during the extraction process are summarized in Table 3. Phenolic components are most commonly collected by the solvent extraction method. Here, the citrus waste is treated with solvents and soaked for a defined time, centrifuged, and then the supernatant is filtered. Following the filtration, the aliquot of the filtrate is concentrated through the evaporation of the solvent. The yield of the recovered various phenolic compounds also depends on the nature of solvents used for extraction. A comparison of solvent type-yield amounts is illustrated in Figure 6.

The extract was then re-dissolved in distilled water and stored at 4 °C prior to the purification step. During purification, sugars and organic acids were removed from the crude extract using the column chromatography technique. Finally, the phenolic compounds were collected. Sophisticated methods for the extraction of phenolics and flavonoids include optimized microwave-assisted extraction (MAE) [59,60], ultrasound-assisted extraction (UAE) [61], pulsed-electric field extraction (PEF), pressurized liquid extraction (PLE) [62], supercritical fluid extraction (SFE), and so on [63]. The method showed better performance in terms of higher yields than conventional techniques and was confirmed by different antioxidant assay systems. It is worth mentioning that the long procedural times of extraction required by conventional methods render phenolic acids susceptible to degradation. Therefore, the application of MAE for the extraction of biologically active compounds, in particular, some unstable phytochemicals, has an edge over conventional methods. The total phenolic contents in the citrus extract were evaluated using the Folin–Ciocalteu assay [50,58,64,65,66,67].

### 2.2. Flavonoids

Flavonoids are the largest group of polyphenolic compounds consisting of a common benzo-*γ*-pyrone structure [68]. Flavonoids are basically 3-ring structures, with two of them being aromatic benzene rings (rings A and B) connected by an oxygen pyrane ring (ring C) and varying numbers of hydroxyl groups in different positions on the ring (Figure 7). There are mainly three groups of flavonoids found in citrus fruits, viz. aglycons, glycosylated flavones (luteolin, apigenin, and diosmin glucosides), and polymethoxylated flavones [34]. The molecular structures of the major flavonoids are shown in Appendix A. Flavonoids constitute 10% of the dry weight of the citrus fruit. These take part as precursor molecules or intermediate metabolites in the formation of a variety of constituents, e.g., pigment (anthocyanidins); and regulate photosynthesis and redox reactions. The main flavonoids found in the citrus fruits are hesperidine, narirutin, naringin, and eriocitrin [10,69]. Hesperidin and narirutin are found mainly in the flavedo part of the peels and other solid residues, whereas naringin and eriocitrin are predominant in liquid residues [70]. Hesperidin concentration has been found to be higher in peels than in juice or seeds, probably responsible for fruit coloration. Ruby red grapefruits contain the highest percentage of naringin. The rind core and segment membrane contain 75–90% of the total flavonoids. Approximately 90% of the naringin is found in peels, rags, and pulp, 0.02–0.03% in juice, whereas the hesperidin content in the juice is 0.015–0.025% [22]. Peels and seeds are rich sources of phenolic compounds, which include both phenolic acids and flavonoids. The phenolic contents were found to be more concentrated in the peels than seeds [71,72]. The composition of the seeds and peels is not always the same. For instance, in lemons, the seeds principally contain eriocitrin and hesperidin, and the peel contains higher amounts of neoeriocitrin, naringin, and neohesperidin. The concentrations of different glycosylated flavanones have also been observed to vary. For example, neoeriocitrin and naringin were found in almost equal concentrations in the peel, whereas eriocitrin was found to be 40 times more abundant in the seeds compared to naringin [34]. Some minor flavones, namely, apigenin, luteolin, and diosmetin-derived compounds and flavanones, namely, eriodictyol, naringenin, and hesperetin-derived compounds, are found in bergamot peels [10,73]. Flavonoids were commercially produced from citrus waste for the first time in 1936 and reported for activities resembling those of vitamins (Vitamin-P) [22]. Later, some more physiological activities other than vitamin were reported which include antioxidant activity, antimicrobial properties, etc. The first flavonoid commercially produced was hesperidin. The process included liming of ground orange peels which coagulates the pectic materials present in the peel waste, leaving behind hesperidin chalcone in the solution. The peels are further subjected to powerful presses to obtain the maximum yield. They are filtered and acidified with hydrochloric acid to a pH of 6.0. The solution is then heated and allowed to stand overnight, which yields crystals of hesperidin from its chalcone. Hesperidin is filtered and dried to obtain “cakes”. Another flavonoid naringin (flavanone glycoside) was first separated from grapefruit peel and commercially used to induce a bitter taste in beverages, confections, marmalades, and as a raw material to produce rhamnose, *p*-coumaric acid, phloglucinol, and dyes [27]. 

The extraction of flavonoids from citrus waste requires preconditioning of the raw materials. The raw materials are either taken in fresh or frozen or in dried form. In frozen or dried form, they can be milled, ground, and homogenized easily in an appropriate solvent for extraction. It has been observed that freeze-dried samples retain higher amounts of flavonoids than air-dried samples [74]. This might be attributed to a thorough rupture of cell walls to facilitate extraction. The solvent extraction method is considered to be the simplest method for extraction of phytochemicals. The yield of the extracted phytochemicals, however, depends on the properties of the solvent employed, e.g., polarity, extraction duration, temperature, sample to solvent ratio. In addition, the yield also depends on the physical characteristics and pretreatments employed for the sample. The sample is dried and ground into small particles to facilitate extraction of phytochemicals from that concentrated in solid form into the liquid solvent media. This method is preferred so as to avoid or shorten the additional steps of evaporation or decantation of the water content in the extracted amount. In general, organic solvents are used for the extraction of most of the phytochemicals. The process includes two main operations, viz. a simple maceration process in which the phytochemicals diffuse out into the solvent medium from the citrus matrix [75], and a centrifugation process in which the aqueous phases can be separated out of the extracted phytochemicals [76]. The main steps involved in the different extraction methods reported for flavonoids are shown in Appendix A. The commonly used solvents in flavonoids extraction are methanol, ethanol, acetone, ethyl acetate, and their combinations. The solvent ratios can be adjusted according to the sample type, peel waste or pulp waste, depending on the water content in the sample. Methanol has been found to be more effective in extracting flavonoid compounds of low molecular weight and aqueous acetone in extracting high molecular weight flavonoid compounds [77]. 

Other conventional methods of extraction using solvents are Soxhlet extraction, vortexing, centrifugation, and hydrodistillation. Hand pressing and cold pressing methods do not involve solvents. A pictorial representation of the experimental set-up of these methods has been illustrated in Appendix A. In recent years, a number of modern methods of extraction have been developed which include microwave-assisted extraction, ultrasound extraction, and techniques based on high pressure or temperature application and compressed fluids, e.g., high-pressure solvent extraction, subcritical water extraction, supercritical fluid extraction. These nonconventional methods consume less solvent, time, and energy. Methods involving microwave and ultrasound employ wave energy to heat solvents, ensuring less time, solvent, and energy consumption, and efficient extraction. Additionally, they facilitate a better yield at a lower cost [10,11]. Lo Curto et al. introduced an innovative process for simultaneous production of pectin along with hesperidin from citrus wastes. In this method, the orange peel waste is first pretreated with calcium hydroxide, hydrochloric acid, etc., followed by extraction of the compounds via the solvent extraction method. Alkaline treatment or pretreatment with calcium hydroxide induces the insolubilization of pectin present in the complex mixture of citrus waste under an extraction process. The insoluble pectin is now easy to filter out, leaving behind flavonoids in the solution. The pectin otherwise remains as a hydrosoluble entity in the solution and hampers the process of crystallization of flavonoids and separation of the same. Furthermore, liming facilitates isomerization of the flavonoids and solubilizes the derived chalcones. The filtered liquid is then acidified with hydrochloric acid to facilitate the inverse reaction and separation of soluble flavonoids. Ethanolic extracts carry larger quantities of hesperidin compared to water extracts. The yield of hesperidin from the pretreated and dried orange peels is greater (3.7–4.5%) compared to that from fresh orange peels (1.8–2.3%). The better yield is attributed to the solubilization of pectin during pretreatment [18,78]. Enzyme-assisted extraction method has also been tested for extracting phenolic compounds and flavonoids from citrus wastes [64]. The efficacy of this method depends on the activity of the enzymes on the degradation of the cell wall. These include glucanases and pectinases, which break down the proteins and carbohydrates present in the cell walls. This allows exposure of intracellular materials to the solvent media and the compounds accessible for extraction. The composition of major flavonoids determined from methanolic extraction of peels of different citrus fruits cultivated in Taiwan is presented in Table 4.

The enzyme concentration of 1.5% (*w/w*) of the peel sample size is reported to be optimum for the highest extraction [64]. The steps involved in the different extraction and purification techniques for flavonoids from citrus peels are summarized in Appendix A. Total flavonoids are calculated as hesperidin or naringin. Flavonoids along with polyphenols can be tested qualitatively using (a) the ferric chloride test [79], (b) cyanidine reaction [80,81], (c) borocitrate reaction [82], and (d) the alkali test [80,83,84]. Quantitatively, it can be estimated by conventional methods, such as filter paper chromatography [85,86,87,88] and modern chromatographic and spectroscopic techniques, viz. HPLC-DAD (diode array detection), LC–MS/MS, and GC–MS [11,27]. 

Flavonoids possess versatile properties, such as antioxidant, anti-carcinogenic, anti-inflammatory activities, as well as the ability for lipid antiperoxidation [89,90]. Generally, seeds have been found to possess greater antioxidant compounds than those obtained from peels [34]. The compounds in the flavonoids group of compounds possess unique identification/detection patterns specific to each species, which make them very good markers to spot adulteration in commercial fruit juices [91,92,93]. Due to their ability to protect against peroxidation of oxygen sensitive foods, these compounds are also used in food stabilization. Some of the citrus-derived flavonoids are known to be very effective as repellents or toxins and utilized in plant improvement experiments to obtain more resistant crops. In addition to this, flavonoids have also been researched extensively in the field of food technology, for their known properties to provide a bitter or sweet taste and as bitterness inhibitors [68]. Some glycosylated flavanones are easily converted into corresponding dihydrochalcones, which are potent natural sweeteners [94,95]. According to Sanfélix-Gimeno et al., an average European spends up to €454.7 annually in flavonoids containing cardiovascular drugs [96]. 

### 2.3. Phenolic Acids

Phenolic acids are aromatic secondary plant metabolites, possibly the precursors for vinyl phenols, and off flavors formed in citrus products during storage [97]. Phenolic acids consist of two prominent categories based on their structural carbon frameworks, (i.e., number and positions of the hydroxyl groups present on the aromatic ring), viz. hydroxycinnamic acid and hydrobenzoic acid (Appendix A). Phenolic acid and their derivatives are primarily found in the vacuoles of the tissues in seeds, leaves, roots, and stems. Citrus fruit peels contain four main phenolic acids, namely, caffeic, *p*-coumaric, ferulic, and sinapinic acids. Ferulic and sinapinic acids are most abundantly found in the peels of sour oranges [34]. The majority of phenolic acids are found in bound form, either linked with the structural components in the plant tissues (cellulose, proteins, lignin) or larger polyphenolic molecules (flavonoids) and smaller organic compounds (glucose, quinic, maleic or tartaric acid) or terpenes through ester, ether or acetal bonds. Only a minor fraction of phenolic acids has been known to exist as “free acids”. The bound form of the phenolic acids gives rise to a vast range of derivative compounds with complex properties, rendering it difficult to isolate and analyze. Phenolic acids are usually extracted in aqueous alcoholic or organic solvents, e.g., hot water, methanol, ethanol, acetone, ethyl acetate, and so on. The basic principle behind the extraction of phenolic acids is the polarity of the molecules, acidity, and hydrogen-bonding capabilities of the hydroxyl/carboxyl groups present on the aromatic ring. The saponification or alkaline hydrolysis and acid hydrolysis are often included as a pretreatment step prior to the extraction process. This includes both mild hydrolysis employing 1 M NaOH at RT to cleave down ester linkages into carboxylic acid and hot alkaline hydrolysis employing 4 M NaOH at 170 °C for 2 h to cleave both ether and ester linkages. To date, no fixed single method of hydrolysis has been developed, and for every extraction process, a combination of methods is generally employed. The compositional yields of phenolic acids in different citrus varieties are summarized in Table 5.

The steps involved in different extraction processes employed for phenolic acids are summarized in Appendix A. The extraction techniques generally employed for the extraction of phenolic acids are Soxhlet extraction, vortexing followed by centrifugation, sonication, mechanical stirring, microwave irradiation, pressurized liquid extraction in boiling methanol, etc. Following the solvent extraction, the resultant extracts are subjected to fractionation. The latter is based on the acidity of the molecules in the extract. The *p*Ka of the phenolic hydrogen is around 10, and that of phenolic carboxylic acid proton is between 4 and 5. This indicates that removal of neutral compounds is an essential step in order to separate phenolic acids as a whole and isolate the different constituent acids further. This is done after the treatment with NaOH and a sequence of acidification, treatment with NaHCO_3_, and employing further extraction steps to isolate phenolic acids. In this course, the pH is first adjusted to neutral to facilitate the removal of flavonoids and other polyphenolic compounds using ethyl acetate, and then the remaining solution is acidified to the pH of 2 to carry out the extraction of phenolic acids. The fractionation process involves a solid-phase extraction technique to remove unwanted components from the sample. By varying the pH of the eluent solution, larger phenolics and sugars are separated from smaller phenolic components. A relatively lesser employed technique for the extraction of phenolic acids is enzymatic extraction and has been reported to mainly obtain ferulic and *p*-coumaric acids. The enzymes generally employed are pectinases, cellulases, and amylases to rupture carbohydrate linkages. The acids are released on the cleavage of an acetal and hemiacetal bond found between carbohydrate moieties and the hydroxyl groups off the aromatic ring without influencing ester cleavage reactions. Further purification and analysis are carried out using chromatographic techniques. In recent years, the analytical techniques extensively employed for the characterization and determination of phenolic acids are HPLC with a reverse-phase column (most commonly C_18_ stationary phase) equipped with UV and/or LCMS. The biological roles of phenolic acids include protective action against oxidative damage and diseases, such as coronary heart disease, stroke, and cancers [99]. 

### 2.4. Limonoids

These are naturally occurring triterpenes found in abundance in citrus fruits. Limonin is a highly oxygenated triterpenoid dilactone and found in plenty in citrus fruits, particularly in seeds [100]. The basic structure typically contains five rings, including a furan ring attached at C-17 oxygen containing functional groups, a 14-, 15-epoxide group, and a methyl or oxymethylene at C-19. Limonoids are further categorized into two classes, viz. limonoid aglycones and limonoid glucosides. Aglycones include ichangin, limonin, nomilin, obacunone, diacetyl nomilin, isolimonic acid, nomilinic acid, iso-obacunoic acid, deacetlynomilinic acid, etc. Limonin and nomilin are known to impart bitterness to the citrus juices and deteriorate the juice quality. Therefore, it is considered as a challenge for the citrus juice industries. This problem had been constantly under investigation for very long time. The molecular structures of main limonoids found in the citrus are shown in Appendix A. Limonin was first isolated and identified in 1841 by Bernay [100]. In citrus plants, limonoids are found in several parts, viz. fruits, seeds, bark, and roots. The composition of limonod aglycones and limonoid glucosides is shown in Table 6. Limonoids had been customarily extracted using benzene, precipitated using petroleum ether, and crystallized using methylene chloride. In later years, usage of benzene was restricted and prohibited as it is carcinogenic. Limonoid aglycones are low-polarity compounds, normally insoluble in water, and therefore can be extracted using organic solvents, such as dichloromethane, ethyl acetate, and acetone using reflux techniques. 

Post-extraction, limonoid aglycones are isolated using conventional open column chromatography. The latter consists of alumina or silica gel as an adsorbent. This is followed by fractionation and further purification using silica gel chromatography and high-performance liquid chromatography, respectively, and analyzed using spectroscopy. On the other hand, limonoid glucosides are polar compounds and extracted using polar solvents, e.g., H_2_O or MeOH. These include limonin glucoside, nomilin glucoside, obacunone glucoside, ichangensin glucoside, nomilinic acid glucoside, deacetylnomilinic acid glucoside, etc. The common steps of separation, isolation, and purification of limonoids are shown in Appendix A [5]. The steps involved in the different extraction methods for obtaining limonoid aglycones and glucosides are illustrated separately in Appendix A. In recent years, modern techniques, viz., supercritical CO_2_ (SC-CO_2_), and hydrotropic extraction have been introduced to extract limonoids from citrus waste. The former enables saving large quantities of organic solvents saving the environment. However, the cost involved in the consumption of energy to create high pressure limits its practical applicability. In the hydrotropic method, the usage of hydrotropes increases the solubility of limonin in water but leaves large quantities of alkali metal (sodium) salts, rendering its purification challenging. Limonoids are also extracted using solid–liquid extraction and isolated and purified using column chromatography, preparative HPLC, and flash chromatography. The purified compounds are further analyzed using HPLC-UV, GC–MS, and LC–MS [100,102]. It has been observed that it is very difficult to obtain limonoids in large quantities even after repeated purification processes involving HPLC methods. Like flavonoids, limonoids also exhibit action against chronic diseases, antioxidant, anti-inflammatory, antiallergic, antiviral, antiproliferative, antimutagenic, and anticarcinogenic properties. Recent studies have revealed various health benefits and pharmacological aspects of limonoids. These possess antibacterial, antifungal, antiviral, antioxidant, and anticancer properties [103]. Liminoids are known to induce detoxification of glutathione S-transferase in the liver cells of mice and rats [104]. In addition to this, obacunone, nomilin and limonin have shown activity against the growth of *Culex* mosquito (*Culex quinquefasciatus*) larvae, *Spodoptera frugiperda* larvae, and many insects and pests [100]. 

### 2.5. Coumarins

Coumarins and furanocoumarins (psoralen) are mainly present in the peel oils. These are nonvolatile in nature and therefore not found in the distilled oils, and they exhibit strong absorption in the ultraviolet region. The molecular structures of the commonly occurring coumarins present in the citrus are shown in Appendix A. The numbered positions may be substituted with hydroxyl, methoxyl, isopentenyl (prenyl), isopentenoxyl, geranoxyl, and oxygen containing modifications of the terpenoid side chain. The highest amount has been reported in lime oils (~7% by weight in the cold-pressed oils) and lowest in orange oils (≤0.5%) Some of the coumarins, e.g., seselin and xanthyletin, present in the citrus roots have been discovered to inhibit certain enzyme systems. Furthermore, substituted furanocoumarins were observed to sensitize the skin to sunlight and be toxic to fishes [105]. Peucedamin was discovered to be poisonous and showed narcotic action on fishes. Several coumarins have been reported to exhibit antimicrobial activities against bacteria, yeasts, and molds and inhibit the fermentation process. Bergapten shows the skin sensitivity of lime pickers. Natural coumarins, viz. scopoletin, furanocoumarins, and umbelliferone, have been known to play protective roles in plants against fungal infections by *Saccharomyces cerevisiae*, *Aspergilillus niger*, *Penicillium glaucum*, etc. In addition to this, umbelliferone along with other coumarin compounds have also been found to exhibit activity against gram-positive as well as gram-negative bacteria [106,107]. 

In recent studies, it has been demonstrated that furanocoumarins are more active regarding antifungal activities (mycelial growth and spore germination) compared to coumarins. Bergapten and limettin exhibited the highest inhibitory action against *Colletotrichum* sp. and has been found to be greater than previously reported phytoalexins scoparone and umbelliferone. Bergapten and limmetin together as a mixture exhibited even greater antifungal activity than standalone compounds. The composition of coumarins, extracted using methanol, from different varieties of Columbian citrus fruits is shown in Table 7. Ramirez-Pelayo et al. conducted extraction and purification of coumarins and reported six isolated compounds with quantitative estimation. For this purpose, the citrus peels were cut and washed and extracted in methanol using ultrasound-assisted extraction for 30 min followed by filtration, evaporation to a reduced volume, and fractionation using a silica gel column employing petroleum ether–dichloromethane and dichloromethane–diethylacetate as mobile phase. The eluted compounds were monitored using thin layer chromatography. The fractions were then subjected to chromatographic separation using a Sephadex LH-20 column employing ether–dichloromethane–methanol (2:1:1) as mobile phase. The eluted fractions were analyzed using HPLC and UPLC–MS [108].

### 2.6. Synaphrine

Synaphrine is a sympathomimetic alkaloid with the chemical formula C_9_H_13_NO_2_ found as a primary constituent in bitter orange extracts. It comprises approximately 90% of the total phytoalkaloids present in citrus and can be extracted from the aqueous and ethanolic extraction from dried and unripe fruits of bitter oranges (*C. aurentium*) [109]. Synaphrine exists in three different isomeric forms, viz. *p*-synaphrine, *m*-synaphrine, and *o*-synaphrine (Appendix A). *p*-Synaphrine is found in highest concentration in young fruitlets. On maturation, its concentration declines. Its concentration is approximately 0.20–0.27 mg/g in citrus pulp, 53.6–158.1 µg/L in juice, and 1.2–19.8 mg/g in dried fruitlets [65]. The steps involved in the extraction of synaphrine are shown in Appendix A. The aqueous extract is cleaned up employing solid phase extraction with a strong cation-exchange phase followed by derivatization with suitable reagents. The derivative of *p*-synephrine is then subjected to analysis using chromatographic and spectroscopic techniques, viz. LC–MS, GC–FID, GC–MS, and nuclear magnetic resonance (^1^H and ^13^C NMR) [110]. *p*-Synaphrine is a phenylethylamine derivative with a hydroxyl group in the paraposition on the benzene ring of the molecule. It has been widely referred to as a stimulant and found to exhibit cardiovascular activities and positive effects on energy expenditure and sports performance, fat oxidation and weight loss/weight management, carbohydrate mobilization and appetite control, mental focus, and cognition [111].

### 2.7. Pigments

Pigments are color-producing compounds and mainly found in the outer peel or flavedo part and juice sacs. These are largely concentrated in the microplastids and chromatophores. The characteristic green color of the mature lime fruits is found in the cell sap of the peel, and the characteristic orange color of oranges and tangerine is found in the cell walls of the peel. Citrus peels contain two kinds of natural pigments, viz. lipid soluble carotenoids and water-soluble yellow pigments. Blood oranges and grapefruit have color-producing pigments in their cell sap. The main pigment molecules are *β*-carotene, lycopene, and anthocyanins. The green peel color is due to chlorophyll a and b, which later decrease in quantity with a simultaneous increase in the amount of carotenoids. The latter are important from the viewpoint of nutrition, as these have vitamin-like activity [22,112,113,114,115,116,117]. Carotenoids are broadly classified in two main classes: hydrocarbon carotenoids or carotenes (e.g., *β*-carotene, lycopene), and oxygenated carotenoids or xanthophylls (e.g., *β*-cryptoxanthin, lutein, violaxanthin). The molecular structures of common citrus carotenoids are shown in Appendix A. The composition of pigments determined from the peels of different citrus varieties grown in Taiwan is shown in Table 8 (quantitative figures furnished by same experimental research have been included purposefully to negate random errors; experimental results with systematic errors are generally preferred). An additional table displaying some more quantitative information is added in Appendix A. The methods of isolation and identification of carotenoids and chlorophyll pigments in citrus fruits and citrus fruit products are shown in Appendix A. These contribute to various colors to the citrus fruits ranging from yellow to red [118]. The carotenoids content is higher in the peel compared to the pulp [119]. *β*-cryptoxanthin is a precursor to vitamin A and, therefore, also known as pro-vitamin A. *β*-cryptoxanthin is a strong antioxidant and protects our body cells and DNA from damage by free radicals. Many organic solvents are utilized for the extraction of carotenoids from orange flavedo. A number of solvents or combination of solvents are used for the extraction of carotenoids, e.g., a mixture of isopropanol and petroleum ether, acetone followed by hexane, and diethyl ether and methanol [120,121,122] have been reported to yield good quantities. The role of nature of solvent on the yield of carotenoid extraction is shown in Figure 8a,b, and their stability in different pH solutions in Figure 8c. Pigments extracted from solvent extraction methods are subjected to separation and purification using advanced techniques, such as gel permeation chromatography or gel filtration incorporating lipophilic gels for removing undesired compounds [123]. Further purification was carried out using column chromatography, countercurrent extraction and determination by colorimetric apparatus as well as HPLC techniques. Monica et al. [124] carried out a series of comparative study on isolation of anthocyanins from citrus peel extract in water employing a variety of resin columns, viz. EXA-118, EXA-90, and EXA-31 methanol and ethanol as eluting solvents with a concentration ranging from absolute to 50% with water. EXA-118 and EXA-90 were both selected on the basis of their excellent adsorption capacity for anthocyanins due to their ideal pore radii and high surface areas; almost double in the case of EXA-118.

EXA-31 was chosen due to its partly hydrophilic nature but proven to exhibit insufficient efficiency. The study concludes that the extraction of anthocyanins using resin column showed a co-extraction of flavonoids, hydroxyl cinnamates, flavanone glycosides, as well as limonin. Selective extraction of anthocyanins to maximum purity was observed with EXA-118 using 50% aqueous methanol. This combination yielded maximum enrichment in anthocyanins in the extracted phytochemicals and maximum reduction in the co-extraction of hydroxycinnamates, and other flavonoids. In another study, Mauro et al. [124] utilized orange pulp wash obtained from citrus-processing plants for the extraction of anthocyanins employing six different commercial food-grade resins made of styrene–divinylbenzene with pore sizes ranging between 70 and 150 Å and a surface area of 600–800 m^2^/g. However, the presence of other phenolic compounds, such as hesperidin and hydroxycinnamic acid, could not be ruled out. The pigments extracted from citrus peels have been found to be a valuable replacement for synthetic colorant pigments. Carotenoids are also synthesized by bacteria, algae, fungi, and green plants from acetyl–coenzyme *A* [125]. These are now widely used as natural colorants for foods instead of artificial colorants. The latter have shown harmful effects on human health. Processed foods, such as beverages, dairy products, confectionery margarine, pasta, etc., contain *β*-carotene as colorant. Pigments from citrus fruits are utilized in providing additional coloring to food products.

## 3. Value-Added Products

### 3.1. Oils and Lipids

Citrus oils include both seed oil and peel oils or essential oils. The citrus oils are volatile aromatic compounds found abundantly in peels and seeds. To obtain citrus seed oils, the hulls are first removed from the seeds by cracking and oil is extracted in screw expellers. This produces oils and press cakes as residue which still contains 14–16% of oils. Heating is avoided to prevent the oil from getting damaged. Solvent extraction is an efficient technique to recover this residual oil in the press cakes. Pressed cakes are also rich in fibers, which amounts to approximately 26%, and proteins up to 21% [27]. The steps involved in obtaining seed oil are shown in Appendix A. The peel oils are present in the flavedo part of the citrus fruits and extracted primarily by cold pressing in large scale extraction process. The peels remaining after juice extraction are conveyed through tapered screw presses which revolve to press the peels tighter and tighter to release the oil from the flavedo. The resultant oil juice emulsion is washed using water sprays. The peel oil is separated out and collected as emulsion is further screened in several steps which include centrifugation (removes solid particles and separate water); cold storage (precipitating sterols and waxes); decantation (collects 99.9% pure oil); polishing, concentrating, and storage. The main challenge in this process is separation of albedo from flavedo. Oils are either less abundant or absent in the albedo part of the peels. The latter is rich in pectin. For this purpose, the peel is shaved, grated or abraded to obtain the flavedo rich part of the peel, which makes the rest of the oil extraction process very simple. The oils are stored in airtight bottles at a cool place to avoid oxidation and contact with moist air, the failing of which results in oxidation of limonene to carveone and carveol and isocitral into *p*-cymene. The cold-pressed oil can also be subjected to a further concentration process which involves vacuum distillation, and this process removes limonene. Approximately 90% or more of the limonene, also called “stripper oil”, is obtained as a byproduct in this process. The commercial methods of producing oils from citrus waste have been reported in the literature [27]. 

Essential oils are mainly present at different depths in the peel and cuticles of the fruit. These are released when oil sacs are crushed during juice extraction. The molecular structures of the main water-soluble and -insoluble volatile constituents found in citrus wastes are shown in Appendix A. These are volatile in nature, possess versatile characteristics, and find a wide range of applications. The major component of the essential oil is *d*-limonene, which is used as a green solvent for the determination of fats and oils and considered safer than petroleum solvents [126]. The other components are linalool, aldehydes, citral, citranellol, etc. The minor components include *α*-pinene, *β*-pinene, geraneol, acetic acid, alcohols, etc. The essential oils are best removed immediately after the generation of wastes as bacteria present in the waste bring in compositional changes—conversion of *d*-limonene in citrus oils to *α*-terpineol [127]. Essential oils are extracted conventionally by steam distillation and cold pressing. In cold pressing, the peel and cuticle oils are removed mechanically. The yield is a watery emulsion, which is then centrifuged to recover the essential oils [128]. Steam stripping and distillation methods are relatively simpler methods for removing oil components from oil-milled sludge. Distillation is sometimes considered as an economical way to recover the oils (with a better yield of 0.21%) compared to cold pressing (yield of 0.05%) [128]. During the distillation process, the citrus peels are exposed to boiling water or steam. The oils are released into water and then collected through distillation. The steam and essential oil vapors are condensed and collected in a specialized vessel called “Florentine flask” [129]. Compared to conventional steam distillation (SD), the modern distillation technique is effective as it saves time and energy. The modern methods include UAE, SCE, MAE, and enzyme-assisted extraction [130,131]. An increase of 44% in the extraction of essential oils has been reported from Japanese citrus employing UAE compared to conventional methods due to the effective rupturing of peel cells under the effect of ultrasound treatment. The microwave technique has been shown to improve the recovery of *d*-limonene from orange peels in a relatively shorter period of extraction duration [132]. Microwave steam distillation (MSD) is a highly efficient method which not only accelerates the extraction process many folds but also enables recovery of essential oils without causing any changes in the oil composition. The effectiveness of MSD over SD is attributed to the more rapid rupture of the cell wall under strong microwaves and release of cell cytoplasm. 

The amounts of volatile constituents present in commercially extracted citrus juices vary between 0.008% and 0.075%. These can be removed from the citrus by steam distillation methods. These contain a large amount of water and volatile oils that can be distilled off easily, and where oil is at a lower temperature than water. In addition to organic constituents, gaseous substances are also present, such as carbon dioxide (22.0–41.7 mL/L of citrus juice), oxygen (2.2–4.02 mL/L of citrus juice), and nitrogen (9.7–13.9 mL/L of citrus juice) in the citrus juice. Low-temperature distillation or a petroleum ether extraction can remove all the odor and flavor producing compounds from the citrus juice, i.e., unsaturated hydrocarbons and alcohols [22]. SC-CO_2_ extraction of essential oils from Kabosu citrus (*C. sphaerocarpa* Tanaka) peels at 20 MPa and 80 °C showed a yield 13 times greater than conventional cold pressing methods [133]. Recently, enzyme pretreatment employing cellulose enzyme prior to essential oils extraction from citrus peels was studied. An increase in the yield by 2 to 6 times for orange and grapefruit peels was observed, indicating that a combination of different extraction techniques can render the extraction process more effective [130,134]. 

Fatty acids are primarily present in seeds of the citrus fruits. The main lipids found in citrus are: linolenate, linoleate, palmitate, palmitoleate, stearate, oleate, conjugated diene, triene, and tetraene (Table 9). The molecular structures of the main fatty acids found in citrus seeds are shown in Appendix A. The rind part of the citrus fruits has been found to contain oleic, linoleic, linolenic, palmitic, and steric acids. In addition to this, glycerol, phytosterolin, ceryl alcohol, along with small quantities of resin and colored compounds are also found in the rind. The pulp contains oleic, linoleic, linolenic, palmitic, stearic, and cerotic acids. In addition to this, the pulp also contains pentocosane, glycerol, phytosterol, and phytosterolin. These are insoluble in water but soluble in nonpolar organic solvents, e.g., diethyl ether, petroleum ether, chloroform, and benzene. Unlike essential oils, these are not volatile in steam. Lipids obtained from citrus can be divided in three major classes, viz. simple lipids (natural fats, esters of glycerol and fatty acids, citrus seed oil), compound lipids (compounds of fatty acids with an alcohol or glycerol), e.g., lecithins, cephalins, and derived lipids (fatty acid), large chain alcohols, sterols, and hydrocarbons and carotenoids. Lipids are present in much smaller amounts in citrus fruits and occur in suspension after the extraction process. Lipids are extracted from this suspension by centrifugation, filtration, followed by solvent extraction using acetone [27]. 

Various types of citrus oils obtained from different varieties of citrus fruits find applications in flavoring food products, confectionaries, beverages, pharmaceutics, cosmetics, perfumes, soaps, etc. Utilization of citrus essential oils in preservation of a variety of food and derived products is extensively reviewed in the literature [12]. Citrus seed oils are utilized in making soaps, detergents, and preparation of fatty acid derivatives and, in some cases, treatment of textiles and leather. It is generally utilized as cattle feed as protein supplement. However, it has been found to not be appropriate for swine or even toxic to chickens owing to its limonin content. For human consumption, this oil is further refined so as to remove the bitter components (limonin), fatty acids, alkalis, etc. The refined neutral oils are further hydrogenated, winterized, and chilled to separate glycerides. In addition to these, essential oils along with lipids are also used in foods and medicine, pharmaceuticals, and cosmetic products. These exhibit antibacterial, antifungal, and insecticidal properties. Limonene is used in manufacturing plastics and isoprene.

### 3.2. Citrus Molasses

The liquid part of the citrus waste contains 10–15% of soluble solids. Approximately 50–70% of these soluble solids are sugars which are fermentable or otherwise perishable. Due to this, the biological oxygen demand (BOD) of the liquor is 40,000 to 100,000 ppm and capable of creating serious pollution problems to the environment [27] if dumped untreated, because it can percolate through the ground and reach the underground water table and lakes and make the aquatic ecosystem vulnerable. This liquor part of the citrus waste can be converted into molasses. Molasses are thick viscous liquid, dark brown to almost black in color, obtained as an end-product after citrus juice extraction. It is an important byproduct obtained from citrus waste processing. It is very bitter in taste and nearly unpalatable. It is an attractive source for limonin glucosides [136,137]. It contains high amounts of sugars (60–75%) and is used as a substrate for fermentation [31]. It is either consumed as a raw material in distilleries or reincorporated into dried citrus pulp. It can also be mixed with citrus pulp and either directly converted into cattle feed or mixed with other nutrients to create mixed feed products or added to grass silage and used as animal feed [138]. Since it is not synthesized artificially or available commercially, the extraction and purification of molasses from the citrus wastes obtained from juice processing plants could increase its commercial value. To produce molasses out of citrus waste liquor, the latter has to be screened for the removal of all kinds of suspended solid particles/tissues by passing through vibrating screens followed by flash heating and concentrated to a thick viscous liquid, dark brown to black in color and extremely bitter in taste. Typically, it is 72 °Brix, a pH of 5 and viscosity of 2000 centipoises at 25 °C. It contains total sugars, nitrogen-free extract and sucrose amounting to 45%, 62%, and 20.5%, respectively, of the total solids. Other constituents include niacin, riboflavin, pantothenic acid, pectin, fats, and minerals such as K, Na, Fe, Cl, P, Si, Mn, Si, and Cu in trace amounts [27]. The stages of typical citrus waste processing at mass scale to generate molasses are illustrated in Figure 9 and the compositional details of the same are shown in Figure 10a,b.

### 3.3. Dietary Fibers, Carbohydrates, and Sugars

Citrus waste is a rich source of dietary fibers which amounts to 50–60% cellulose and hemicelluloses and can be classified into two main categories, viz. soluble and insoluble fibers. Soluble fibers include pectin, gum, mucus, and some portion of cellulose. On the other hand, insoluble fibers include cellulose, hemicelluloses, and lignin [130]. Pectic substances are complex colloidal carbohydrates present in the middle lamellae of the plant tissues between adjoining cell walls. These are considered as cementing materials which bind the cells together in a tissue. Literature on pectic substances amounts to over 4000 published reports and articles and hundreds of patents. Pectic substances are composed of polymerized galacturonic acid, galactose, and free acid radicals of galacturonic acid, especially of large molecular weight. The free carboxyl groups of the polygalacturonic acids may be found either partly esterified by –CH_3_ groups or partly neutralized by one or more bases. Pectic substances found in citrus fruits include pectinic acid (colloidal polygalacturonic acid with methyl ester groups), pectin (water-soluble pectinic acids with varying amounts of methyl ester), protopectin (water-insoluble pectic substances), and pectic acid (colloidal polygalacturonic acid without methyl ester groups) [139,140]. The composition of pectin in different citrus varieties is shown in Table 10. Pectin is made of linear *α* (1→4)-linked D-galacturonic acid (GaIA) units constructing an overall heterogeneous polysaccharide structure. Upon hydrolysis, these yield pectinic acid. The latter is colloidal polygalacturonic acids containing methyl ester groups. These form gels with sugars (65%), acids, and certain metal ions. Pectic acids are colloidal polygalacturonic acids free from methyl ester groups. Pectic acids are determined by alcohol precipitation and weighing, demethylation by a base, and precipitation by calcium salts [141,142]. The acid groups are estimated by titration, by decarboxylation with hot acid, and determination by liberated CO_2_ [143]. Pectic substances are also estimated by measuring the amount of a colored complex formed with carbazol and sulphuric acid [144]. Citrus fruits are one of the richest sources of pectic substances [22,145]. 

Water-soluble pectinic substances are called pectins. These contain varying levels of methyl ester content with 55–70% of carboxyl groups, sugars, and acids. Pectin is produced under acidic conditions at an elevated temperature of ~100 °C [146]. The most common extraction methods include ultrasound extraction [147] enzymatic extraction [148], microwave extraction [149], and subcritical water extraction [126]. The subcritical water extraction method is very effective for the hydrolysis of lignocellulosic materials and pectin extraction from citrus peel wastes [126,150,151]. The effect of temperature on the properties of pectins during subcritical water extraction has been investigated and reported by Wang et al. [152]. Pectin is commercially produced in the chemical industries using hydro-alcoholic solvents, which enable the extraction of both flavonoids and pectins. In the canning industries, generally, hot chemical peeling using NaOH is employed, which may lead to lower flavonoid contents in the extracts. This treatment leads to the opening of the flavonoid skeleton to the chalcone form, making it more soluble in the solvent phase. The steps involved in different extraction methods for obtaining pectins from citrus waste are illustrated in Figure 11a–l. The pectin content in the food industry byproducts is ~2 to 10-folds higher than that obtained from the chemical industry byproducts. The annual pectin production worldwide exceeds 60,000 tons and constitutes a billion-dollar market [153]. Pectin finds a variety of applications in food and pharmaceutical industries as a thickener, texturizer, emulsifier, and stabilizer. In addition to this, it is also used in fillings, confectionaries, dietary fibers, supplements, drug delivery formulations, and as a naturally gelling agent in the production of jams and jellies [130]. The pectin content in citrus waste ranges between 12% and 25% of dry matter and commercially extracted from the same.

Cellulose is a linear polymer of poly-*β*(1→4)-d-glucose units and is the main constituent of the cell wall structure in plant tissues. Citrus fruit peels and juice residue are an abundant resource of cellulose. Cellulose in the plant cells is organized in the form of crystalline nanofibers surrounded by a noncellulosic matrix, e.g., pectins are extremely thin fibers of 3–4 nm. Such a specific structure enables a variety of applications for cellulose nanofibrils, viz. reinforcement nanomaterials in plastic synthesis, biosensors, drug delivery systems, and biodegradable packaging materials [154]. The steps of obtaining cellulose nanofibrils from citrus waste are shown in Appendix A. 

Carbohydrates and sugars are both present in large amounts in citrus wastes. Analysis of sugars and sugar mixtures in carbohydrates has significant importance in the food and beverage industries. Citrus waste can be treated and transformed into edible sugars. The steps involved in the production of sugars from citrus wastes are shown in Appendix A. The main sugars present in citrus fruits are glucose, fructose, and sucrose, which vary in qualities from less than 1% in limes to approximately 15% in oranges. The quantity of sugars in the citrus fruits largely depends on the climate, temperature, quantity of nutrients in the soil, rootstock, etc. 

Citrus fruits constitutes both reducing (galactose, fructose, and glucose) and nonreducing (sucrose) sugars. In mature orange fruits, both types of sugars are found to be present in equal quantities, whereas in other citrus fruits, such as limes and lemons, which are relatively less sweet than oranges, reducing sugars amounts are found in predominantly greater amounts than nonreducing sugars. On the other hand, in tangerines, nonreducing sugars are found in greater quantities over reducing sugars [138].

### 3.4. Xanthan Gum

High molecular weight of several million Dalton polymeric compounds derived from plants, seeds and seaweeds, generally termed as gum and xanthan gum, is one such biological polymer. Structurally it is a hetero-polysaccharide produced by microbes, *Xanthomonas compestris*. It was first commercialized in the 1960s, and now, its annual production worldwide is approximately 30,000 tons. Xanthan gum finds its main commercial applications in the food, pharmaceutical, and petrochemical industries as a viscosities enhancer and stabilizer [157]. The quantity of acetyl and pyruvic contents present in xanthan gum can vary depending on culture conditions and the micro-organism used resulting in polymer solutions exhibiting different molecular weight, composition, and rheological behavior. Other constituents are glucose–glucoronic acid, mannose, pyruvate, acetate, galactose, etc., depending on the *Xanthomonas* species. The steps involved in the different methods of commercial production of xanthan gum are shown in Appendix A. The pyruvate content is decided by the media composition under the influence of temperature, pH, dissolved oxygen, and so on. A high pyruvate content of approximately 4–4.8% in Xanthan gum exhibits an enhanced thickening behavior compared to the lower pyruvate content of 2.5–5% [158]. However, pyruvate-free xanthan is utilized in enhanced oil recovery (EOR). The main reason can be attributed to the fact that in the latter case, microgels are not formed. 

Bilanovic et al. [159] and Green et al. [160] reported their comparative study on four different fractions of citrus waste as substrate material for the production of xanthan gum by the fermentation process, viz. (a) whole citrus waste, (b) pectic extract, (c) hemicellulosic extract, and (d) cellulosic extract. Citrus waste is considered to be a cheap and inexpensive substrate to obtain the final xanthan product at a minimal expanse instead of employing glucose or sucrose. Furthermore, genetically modified lactose utilizing *X. compestris* has proven to have additional benefits in this direction. The whole citrus waste has been found to be a very good alternative for glucose media as it delivers 37% higher yield compared to the standard glucose medium. The water-soluble contents present in the citrus waste, viz. pectin, organic acids, and simple carbohydrates, were observed to be readily converted into xanthan. On the other hand, complex carbohydrates, viz. hemicellulose and cellulose, yielded an amount of xanthan reduced by 36% and 60%, respectively, owing to their lower biodegradability compared with whole citrus waste.

### 3.5. Organic Acids

Citrus fruits are also known as acid fruits for the reason that their juice is rich in soluble organic acids and sugars. The main contributors to the acidity are citric and malic acids. In addition to these two, other acids are also present, viz. lactic, tartaric, benzoic, succinic, oxalic, formic, etc. The mixture of citric and malic acids, in combination with their salts in the citrus fruits, forms a buffering system in juice which resists a significant change in acidity (pH) on dilution.

#### 3.5.1. Citric Acid

It is a six-carbon tricarboxylic acid and formed as an intermediate during the tricarboxylic acid cycle (TCA). It was first isolated from lemon juice. Citric acid constitutes approximately 60% of the total soluble solids of the edible part in lemon [27]. Citric acid is found in free form in the citrus juice. Citric acid can be extracted from juice or pressed juice obtained from pith and pulp waste by adding calcium oxide or lime. The latter combines with citric acid to form calcium citrate and separates out as a precipitate. This is collected by filtration, and citric acid is recovered from its calcium salt by adding sulphuric acid. The steps involved in the extraction of citric acid are shown in Appendix A. In recent years, the global annual production of citric acid has exceeded 1.4 million tons, with a rising trend in demand and consumption. The industrial production of citric acid is carried out through fermentation using *Aspergillus niger.* Citric acid is also produced through fungal fermentation and chemical synthesis, but the latter is an expensive technique. The commonly utilized microbes reported on producing citric acid are *Penicillium janthinellum*, *Penicillium restrictum*, *Trichoderma viride*, *Mucor pirifromis*, *Ustulina vulgaris*, and various species of the genera *Botrytis*, *Ascochyta*, *Absidia*, *Talaromyces*, *Acremonium*, and *Eupenicillium* [161]. Aravantinos-Zafiris et al. investigated three different strains of *A. niger* and reported the best yield of citric acid by *A. niger* NRRL 599, followed by NRRL 364 and NRRL 567, respectively [162]. Ninety percent of the entire supply of citric acid worldwide is produced with a fermentation process. The two very popular fermentation methods of citric acid production from citrus waste are solid state fermentation (SSF) and the submerged liquid surface fermentation process. The latter has been a more widely adopted technique employing *A. niger*. In recent years, the SSF technique has been observed to demonstrate an edge over the submerged fermentation method. The main advantages of the former are low water content, consequently lower values of water activity, and enhanced aeration facilitating O_2_ and CO_2_ exchange between the gas and substrate matrix. 

Torrado et al. investigated the production of citric acid from orange peels using *A. niger* CECT 2090 in the SSF technique and compared to the results reported on submerged fermentation. They concluded a maximum yield of 193.2 mg/g of dry orange peel in 85 h of incubation, which was greater than that obtained from submerged fermentation. Furthermore, the SSF technique proved to be a versatile technique which did not need any additional nutrients or treatments aside from sterilization [162]. The crude citric acid produced from fermentation is added with lime and subsequently sulphuric acid to recover purified citric acid [27,163]. Since citric acid does not go into phase transition, the simplified production processes can generate a good amount of purified citric acid. Highly purified citric acid involves further implementation of extraction and purification techniques, viz. the spray technique, solvent extraction, the adsorption and ion exchange technique, membrane separation, and estimation and determination procedures [164,165]. In the solvent extraction method, the solvent chosen for the purpose has little or no solubility in aqueous phase, e.g., *n*-octylalcoholtridodecylamine and isoalkane [166]; a mixture of butylacetate and *N*,*N*-disubstituted alkylamide [167]. The citric acid is finally obtained by either distilling off the remaining solvent or washing off the extraction product with water. From the final aqueous solution, purified citric acid is crystallized and separated. The impurities are removed by passing compressed CO_2_ in concentrated citric acid solution in acetone to create an antisolvent effect of CO_2_, thereby leading to the removal of residual impurities [168]. 

Solvent extraction methods provide an advantage over chemical methods as they help in the prevention of the usage of lime and sulphuric acid [169]. The adsorption and ion exchange techniques have an advantage due to quick recovery, high capacity, specificity, and low regeneration consumption. Further, these techniques do not produce or leave byproducts, e.g., calcium sulphate. However, one of the disadvantages of employing these techniques is the high cost involvement in analytical grade chemicals, i.e., large requirement of desorbing solvent resulting in dilution of the recovered citric acid, thereby creating waste liquor in large quantities. Membrane separation methods employ a thin membrane capable of allowing selective mass transport of solute and solvent molecules across itself. The commonly employed membrane separation methods are electrodialysis, reverse osmosis, nanofiltration, ultrafiltration, etc. Electrodialysis was first employed for the separation of citric acid in the 1970s and proved to be very expensive [165]. The citric acid purification process includes several treatments. In addition to usage of chemical reagents, several extraction processes are also involved. The spray extraction process is believed to purify citric acid collected as a resultant product after the fermentation process. It utilizes organic solvents, such as ethanol or acetone, to remove impurities, primarily sugars and other organic acid byproducts, e.g., oxalic acid and malic acid. The purification process further involves fractional precipitation of the remaining sugars at a low pressure of ~20 bars. Citric acid and malic acid both precipitate in the same range of pressures, whereas oxalic acid remains behind in the solution as it can be precipitated only under higher pressures of more than 120 bars. This method furnishes purified citric acid with malic acid as little amounts of impurity. 

Application of super critical carbon dioxide (SC-CO_2_) spray extraction is also recommended for the extraction of citric acid. Dajs and Henczak investigated and reported on reactive extraction of citric acid from aqueous solutions using SC-CO_2_ techniques involving tri-*n*-octylamine (TOA) as a reactant. The latter is utilized to form a complex with citric acid [170]. Citric acid as well as lactic acid can be determined spectrophotometrically via an acetic anhydride pyridine method, gas chromatography, HPLC and high-resolution NMR. It is most widely used in food and cosmetic industries as well as in an average human’s diet. It is used as a preservative in tomato juice, ice cream, sherbets, beverages, jams, jellies, meat products, etc. It is used in the foods as a pH adjustment and flavor improvement agent. Sodium citrate is used as a pH controller and emulsifier for processed cheese. Citric acid exhibits antioxidant properties by the virtue of its chelating activity towards metal ions, which catalyzes oxidation. Further, the chelation of metal ions blocks the substrate for the growth of bacteria, thereby diminishing food spoilage [171]. 

#### 3.5.2. Lactic Acid, Succinic Acid, Pyruvic Acid, and Vinegar

It is an alpha-hydroxy acid due to the carboxyl group sitting next to the hydroxyl group in its chemical structure. It is found in abundance in citrus peel juice and can be commercially produced through fermentation of waste juice, which includes citrus peel and pulp juice, pressed liquor or citrus molasses containing variable amounts of fermentable sugars. Lactic acid finds applications in the commercial production of plastics. In one of the studies, Kagan et al. [172], carried out fermentation of citrus waste juice employing a naturally occurring *Lactobacillus* (*Lactobacillus delbriickii*, ATCC 9649) isolated from a fermenting grapefruit juice and achieving 90% efficiency in 4–5 days in converting the sugars present in citrus waste juice into lactic acid. The steps involved in the extraction method for lactic acid production from citrus waste are shown in Appendix A. The main challenge in the production of lactic acid via a fermentation process is its recovery and purification. In this regard, fractional distillation cannot be applied as lactic acid possesses a high boiling point. In the end of the extraction process, calcium carbonate is further added to render the entire solution alkaline. This precipitates calcium citrate and can be removed by filtration. The filtrate is treated with activated carbon, filtered again, and crystallized to obtain purified crystals of calcium lactate. To this is added sulphuric acid, which precipitates calcium sulphate, leaving behind lactic acid. If ammonium hydroxide is added instead of calcium carbonate during fermentation, then ammonium lactate remains in the solution during the entire process of fermentation, clarification, and concentration and converted into butyl lactate. This butyl lactate is sufficiently volatile and is readily purified by means of fractional distillation. The resultant butyl lactate solution is concentrated and either collected as such or hydrolyzed to obtain purified lactic acid [27,172].

Succinic acid is a dicarboxylic acid produced mainly by chemical routes which include catalytic hydrogenation, paraffin oxidation or an electrolytic reduction of maleic acid or maleic anhydride. Alternately, this can also be produced from fermentation employing microorganisms, such as *Mannheimia succiniciproducens Anaerobiospirillum succiniciproducens*, *Basfia succiniciproducens*, and *Actinobacillus succinogenes*. Among these, *Actinobacillus succinogenes* has been found to produce large quantities of high concentrations of succinic acid based on their ability to utilize CO_2_ and volarizing monosaccharides under anaerobic conditions [173]. Removal of *d*-limonene and essential oils from the citrus waste is a prerequisite for the fermentation process as it exhibits antimicrobial properties. Pectin and biomethane are also produced as byproducts in this process. The steps involved in the production of succinic acid are shown in Appendix A. Succinic acid is utilized in many industrial applications, viz. production of polyester polyols, polybutylene, succinate–terphthalate resins, coatings, pigments and in food and pharmaceutical industries as a flavoring agent and sweetener [27,173].

Pyruvic acid is a 2-oxo monocarboxylic acid, a 2-keto derivative of propionic acid formed as an intermediate compound formed during the metabolism of carbohydrates, proteins, and fats. Commercially, pyruvic acid is produced from fermentation of sugars. The peels and pulp residues in the citrus waste are rich in pectic substances and soluble sugars. This can be utilized as a carbohydrate source to produce pyruvic acid via a fermentation process. In this process, various kinds of microbes, primarily yeasts, have been investigated. Among the different strains of yeasts, *Candida utilis* IFO 0396, *Debaryomyces coudertii* IFO 1381 were found to yield maximum. *Debaryomyces coudertii* IFO 1381, *Candida utilis* IFO 0396, *Hansenula fabianii* IFO 1370, *Hansenula miso* IFO 0146, and *Debaryomyces nilssoni IFO 1255* have been reported to yield pyruvic acid via fermentation. The steps involved in the production of pyruvic acid using *Debaryomyces coudertii* are shown in Appendix A. Pyruvic acid is used as a reagent in clinical analysis and as a substrate for enzymatic synthesis of amino acids, such as tyrosine and tryptophan [174]. 

Vinegar is typically an aqueous solution containing 5–20% acetic acid by volume and finds a variety of applications in food preservation, as a flavoring agent for vegetables, sausages, salads, etc. Vinegar obtained from orange peel and pulp waste possesses a fine flavor and is as popular as apple vinegar in commercial markets. The method of vinegar production from citrus waste is similar to that of the production of other fruit vinegars. The peel oils have to be removed first from the citrus waste, which is antibacterial in nature and blocks growth of microbes employed for fermentation. In general, the citrus waste substrate containing more than 9% sugars is ideal for the production of vinegar. Therefore, the substrate possessing lower sugar content is concentrated to achieve the optimal concentrations for ideal vinegar production. The sugars present in the citrus waste substrate are first fermented to ethanol by yeast (*Saccharomyces cerevisae*) followed by oxidation of the alcohol by *Acetobactor aceti* to obtain vinegar, as in Appendix A. The best result is achieved in 14 days, resulting in a highest yield of 75% *v/v* vinegar from fermentation sweet orange peels and water in a ratio of 1:25 [27,175].

#### 3.5.3. Vitamins

Vitamins are a collection of specific chemical compounds required in small quantities in our daily diet to ensure the sound health of body and mind. Vitamins are not synthesized in our body and have to be taken from an outside source. Citrus fruits are rich in vitamin C (ascorbic acid + dehydroascorbic acid) and contain small amounts of pro-vitamin A, B-complex, and other factors. The composition of ascorbic acid in different parts of citrus waste is shown in Table 11. Vitamin C is chemically L-ascorbic acid. It undergoes reversible redox reactions and interconversion of ascorbic acid and de-hydroascobic acid. The steps involved in the extraction of vitamin C are shown in Appendix A [176]. Vitamin C has been known to be crucial for the growth and repair of tissues in all parts of our body and stimulating the immune system. It participates in forming important proteins used to make skin, tendons, ligaments, and blood vessels, heal wounds, and repair scar damages to the tissues, keeping cartilage and bone tissues and teeth healthy. 

It acts as an antioxidant and helps in the removal of free radicals and treat cancers. It has been a popular remedy for the common cold since ancient times [177]. Riboflavin is commercially produced in two main forms, viz. pure form and vitamin-rich concentrates, either by chemical synthesis or fermentation processes. Extensive discussion on progressive methods for the production of riboflavin along with modification of the previously existing methods has been reported in the literature [178]. Citrus molasses with 2–6% sugar concentration are considered to be an appropriate substrate for riboflavin production employing yeast fermentation. The most commonly employed yeast strains for the commercial production of riboflavin from molasses are yeast-like organisms, viz. *Ashbya gossypii* and *Eremothecium ashbyii*. The fermentation media may sometimes require being supplemented with a variety of proteins and carbohydrate sources to obtain a decent yield. The pH is required to be set between 6.6 and 8.0. Gaden et al. conducted fermentation of diluted citrus molasses containing 6% sugar employing *E. ashbyii* and reported a maximum yield of 0.7 g/L in 7 to 9 days of inoculum, maintained at a pH of 6.7–8.0. However, *A. gossypii* was found not capable of synthesizing riboflavin, when particularly citrus molasses were taken as the fermentation media [179].

#### 3.5.4. Citrus Feed, Feed Yeast, and Industrial Alcohol

Citrus waste, owing to their nitrogen deficiency, is not suitable for dumping at barren unfertile land/dumping grounds. Alternately, it has been noted that it can be converted into animal feed so that the utilizable sugars and other nutrients may add value to the food. Typical citrus waste contains a huge amount of moisture due to its hydrophilic nature of pectin, fermentable or otherwise perishable sugars, and large quantities of nitrogen-free extracts (~63%), a very small quantity of crude protein (~6.2%), etc. Transporting these wastes without drying is challenging, and it is directly a cumbersome as well as expensive process. In many countries, it is directly fed to the cattle, particularly in tropical and subtropical countries where humidity level in the atmosphere is usually high. Alternately, citrus is processed into animal feed through a number of steps shown in Appendix A [27].

The pressed liquor extracted from the citrus waste is also converted into substrate for industrial fermentation, as this contains sugars. The products are called feed yeast and are rich in protein and the vitamin B complex. The generally employed yeasts are strains of *Torulopsis utilis*. On average, the crude protein content of *Torula* yeast obtained from citrus press liquor is 45 to 55 percent and as digestible as feed yeasts obtained from other sources. This includes aminoacids arginine, cystine, glutamic acid, glycine, histidine, isoleucine, lycine, metheonine, phenylalanine, threonine, tryptophan, tyrosine, and valine. In addition to this, it also constitutes nitrogenous nonprotein contents, viz. choline, purines, pyrimidines, glucosamine, etc. The important mineral found in feed yeast is phosphorus. Further, feed yeasts are a good source of the vitamin B complex which includes thiamine, riboflavin, niacin, pantothenic acid, and ergosterol. This can be a good nutrition supplement in a human and animal diet program. The steps of production of feed yeast from citrus-pressed liquor are shown in Appendix A. It is generally carried out via a batch method and a continuous method. The latter became more popular as it enabled obtaining the harvest in a continuous manner. The nutrient solution is fed into an aerated propagator in a continuous manner, and final yeast slurry as a product is simultaneously harvested within less than 3 h of feeding in the raw material. Practically all the sugars present in the pressed liquor are consumed during the fermentation, and a lower percentage of sugar content in the feed material yields a higher percentage of the yield, i.e., 2% of sugar in the feed mash will yield 40% based on the amount of sugar consumed. Since citrus waste is nitrogen-deficient, the fermentation process has to be supplemented with nitrogen in the form of ammonium hydroxide and ammonium sulphate and phosphorus in the form of trisodium phosphate. The microbes consume these and leave behind sulphuric acid, which maintains the pH of the fermentation bath [27].

The sugars present in citrus fruit wastes can also be converted to alcoholic beverages, such as wines, brandies, and cordials by yeast fermentation. Among these alcohols, sweet citrus wines are generally preferred due to their sweet flavor. To carry out a successful fermentation of citrus-pressed juices and liquor, sometimes molasses are required to be added so that the sugar content can be increased up to 10–12%. The fermentation is carried out employing *Torulopsis utilis* and sulphuric acid is added to adjust the pH of 4.0. The fermentation process is completed in 3 days, resulting in approximately 90% of the yield along with yeast as a byproduct and the remaining residue, which can be further converted into cattle stock feed. The precaution taken in this case is screening out/removal of citrus peel oil, primarily limonin, from the alcohols either before the fermentation process employing extraction methods or by conveying through rectifying columns after the production of alcohol as in Appendix A [27]. Recent reports on various extraction processes employed for obtaining the important bioactive molecules and value-added products from citrus wastes along with their relative merits are recorded in Table 12.

## 4. Separation, Isolation, and Purification of Compounds Post Extraction

Crude citrus waste usually contains large amounts of carbohydrates and lipoidal compounds along with bioactive phytochemicals, e.g., polyphenols, carotenoids, sugars, and fibers. To extract a particular component or class of compounds, e.g., concentration of the respective compound has to be enriched in the sample. Such an endeavor includes sequential extraction or liquid–liquid partitioning and/or solid-phase extraction (SPE). These methods facilitate enrichment of particular fraction based on the polarity and acidity of solvents. Elimination of lipoidal fraction can be carried out by employing an aqueous phase to the crude sample along with a nonpolar solvent, e.g., *n*-Hexane [197]; dichloromethane [198], or chloroform [199]. The lipoidal fraction can be separated with the nonpolar solvent very easily. To remove sugars, polar nonphenolic compounds and organic acids, solid phase extraction may be employed. The latter is a rapid method of extraction and can be automated. In addition to this, it is economical. It employs a variety of sorbent materials with cartridges. C_18_ cartridges are widely employed for the separation of phenolic compounds. An aqueous solution containing phenolic compounds is passed through a preconditioned C_18_ cartridge, and the collected fluid is washed with acidified water to remove sugars, organic acids, and other water-soluble components to obtain a phenolic enriched sample. The polyphenols are then separated using absolute MeOH [200] and/or aqueous acetone [197]. Li et al. reported on the basis of their comparative study on different extraction methods, viz. pressurized liquid extraction [62], ultrasonic assisted extraction (UAE), Soxhlet extraction (SE), and heat reflux extraction (HRE) for obtaining hesperidin, nobiletin, tangeretin from *C. reticulata* peels, and employing LC-DAD–ESI/MS for quantitative analysis that the PLE process required less extraction time with higher extraction efficiency with 70% aqueous MeOH [201]. In another comparative study on standard and innovative experiments [190], it was observed that the sequential extraction using instant controlled pressure drop (DIC) followed by ultrasound-assisted extraction (UAE) resulted in higher yields compared to hydrodistillation (HD) and solvent extraction methods for the extraction of essential oils and antioxidants. Pretreatment with DIC improves extraction yields of both essential oils (EOs) as well as antioxidants. EO yield achieved by HD was 1.97 mg/g dry material in 4 h compared to 16.57 mg/g dry matter by DIC in 2 min. Hesperidin and naringin extracted from DIC-treated orange peels with UAE were 0.83 ± 16 × 10^−2^ g/g dry matter and 6.45 × 10^−2^ ± 2.3 × 10^−4^ g/g dm, respectively compared to 0.64 ± 2.7 × 10^−2^ g/g dm and 5.70 × 10^−2^ ± 1.60 × 10^−3^ g/g dm (dry matter), respectively, by SE [190].

The extracted compounds are usually a complex mixture of phytochemicals. These extracted compounds come out from different tissues of the citrus fruit waste. To enrich the concentration of each type/category of phytochemicals, purification and isolation processes have to be carried out. The basic purification process includes column chromatography, high-speed countercurrent chromatography (HSCC) and high-performance liquid chromatography (HPLC). Other solvent combinations commonly used are hexane: *n*-butanol, ethyl acetate: hexane, butanol: water, chloroform: methanol, etc. Water is most commonly used for the detection of most of the phytochemical extracts from fruit parts, including citrus. UV-visible, Mass spectroscopy, and HPLC are the main detection techniques employed for the detection and determination of bioactive compounds extracted from citrus fruit waste. For detection, UV-visible at λ = 254 and 366 nm is usually employed and compared with standard compounds. The UV spectra of flavones and related glycosides are identified by two strong absorption peaks at λ = 300–338 (band I) and λ = 240–280 nm (Band II), corresponding to the presence of a B-ring cinnamoyl system and A-ring benzoyl system, respectively. Substitution of functional groups on either ring-A or ring-B may introduce hypsochromic or bathochromic shifts in the absorption spectra, which can be very useful for identifying the compound’s structure [202]. The photodiode array (PDA)/UV spectra of some of the most common flavonoids, viz. naringin, hesperidin, didymin, tangeretin, and nobiletin display a prominent and characteristic peak in the range λ = 210–400 nm.

### 4.1. Silica Gel Column Chromatography and Preparative Thin Layer Chromatography (Prep-TLC)

This allows obtaining high concentrations of every single compound in the complex mixture obtained after extraction. The separation processes were attributed to the binding property of the silicon atoms in the silica gel column, which form hydrogen bonds through the hydroxyl groups present on it. The components in the mixture are separated according to the difference in the binding forces of the functional groups present in the molecule, which results in its adsorption to the silica gel. The compounds with higher polarities are adsorbed easily by the silica gel, whereas the compounds with lower polarities are easily carried out by the mobile phase (solvent) and collected at the bottom. Preparative thin layer chromatography (*prep*-TLC) is the most economic separation method and does not require sophisticated instrumentation. *Prep*-TLC is practically not an advantage where large amounts of samples are to be treated, as this method is only limited to small sample sizes only. The sample or extract mixture is placed or loaded on a thin layer chromatogram plate (TLC plate) containing silica slurry evenly spread over a glass plate or plastic/polymer sheet. The mobile phase is generally a mixture of solvents, e.g., benzene: acetone (3:1, *v/v*), water: acetonitrile. As the mobile phase progresses across the length of the silica plate, the constituent compounds in the extract mixture are separated according to their polarity and strength of bonding between the organic molecule and silicon atoms in the TLC [203]. Benzene: acetone (3:1, *v/v*) was employed for the separation of polymethoxyflavones (PMFs) from various citrus peel oils [204]. Machida and Osawa reported on the isolation of PMFs from *C. hassaku* peels using a combination of separation methods which included *prep*-TLC and column chromatography. They obtained the crude extract using a conventional separation method by refluxing the citrus peels with ethanol and concentrating the extract to a reasonable volume for further partitioning using ether and water. The resultant compounds mixture obtained after partitioning was separated on silica gel using a benzene–acetone mixture which detected eight different PMFs on the *prep*-TLC plate. The separated compounds are visualized under fluorescent light, and the individual bands are collected using column chromatography and analyzed using HPLC and mass spectroscopy [205]. All major compounds classified under polymethoxylated flavonoids (PMFs) were isolated by the late 1960s, and abundant extraction reports have become available since then. This encouraged an entire new field in nutrition science to implement these bioactive compounds extracted from citrus in various health and nutrition programs. Gradually, these matched consumers’ interest, and presently, inclusion of natural products in health and nutrition products has become a top priority. The first PMF isolated from citrus was tangeretin from tangerine (*Citrus nobilis deliosa*) oils and reported by [206]. Soon after this, nobiletin isolated from mandarin (*C. nobilis*) and sinensitin from orange were reported by [207,208], respectfully. Enzymatic extraction from dried peels of *C. sinensis* Osbeck yielding 564 mg of crude PMFs out of 100 g of dried peel powder of *C sinensis* was reported by [209]. In this process, 100 g of dried peel powder was extracted with 60 °C for 2.5 h followed by concentration of the extract volume treated with diethyl ether (600 mL) and washed with 0.4% NaOH solution until removal of color. The final clear extract was then freeze-dried to obtain a solid powder of crude PMFs. 

He et al. (2011) reported on the simultaneous quantification of flavanones, hydroxycinnamic acids, and alkaloids in pulps and peels from different citrus species using HPLC-DAD–ESI/MS without involving any complicated sample pretreatment procedure [210]. The HPLC method provides a simple and precise means to determine the contents of flavanone glycosides and polymethoxylated flavones (PMFs), e.g., citrus *Pericarpium Citri Reticulatae* (*Citrus reticulata* ‘Chachi’) during storage [211]. Recently, bioactive compounds present in *Citrus limon* byproduct dried powder (CBP) were analyzed with a relatively more advanced technique of reversed-phase high performance liquid chromatography (RP-HPLC) coupled with electrospray ionization time-of-flight mass spectrometry (ESI-TOF-MS) operating at a negative ion mode. The technique allowed the determination of approximately forty metabolites, out of which, six were reported for the first time in CBP [212].

### 4.2. Preparative High-Performance Liquid Chromatography (Prep-HPLC) 

It is used for isolation and purification of valuable bioactive compounds at an industrial scale with quantities in kilograms. The term preparative is attributed to large columns in operation with high flow rates of the eluting solvents. This does not necessarily refer to the size of the instrument or volumes of mobile phase, but the objective of separation, i.e., analytical HPLC is employed for qualitative identification and quantitative determination/estimation of compounds, whereas *prep*-HPLC is employed for isolation and purification of compounds. In the former case, the sample goes to waste from the detector after the operation, whereas in the latter case, the sample goes into the fraction collector from the detector. Preparative column chromatography was first developed during the 1950s and 1960s, and *prep*-HPLC was first introduced in the 1970s. The latter has an advantage due to inclusion of a high-pressure pump to generate the flow resulting in an increase in the throughput and better packing materials of the column with smaller particle size to ensure effective separation power. Presently, completely automated *prep*-HPLC are available to achieve easy-to-use purification of a large number of valuable compounds (over hundreds of compounds every day) in a relatively smaller duration of time compared to other purification techniques. The optimization in column chromatography as well as in *prep*-HPLC is very difficult or can never be carried out as the output result of a preparative run is judged on the basis of three parameters, viz. purity of the product, yield, and throughput, and these parameters are dependent on each other. Therefore, the process of purification has to be handled in such a way that compounds can be isolated with high purity even if the throughput and yield could be compromised [213]. *Prep*-HPLC equipped with a UV-Vis detector further enhances the efficiency of the isolation and purification process. The widely employed mobile phases in *prep*-HPLC for the isolation of flavonoids from citrus are a linear gradient of acetonitrile in H_2_O. The crude sample is first diluted with dimethyl formamide (DMF), and the flavonoids are collected in *prep*-HPLC within 5–30 min followed by joining all the fractions collected at the end, evaporating to dryness in a rotator evaporator. This can be re-dissolved in appropriate solvent for desired application [76]. Chen et al. carried out isolation of PMFs from cold-pressed Dancy tangerine peel oil using a combination of normal phase chromatography and C_18_
*prep*-HPLC eluted with a combination of solvents with increasing polarity, viz. benzene/ethyl acetate, ethyl acetate, ethyl acetate/2-propanol, and 2-propanol. Further purification of PMFs was carried out with methanol/water and ethanol/water [214]. However, the usage of benzene is restricted as it is carcinogenic and mutagenic and, therefore, harmful for health. In another study, Li et al. reported on isolation of nobiletin from orange peel extract using a combination of normal phase flash chromatography and *prep*-HPLC with solvent combinations: ethyl acetate and hexane, 35% ethanol and 65% hexanes, respectively [215]. Levaj (2008) reported on the purification of hesperidin, naringin, and narirutin extracted from pulp and peels of mandarin, Satsuma (*Citrus unshiu Marcovitch*) cv. Saigon and Clementine (*Citrus reticulata* var. clementine) cv. Corsica SRA 63, employing Zorbax C_18_ column using citric acid, and ammonium acetate in H_2_O and MeOH (60:40) as solvent phase [216].

### 4.3. High-Speed Countercurrent Chromatography (HSCC)

This chromatography technique is used to extract and purify flavonoids employing two-phase solvent systems, flowing simultaneously in the opposite direction. This is a liquid–liquid partition technique without involving any solid support matrix, ruling out the loss of sample to the solid matrix because of adsorption. In addition, this method can be executed employing multiple forms of the gradient elution process, thus enhancing removal of impurities from crude extract as well as the final product. Sometimes, pure compounds can be obtained through a one-step isolation process from the crude extract without any need for sample pretreatment [217]. This method was first reported on purification of PMFs from tangerine peel extract obtained from solvent extraction using light petroleum. The crude extract was then concentrated, frozen, and injected to HSCC equipped with UV-detectors and eluted using a combination of a two-phase solvent system involving *n*-Hexane, ethyl acetate, methanol, and water (1:0.8: 1:1, *v/v*), yielding four PMFs, including nobiletin and tangeretin [218].

### 4.4. Flash Chromatography

This is also known as medium pressure liquid chromatography and relatively a rapid method for isolation and purification of compounds compared to column chromatography. A monitored application of medium pressure to the column enables separation of compounds in large sample amounts, thereby yielding a high quality of purified compounds. Presently, complete automated flash chromatography equipment fitted with robotic fraction collectors and online detection units has enhanced the efficiency of separation, isolation, and purification of constituent compounds in a complex mixture of crude extract along with identification [219].

### 4.5. Supercritical Fluid Chromatography

This method employs a combination of pressure and temperature of mobile phase maintained at a critical point which allows the prevention of loss of sample amount to permanent adsorption of the same to the surface of a solid or stationary phase often seen in open column chromatography. This advantage makes this method one of the ideal methods of separation of citrus phytochemicals, particularly, polymethoxy flavonoids (PMFs). The commonly used mobile phase in this technique as a supercritical fluid is carbon dioxide, along with methanol as a polar modifier. This method was first used for obtaining the authenticity of citrus oils through the quantification of PMFs [220]. Later, it was employed for the separation of PMFs utilizing CO_2_ as a mobile phase and methanol as a polar modifier. In recent years, a combination of separation and analytical instrumentation has made this technique more powerful and high yielding. In one such study, hydroxyl- and methoxy-flavones were separated employing SFC equipped with flame ionization and FT-IR spectroscopy detection [221]. In another study, a large-scale isolation process was carried out on orange peel extract employing a combination of normal phase column separation and SFC to yield nobiletin, tangeretin, 3,5,6,7,8,3′,4′-hepatamethoxyflavone, and 5,6,7,4′-tetra methoxyflavone [222]. The extract was purified using a silica gel flash chromatography prior to the treatment of SFC. Further, the resultant fractions were collected and analyzed by LC-ESI-MS and collected as individual fractions to obtain pure PMF compounds. 

### 4.6. Reverse-Phase High-Performance Liquid Chromatography

It is also known as hydrophobic chromatography as it uses a hydrophobic stationary phase. It is different from normal phase chromatography, which uses silica or alumina resins as stationary phase, and the latter is hydrophilic in nature. The hydrophobic stationary phase in the reverse-phase HPLC consists of covalent bonded alkyl chains, which bind with the hydrophobic or less polar compounds present in the mixture and are carried with the mobile phase. The popular column materials are octadecyl carbon C_18_-bonded silica, C_8_-bonded silica, and cyano- and phenyl-bounded silica. All of these materials are inert and polar substances which provide sufficient packing for an efficient separation process and isolation of phytochemicals. Del Rio et al. studied citrus flavanones and their respective dihydrochalcones through molecular structural elucidation of the compounds separated using reverse-phase HPLC employing C_18_ reverse-phase column and mobile phase consisting of an isocratic-gradient system including water–acetonitrile and acetic acid [223]. In another study, they reported on elucidation of flavanone and flavones extracted from *Citrus aurentium* tissues (fruits and leaves) using reverse-phase chromatography employing a variety of solvents, viz. a mixture of water, methanol, acetonitrile, DMSO, and acetic acid and concluded that DMSO presented the best results regarding separation and isolation of individual compounds or constituent compounds in the exact mixture [224]. The mobile phase employed in this chromatographic process is generally polar (aqueous) solvent, which carries the hydrophilic molecules in the analyte mixture and eluted first, while the hydrophobic molecules remain bonded with the stationary phase. After collecting the hydrophilic polar compounds, the column is run with another solvent with nonpolar properties or organic solvents. In practical uses, the mixtures of water or aqueous buffers and organic solvents are employed, which include acetonitrile and tetrahydrofuran, ethanol, and 2-propanol. Both isocratic (solvent–water composition does not change during the separation–isolation process) and gradient (solvent–water composition changes during the process) are applicable in reverse-phase HPLC. In modern techniques, the added advantage that can be achieved from reversed-phase chromatography is separation of charged analytes using ion-pairing or ion interaction. The latter is also known as reverse-phase ion-pairing chromatography.

### 4.7. Size Exclusion Chromatography

It is also known as molecular sieve chromatography, in which the organic molecules in the extract mixture are separated according to their size, which also defines their molecular weight. This technique is particularly employed to separate large molecules with complex structures, such as proteins fibers, pectin, etc. The column materials employed in this technique are essentially fine porous beads composed of dextran polymers (Sephadex or BioGelP). The molecules in the extract mixture are trapped in the pores of these beads according to their size and facilitate estimation of the dimensions of macromolecules as well as good molar mass distribution [225]. The technique is known as gel-filtration chromatography when aqueous solution is taken as mobile phase and gel permeation chromatography when an organic solvent is employed as mobile phase. Pectin exhibits a complicated behavior in solution due to varying degrees of methoxylated carboxyl groups attached the main *α*-1, 4-linked d-galacturonic acid units. Furthermore, varying amounts of natural sugars present in the pectin composition give rise to a complicated solution behavior observed to be consistent with a delicate compositional balance of hydrophobic/hydrophilic residue [226]. Size exclusion chromatography has also been employed in the recovery of pectin fragments, viz. arabinans, galactans (arabinogalactans) of a relatively lower smaller size ranging between oligomers and polymers [227]. Kuo et al. carried out a series of chromatographic separations on the ethanol extracts of the peels of *C. grandis* followed by the characterization of forty compounds which include seventeen coumarins, eight flavonoids, two steroids, two triterpenoids, one lignan, one amide, and four benzenoids. The work has been summarized in the form of a simplified flow chart in Appendix A, presenting an overall viewpoint on different steps of separation, isolation, and purification procedures [228], and relevant research reports are summarized in Table 13.

## 5. Detection, Analysis, and Structural Determination of Citrus Bioactive Molecules

Structural determination of bioactive compounds extracted from citrus wastes employs a wide range of spectroscopic techniques, e.g., UV-Visible, FTIR, NMR, atomic absorption, and mass Spectroscopy. Spectroscopy involves electromagnetic radiation, which passes through the organic molecule. The latter absorbs a certain amount of the incident radiation and transmits the rest. This absorbed amount of the electromagnetic radiation is measured, and analysis is carried out using the spectrum produced. The absorption of electromagnetic radiation is specific to certain bonds present in the molecule. In this regard, spectra produced from four different regions of the electromagnetic band are utilized, viz. UV-Visible, infrared, radiofrequency, and electron beam. The elemental analysis or determination of minerals is carried out using atomic adsorption spectroscopy (AAS).

### 5.1. Ultraviolet-Visible and Infrared Spectroscopy

It is a very powerful technique to identify saturation in molecule aromatic moiety and other chromophores which absorbs in UV range. Phenolic compounds are identified in UV-Visible range. These include anthocyanins, tannins, polymer dyes, phenolic acid, flavonoids, coumarins, limonoids, sugars, and pigments. The characteristic absorption peak for total phenolic extract is 280 nm, flavones (320 nm), phenolic acids (320 nm), and total anthocyanids (520 nm). The absorption peak corresponding to naringin, hesperidin, and didymin appears at λ = 210–227 and λ = 283–285 nm; tangeretin appears at λ = 210, 250, 270, and 334 nm and nobiletin appears at λ = 210, 271, and 324 nm [240]. A list of absorption peak positions for differen citrus phytochemicals is presented in Table 14 and Table 15. Absorption of IR radiation depends on the kind of vibration changes which occur in a polar molecule when exposed to IR radiation. Therefore, it is also known as vibrational spectroscopy. To obtain a response from IR spectroscopy, the molecule under investigation has to be a polar molecule with a dipole moment. Different bonds that exhibit characteristic absorption/transmission peaks in the spectrum are C–C, C=C, C≡C, C=O, C–O, C=O, O–H, N–H, etc. IR spectrum is helpful in detecting these types of molecules bonding present in the molecule. Fourier transformed infrared spectroscopy (FTIR) is an advanced analytical tool offering high-resolution analytical data to elucidate complex molecules. FTIR is a time-efficient and cost-effective as well as nondestructive investigation technique to furnish fingerprints for structural details of extracted biomolecules.

### 5.2. Nuclear Magnetic Resonance Spectroscopy (NMR)

This technique utilizes the magnetic properties of certain atomic nuclei, e.g., hydrogen atom, the proton, ^1^H, carbon atom, and an isotope of carbon, ^13^C. The spectrum obtained helps in analyzing the types and number of these nuclei in the molecule. Moreover, this also gives an idea about the surrounding environment, i.e., types of atoms in the vicinity of these nuclei, and can be quantified by measuring the chemical shifts in the spectrum. The separation, isolation, and purification steps employing preparative, semipreparative, and thin layer chromatography as well as HPLC are often equipped with NMR for the structural determination of the molecules. An elaborated work on separation, purification, and determination of hydroxylated polymethoxyflavones and methylated flavonoids in sweet orange (*C. sinensis*) peels using ^1^H and ^13^C NMR can be found in [241]. Cordenonsia et al. identified and confirmed the presence of two flavanones, naringin and naringenin in Pomelo (*Citrus máxima*) by ^1^H and ^13^C NMR [242]. Villa-Ruano et al. employed the ^1^H-NMR technique to study the metabolomics profiling of citrus juices and recoded data of 35 metabolites including amino acids, sugars, and organic acids [243]. Cicero et al. used high-resolution magic angle spinning (HR-MAS) NMR for the quantitative analysis of the major metabolites present in the lemon juice of two citrus lemon hybrids, viz. PGI Interdonato lemon of Messina and Interdonato Turkish lemon [244]. The aim was to develop an analytical technique for rapid determination and comparison of the metabolic fingerprints of different commercial products present in national and international markets. For crude samples/mixtures, NMR can be used for the analysis of a broad range of metabolites, including primary and secondary metabolites. However, LC–MS can detect and quantify the metabolites, which are undetectable using NMR spectroscopy with high sensitivity. 

### 5.3. Mass Spectroscopy

MS techniques are extensively employed today to determine the molecules known to provide health benefits, such as flavonoids and other polyphenols. In this technique, the organic molecule is subjected to a bombardment of electron beams or laser light exposure, thereby leading to conversion of the former into fragments. The electric and magnetic fields generated inside the instrument separate the different size fragments according to their mass-to-charge ratio (*m*/*z*). The latter are highly energetic species and move towards the detector with a certain kinetic energy depending on their molecular mass. The charged fragment molecular ions are denoted by [M + H]^+^ or [M − H]^−^, and by knowing the *m*/*z* ratio, the molecular mass of the parent molecule can be deduced. A spectrum is constructed on the basis of the relative abundance of fragmented ions against the ratio of mass/charge. This technique enables determining the relative molecular mass of the organic molecule and exact molecular formula of the same through an analysis of the types of fragments produced during the process. For, example a fragmentation pathway of 5,6,7,4′-tetramethoxyflavanone is shown in Appendix A [241]. This technique provides abundant information required for the structural elucidation of the bioactive molecules extracted from citrus waste. The separation and purification process carried out by HPLC instruments is often equipped with mass spectroscopy facilities for a rapid and accurate identification of the compounds even when purified standards are not available. In recent years, LC/MS has been extensively employed for the analysis of phenolic compounds. Furthermore, electrospray ionization (ESI) is also considered to be a preferred tool due to high ionization efficiency for the fragmentation of phenolic compounds. Depending on various types of ionization sources which are employed as an interface between liquid chromatography (LC) and mass spectrometer (MS), the efficiency of the analysis can be enhanced in terms of improved sensitivity, enabling high throughput analysis of phytochemical molecules. The different ionization sources commonly employed for the fragmentation of complex phytochemical molecules are electron impact (EI), fast atom bombardment (FAB), matrix-assisted laser desorption/ionization (MALDI), electrospray (ESI), atmospheric pressure chemical ionization (APCI) and, more recently, atmospheric pressure photoionization [245] and atmospheric pressure solids analysis probe (ASAP). ESI/MS and APCI/MS offer an excellent mass range and sensitivity for analyzing bioflavonoids [246]. The LC/MS–APCI technique involves charge transfer from dopant molecules (e.g., toluene) to the analytes, and the latter are ionized using photons produced by a vacuum UV lamp. MALDI is useful in analyzing nonvolatile compounds [247]. Recently, MALDI/TOF/MS has emerged as an increasingly popular analytical technique for determining a wider range of biomolecules, viz. bioflavonoids, peptides, and proteins. HPLC paired with UV photodiode array and ESI tandem mass spectrometry detectors (HPLC-PDA-ESI/MS) have been found efficient in determining phenolic compounds and their tentative molecular structures are further confirmed by employing NMR spectroscopy [248]. Gas chromatography (GC)/MS is mainly used for volatile analytes, such as essential oils. It is rarely used in flavonoid analysis owing to the limited volatility of flavonoid glycosides. APCI provides greater sensitivity towards the charged ions and hence has been found to be very useful in analyzing carotenoids. APCI has become the most widely used ionization technique for carotenoids because of its high sensitivity. Hao et al. reported on the detection of main carotenoids *β*-carotene, *β*-cryptoxanthin, lutein, and zeaxanthin in botanical samples with a 100-fold higher sensitivity using APCI rather than ESI [249]. Summarized descriptions based on recent scientific reports on identification, determination, and analysis of citrus phytochemicals via different chromatography methods assisted with spectroscopic techniques are arranged in Table 14, and identification fingerprints information ([M + H]^+^ and λ_max_) obtained through mass and UV-visible spectroscopic techniques are listed in Table 15.

Citrus is the largest single fruit crop industry. Citrus peel waste is utilized to produce important commercial products, such as dietary pectin, vitamins, sugars, essential oils, molasses, volatile flavoring compounds, organic acids, antioxidants, etc. [273]. There are many more important value-added products obtained from citrus wastes, namely, vinegar, citric acid, lactic acid, ethanol, feed yeast, cattle feed, gluconate, fructose, etc. Citrus byproducts are utilized to manufacture more than four hundred types of commercial products. These include medicines (diarrhea rehydration solution, vitamin C tablets or solution, ointments, antiseptics, etc.), cosmetics and toiletries, candles, vinegar, various food products and nutrition supplements, dietary fibers, cleaners, preservatives for animal feeds, and so on. Figure 12 summarizes some of the commercial products available to us. All the photographs were taken from a local supermarket at Gyeongsan, South Korea. The pictures displayed here in this article are purely for academic purposes and not for any advertising purposes. 

## 6. Summary and Future Perspectives

In recent years, researchers across the world have been focusing on developing various processing methods for maximum exploitation as well as utilization of various waste products of citrus fruits. The purpose of this review is to describe the important aspects of citrus waste reuse and management; extraction of different bioactive molecules and various citrus byproducts beneficial for food, nutrition, human health, and economy. Dumping of untreated citrus wastes can cause hazardous effects on the environment in terms of pollution and adverse effects on the ecosystem. Extraction and utilization of bioactive molecules not only enable inexpensive means of obtaining chemicals of economic importance but also exploiting the renewable biomass which continues to grow every year.

In the citrus industries, the pulp wastes remaining after centrifugation and extraction of juice are generally treated with alkali or enzymes to obtain animal feed. Enzymatic treatment is considered as one of the most efficient methods for processing citrus pulp. The products obtained by this method have been shown to have high protein content. However, the protein content is not as high compared to other agro-industrial waste products currently used as components of animal feed. Furthermore, processed citrus waste has shown excellent digestibility in vivo. Citrus peel wastes mainly from oranges and grapefruits are successfully hydrolyzed into simple hexose and pentose sugars like glucose, fructose, galactose, xylose, arabinose, rhamnose, and galacturonic acid (GA). This is accomplished using a mixture of cellulolytic and pectinolytic enzymes. Subsequent fermentation of such compounds to ethanol using either *Saccharomyces cerevisiae* or recombinant strains of *Escherichia coli* or a combination of both has been reported [180,274]. Citrus wastes are best utilized to obtain fibers and food ingredients, e.g., pectins and mucilages [275,276]. The solid and highly concentrated liquid citrus wastes are transformed into citrus molasses, feed yeast, lactic acid, industrial alcohol, vinegar, etc. [22]. In addition to this, many value-added compounds or phytochemicals of industrial importance are efficiently extracted from citrus wastes and utilized in several ways. 

‘Citrus’ itself is a complete and profitable industry. With a continuously increasing production and utilization of citrus fruits and various citrus products across the globe and anticipation of huge progress in the future years, it has now become crucial to address this topic. With integrated research and active collaboration between different scientific and engineering fields, it is possible to achieve maximum utilization of citrus waste and explore large-scale applications. However, there remain several lacunae in the following areas, viz. (a) elaborated profiling of chemical constituents of citrus waste with an aim of detecting as well as extracting higher amounts of bioactive compounds and value-added products, (b) design of process engineering and chemical plants that can adapt to a variety of sources and facilitate production, (c) development of efficient microbial strains for effectively converting citrus wastes into high-value products using either classic mutagenesis or metabolic engineering, (d) information sharing between breeders/cultivators and citrus processing units/industries, waste management plants, and collaboration of industrial/academic researchers, (e) policy on systematic management of citrus waste integrated with modern scientific techniques, (g) design of academic curricula/programs in educational centers and universities and construct workshops to increase awareness, work skills, and efficiency, and (h) efficient transport facilities. The exploitation of citrus bioactive molecules in food and neutraceutics, dietary supplements, and byproducts in food and beverages and pharmaceutics indicates a promising field of future research. Some phytochemicals, such as *d*-limonene, nomilin, and nitrates, which have an adverse effect on the processing methods, must be excluded by efficient quality control systems in food processing units. Detailed investigations on their bioactivity, toxicology, stability, and interactions with other food ingredients or packaging materials should be carried out with a careful assessment in vitro and in vivo. This requires active collaboration and interdisciplinary research of food technologists, food chemists, nutritionists, and toxicologists so as to develop efficient methods of processing of citrus fruits as well as their waste, innovating a user-friendly apparatus and equipment to yield products in small as well as large scales and standardize information and databases on this area of research to inculcate interest among the young generation, or in other words, to provide education to shape the future. 

## Figures and Tables

**Figure 1 foods-08-00523-f001:**
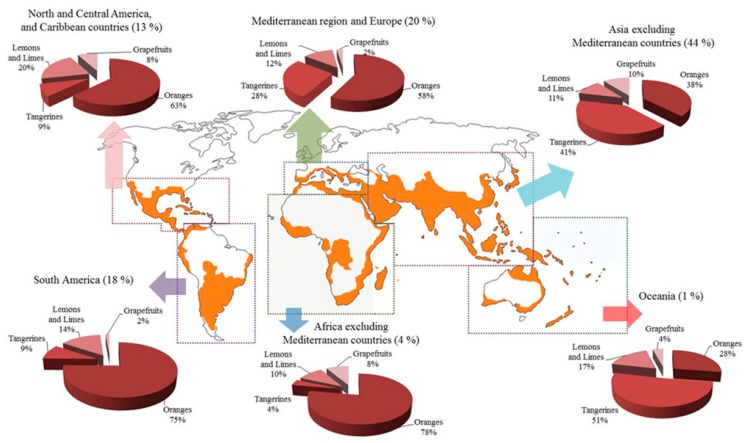
Climate sustainability and the annual production of citrus fruits in different geographical regions across the globe [3,4,15].

**Figure 2 foods-08-00523-f002:**
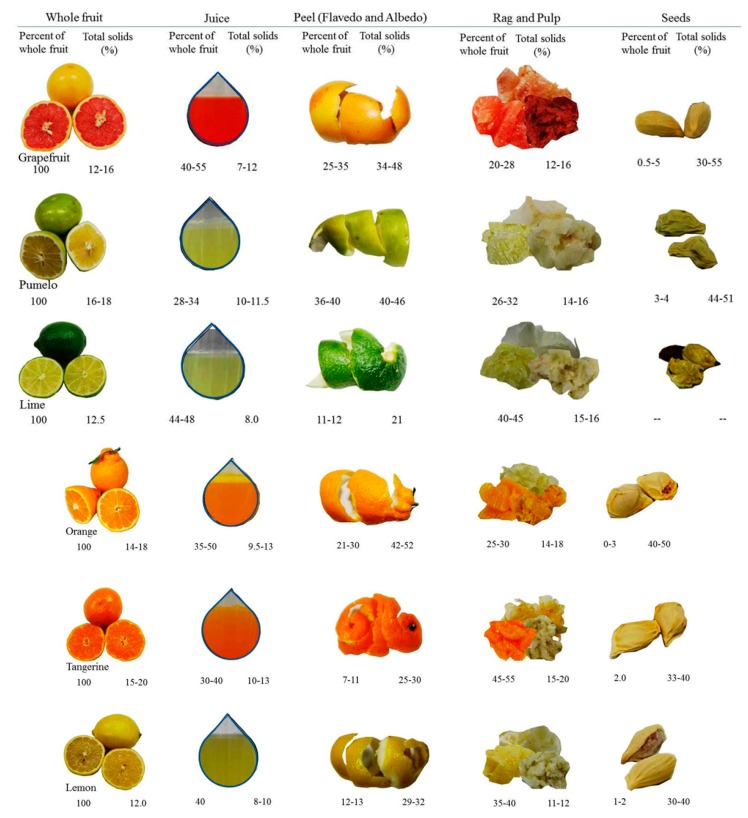
The main citrus varieties and its physical composition in terms of edible juice and inedible waste part, which consists of peel (flavedo and albedo), rags (pith, pulp residue, and segment membrane), and seeds [10,13,21].

**Figure 3 foods-08-00523-f003:**
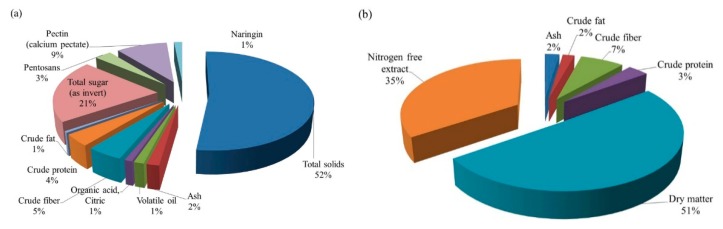
(**a**) The composition of typical citrus waste (peel and rag); (**b**) the composition of dried citrus pulp [9,22,23,24,25,26,27].

**Figure 4 foods-08-00523-f004:**
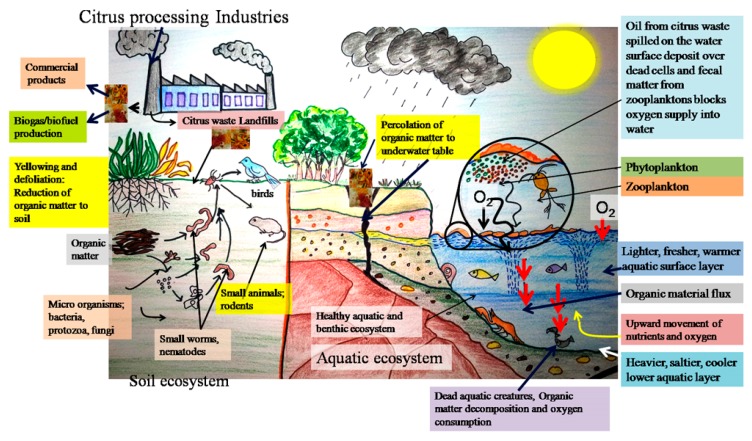
Schematic diagram illustrating the risks and potential threat of damage to soil and aquatic ecosystems and the overall environment by the disposal of untreated citrus wastes.

**Figure 5 foods-08-00523-f005:**
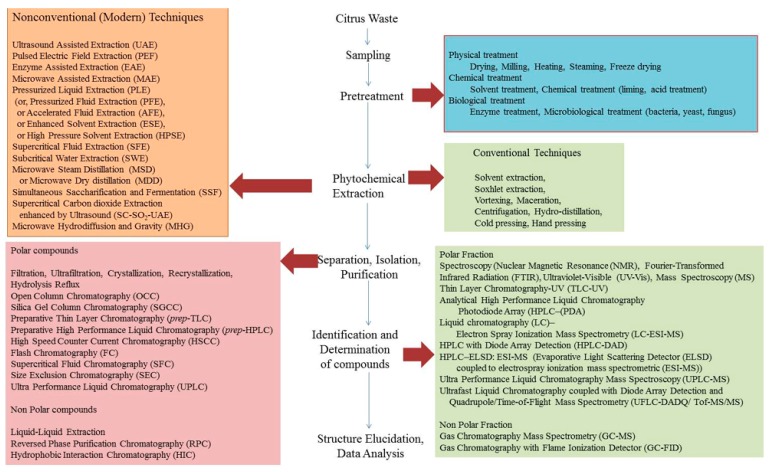
The steps involved in the extraction, separation, isolation, and purification of important compounds from citrus waste and techniques employed for determination and structural elucidation.

**Figure 6 foods-08-00523-f006:**
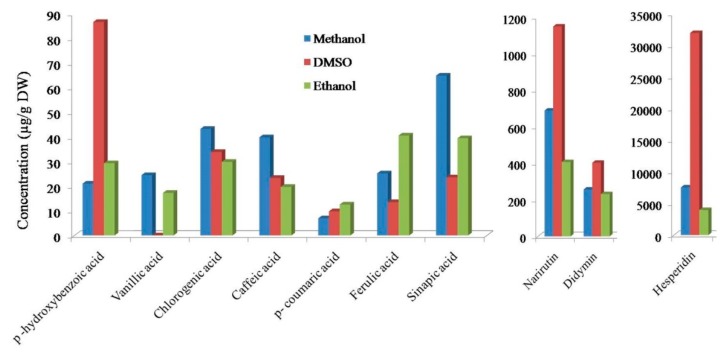
Effect of the nature of solvents on the yield of total phenolics by the solvent extraction process [58].

**Figure 7 foods-08-00523-f007:**
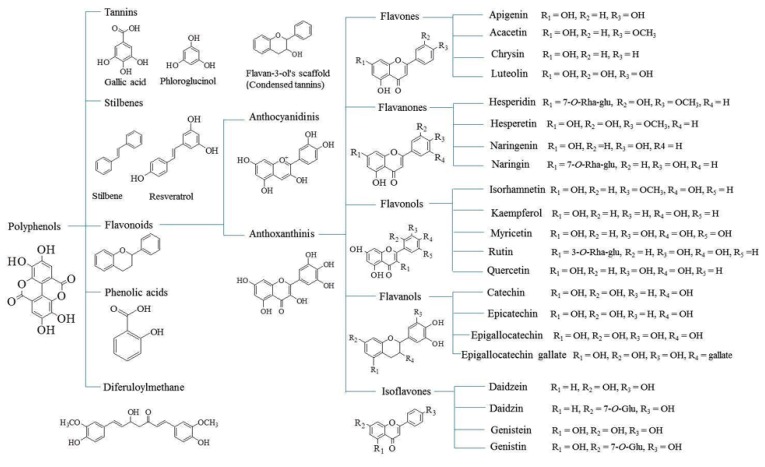
Classification of citrus polyphenols [11].

**Figure 8 foods-08-00523-f008:**
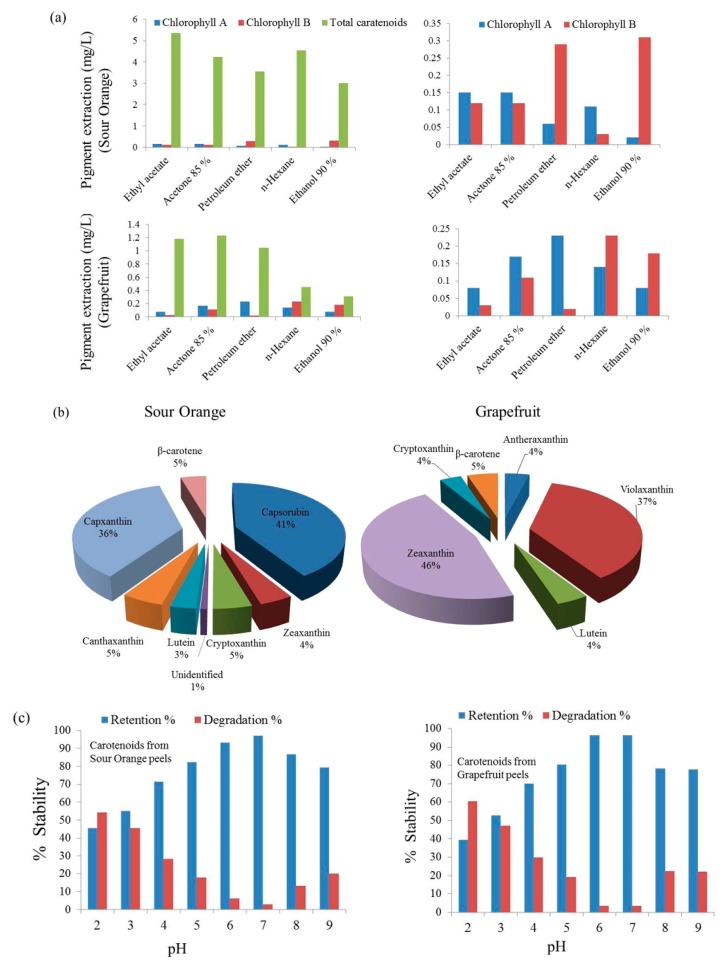
(**a**) Extraction of pigments in different solvents, (**b**) pigment composition, and (**c**) stability of the extracted pigments at different pH [122].

**Figure 9 foods-08-00523-f009:**
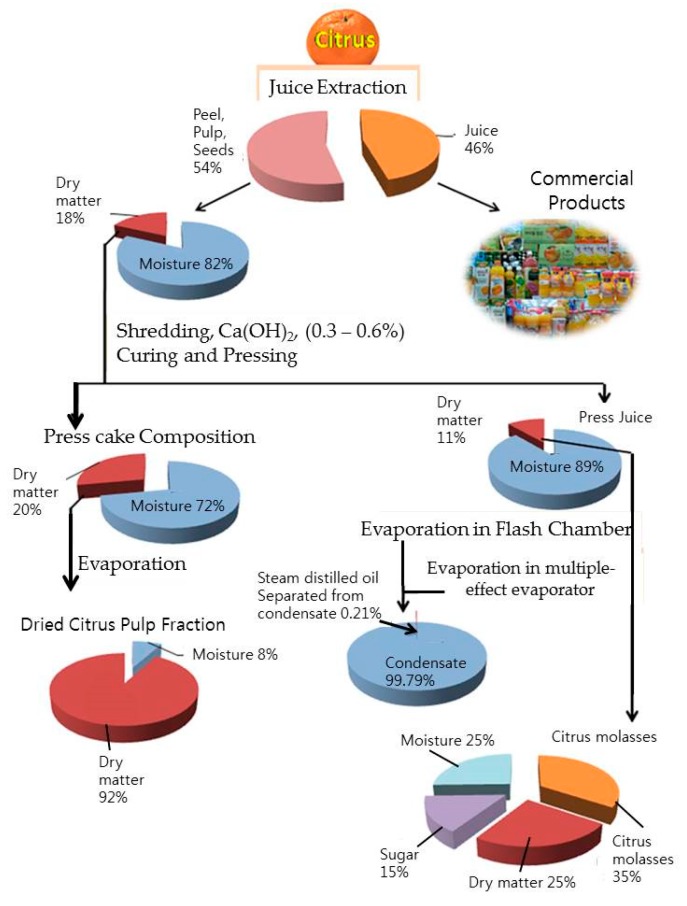
Stages of typical citrus waste processing to generate molasses [22].

**Figure 10 foods-08-00523-f010:**
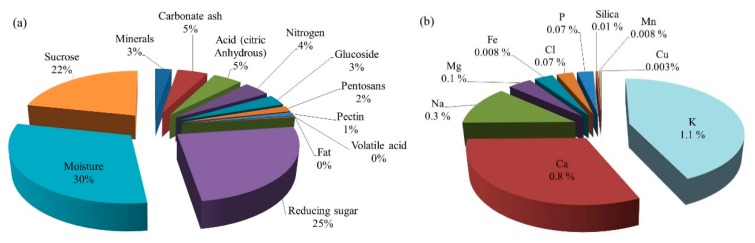
(**a**) Composition of citrus molasses; (**b**) mineral composition of citrus molasses [9,22,23,24,25,26,27].

**Figure 11 foods-08-00523-f011:**
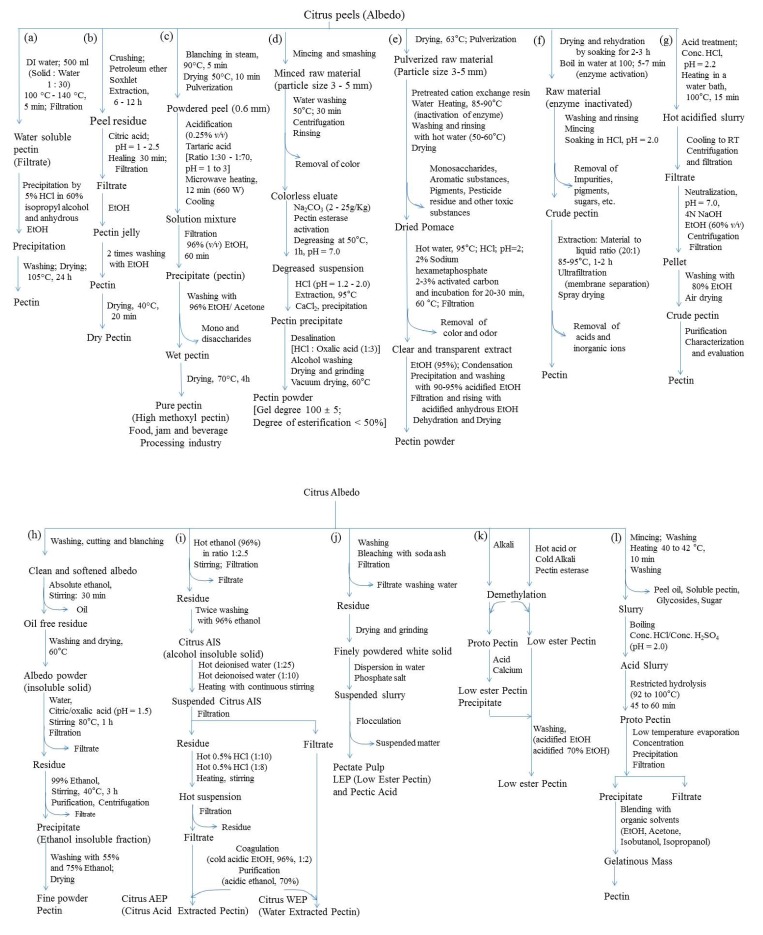
Steps involved in the different extraction methods for obtaining pectin from citrus waste [65,155,156]

**Figure 12 foods-08-00523-f012:**
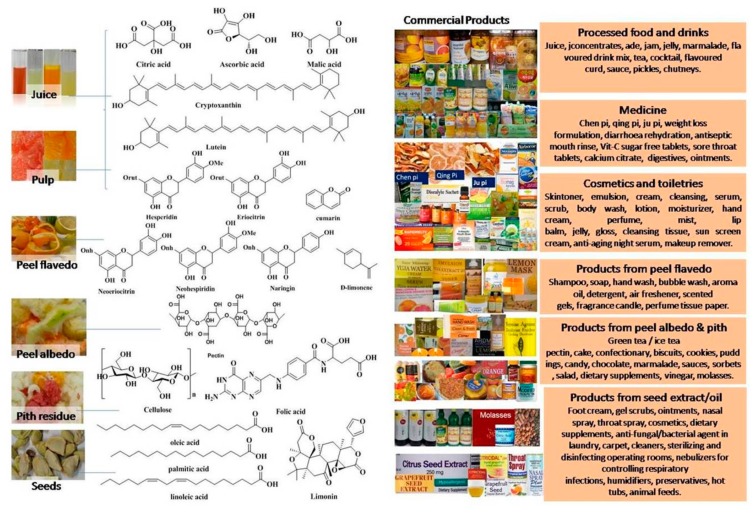
Byproducts obtained from citrus waste processing are utilized in manufacturing various types of commercial products. The photographs were collected from a local supermarket and have been used for academic purposes only and not for any advertising purposes.

**Table 1 foods-08-00523-t001:** The composition of citrus wastes acquired from juice-producing industries; Units are expressed in (g/100 g).

Composition	Citrus Byproduct	Peel	Pulp
Moisture content	8.5 ± 0.20 ^a^	75.3 ± 10.20 ^c^	85.7 ± 0.00 ^d^
Crude protein	12.5 ± 0.87 ^b^	10.2 ± 3.70 ^c^	8.6 ± 0.00 ^d^
Crude fiber	75.7 ± 2.10 ^a^	57.0 ± 10.0 ^c^	7.3 ± 0.80 ^e^
Crude fat	0.5 ± 0.02 ^b^	2.2 ± 6.10 ^c^	4.9 ± 0.00 ^f^
Total ash	8.1 ± 0.41 ^b^	3.3 ± 0.50 ^c^	6.5 ± 0.00 ^d^

Reference: ^a^ [16]; ^b^ [9]; ^c^ [17]; ^d^ [18]; ^e^ [19]; ^f^ [20].

**Table 2 foods-08-00523-t002:** The biological oxygen demand of citrus wastewaters generated during different processing operations [32].

Operation in Citrus Industry	Biological Oxygen Demand (mg/L)
Fruit washing	500
Juice extraction	500–1500
Juice evaporator condenser water	0–500
Molasses evaporator condenser water	0–500
Sectionizing	5000–10,000
Essential oil recovery	20,000–45,000
Peel bin drip	60,000–120,000
Pressed liquor	60,000–120,000

**Table 3 foods-08-00523-t003:** Polyphenol content in different citrus varieties across the globe and the effect of parameters involved in the extraction process.

	Total Phenolic Content		Antioxidant Activity		Extraction Method	Ref.
	**Pulp**	**Peel**	**Pulp**	**Peel**		
Orange (Orlando, *Citrus sinensis*, Saudi Arabia)	123.02	178.90	69.31	67.58	Ethanol (80%), solvent extraction, 70 °C, 3 h TPC—mg GAE/100 g Antioxidant activity—antiradical activity against DPPH radicals (%)	[46]
Lemon (Eureka, *Citrus limon*, Saudi Arabia)	98.38	61.22	59.60	68.57		
Mandarin (Kinnow, *Citrus reticulata*, Saudi Arabia)	104.98	169.54	62.82	46.98		
	**Seeds**	**Peel**	**Seeds**	**Peel**		
Lemon (Guangzhou, China)	13.58	1.99	^a^ 1.37 ^b^ -	^a^ 5.76 ^b^ 5.06	THF (nonpolar)/methanol-acetic acid-water (polar) (50:3.7:46.3, *v*/*v*); Solvent extraction 37 °C, 30 min TPC—mg GAE/g of peel/seed Antioxidant activity— ^a^ FRAP (µmol Fe(II/)/g) ^b^ TEAC (µmol Trolox/g)	[47]
Mandarin (Guangzhou, China)	2.77	3.64	^a^ 12.81 ^b^ 4.35	^a^ 9.99 ^b^ 8.61		
Green pomelo (Guangzhou, China)	3.16	4.25	^a^ - ^b^ 16.40	^a^ 16.83 ^b^ 21.98		
Tangerine (Guangzhou, China)	3.02	3.52	^a^ 11.49 ^b^ 7.67	^a^ 14.99 ^b^ 14.28		
Navel orange peel (Coles Broadway, Sydney)	5.00		1290.00		Deionized water, Soxhlet extraction, Spray drying TPC—mg GAE/g Radical scavenging activity against DPPH-SC_50_ (µg DM/mL)	[48]
Mandarin peel (Coles Broadway, Sydney)	7.30		880.00			
Lemon peel (Coles Broadway, Sydney)	6.00		890.00			
Lime peel (Coles Broadway, Sydney)	6.00		740.00			
Grapefruit peel (Israel)	155.00		1667.00		Ethanol (95%), solvent extraction, Boiling in water bath TPC—mg CAE/100 g (mg of chlorogenic acid per 100 g of fresh fruits) Antioxidant activity—TRAP (nmol/mL)	[19]
Sweet orange peel (Israel)	179.00		3183.00			
Lemons peel (Israel)	190.00		6720.00			
White grapefruit peel (*Citrus paradisi*, Israel)	8.40		^a^ 6.31, ^b^ 14.50		MeOH (80%) Solvent extraction; 90 °C; 3 h TPC—µg GAE/g fresh weight Antioxidant activity— ^a^ TAA (mM TE/g FW) ^b^ NO (AE × 10^3^)	[49]
Jaffa sweetie grapefruit peel (*Citrus grandis* × *C. paradisi*, Israel)	13.90		^a^ 8.52, ^b^ 19.30			
Lemons (cv. Meyer) peel (New Zealand)	59.77		0.58		EtOH (72%), solvent extraction, 3 h TPC—mg GAE/g of fresh peel Antioxidant activity FRAP assay (mM FeSO_4_ equivalent/100 g fresh peel)	[50]
Lemons (cv. Yenben) peel (New Zealand)	118.95		1.28			
Grapefruit peel (New Zealand)	161.60		1.72			
Mandarin (cv. Ellendale) peel (New Zealand)	121.14		1.27			
Sweet orange (cv. Navel) peel (New Zealand)	73.59		0.85			
Lemons peel (*Citrus meyeri*, Mauritius)	1882.00–2828.00		^a^ 11.50–13.30 ^b^ 21.20–26.70 ^c^ 10.70–14.36		Solvent extraction; MeOH (80%) TPC—µg GAE/g of fresh weight Antioxidant activity— ^a^ TEAC (µmol Trolox/g FW ^b^ FRAP (µmol Fe^2+^/g FW) ^c^ HOCl (IC_50_) mg FW/mL	[51]
Mandarins peel (*Citrus reticulate*, Mauritius)	2649.00–6923.00		^a^ 15.10–44.00 ^b^ 20.90–81.30 ^c^ 3.70–14.24			
Sweet orange peel (*C. sinensis*, Mauritius)	4509.00–6470.00		^a^ 21.60–31.20 ^b^ 37.60–56.70 ^c^ 3.98–6.57			
*Citrus sinensis* peel (Algeria)	12.09		^a^ 337.16 ^b^ 482.27		Acetone in water (20–80%), microwave-assisted extraction (MAE), 90–240 s TPC—mg GAE/g dry weight Antioxidant activity— ^a^ DPPH radical scavenging assay ^b^ ORAC assay DPPH radical	[52]
*Citrus sinensis* peel (Algeria)	10.35		^a^ 433.08 ^b^ 456.94		Acetone in water 76%, UAE, 8.33 min	
*Citrus sinensis* peel (Algeria)	6.26		^a^ 450.44 ^b^ 337.97		1 g peel powder placed in two layers of diatom earth and extracted with 50% acetone Accelerated solvent extraction (ASE); 120 °C	
*Citrus sinensis* peel (Algeria)	10.21		^a^ 358.45 ^b^ 523.04		50% aq. acetone; CSE; 60 °C, 2 h	
Citrus juice byproduct (Industry, Brazil)	386.00		^a^ 11,035.00 ^b^ 91,570.00		Aqueous ethanol solution (1:1); Ultrasonication, 30 °C, 15 min TPC—mg GAE/100 g dry matter Antioxidant activity— ^a^ DPPH radical scavenging activity (µmol TE/g DM) ^b^ ORAC (TE/g DM); TE—Trolox eq.	[53]
Citrus pectin byproduct (Industry, Brazil)	170.00		^a^ 2571.00 ^b^ 37,588.00			
Lemon peel (Sarajevo, B&H)	0.48		20.30		Ethanol, ultrasonication TPC—mg GAE/mL Antioxidant activity— IC_50_—concentration of the extract that reduces 50% of Molebdenum ions (total antioxidant capacity spectroscopic method)	[54]
Orange peel (Sarajevo, B&H)	0.45		19.15			
Mandarin peel (Sarajevo, B&H)	0.33		9.13			
Red grapefruit peel (Sarajevo, B&H)	0.28		24.52			
White grapefruit peel (Sarajevo, B&H)	0.19		30.93			

The subscripts (^a^, ^b^ and ^c^) are denoted for the different antioxidant activity methods employed—these exhibit antioxidant properties. TPC-Total Phenolic Content; GAE-Gallic Acid Equivalent; DPPH-2,2-diphenyl-1-picrylhydrazyl; THF-Tetrahydrofuran; FRAP-Ferric Reducing Ability of Plasma; TEAC-Torlox Equivalent Antioxidant Capacity; DPPH-SC_50_-(otherwise called the IC_50_ value), the concentration of the antioxidant causing 50% DPPH^•^ scavenging, or as %scavenging of DPPH^•^ at a fixed antioxidant concentration for all the samples; TAA-Total antioxidant activity; ORAC-oxygen radical absorbance capacity; CSE-cigarette smoke extract; TE-Trolox Equivalents.

**Table 4 foods-08-00523-t004:** Composition of major flavonoids determined from methanolic extraction of peels of different citrus fruits cultivated in Taiwan [36].

Composition	*C. reticulata* Blanco (Ponkan)	*C. tankan* Hayata (Tonkan)	*C. reticulata* × *C. sinensis* (Murcott)	*C. grandis* Osbeck (Wendun)	*C. grandis* Osbeck CV (Peiyou)	*C. microcarpa* (Kumquat)	*C. sinensis* (L.) Osbeck (Liucheng)	*C. limon* (L.) Bur (Lemon)
Total Flavonoid	49.20 ± 1.33	39.60 ± 0.92	39.80 ± 1.02	46.70 ± 1.51	48.70 ± 1.53	41.00 ± 1.37	35.50 ± 1.04	32.70 ± 1.06
Flavanone (mg/g, dried base)								
Naringin	0.54 ± 0.02	0.58 ± 0.01	0.54 ± 0.02	23.90 ± 0.32	29.80 ± 0.20	0.21 ± 0.01	0.36 ± 0.00	1.51 ± 0.05
Hesperidin	29.50 ± 0.32	23.40 ± 0.25	0.93 ± 0.04	0.32 ± 0.00	0.34 ± 0.01	0.10 ± 0.00	20.70 ± 0.38	9.42 ± 0.41
Neohesperidin	0.11 ± 0.00	0.06 ± 0.00	0.13 ± 0.00	0.34 ± 0.00	0.09 ± 0.00	0.02 ± 0.00	0.09 ± 0.00	0.16 ± 0.00
Flavone (mg/g, dried base)								
Diosmin	0.36 ± 0.01	0.33 ± 0.01	0.40 ± 0.01	0.16 ± 0.01	0.12 ± 0.01	1.12 ± 0.03	0.17 ± 0.00	0.13 ± 0.00
Luteolin	0.21 ± 0.01	0.19 ± 0.01	0.20 ± 0.01	nd	nd	nd	0.11 ± 0.00	0.08 ± 0.00
Sinensetin	0.29 ± 0.01	0.41 ± 0.01	0.30 ± 0.01	0.02 ± 0.00	0.06 ± 0.02	0.01 ± 0.00	0.42 ± 0.01	0.22 ± 0.01
Flavonol (mg/g, dried base)								
Rutin	0.29 ± 0.00	0.26 ± 0.01	0.25 ± 0.00	0.18 ± 0.00	0.17 ± 0.00	0.09 ± 0.00	0.23 ± 0.00	0.29 ± 0.00
Quercetin	0.47 ± 0.00	0.26 ± 0.02	0.15 ± 0.00	0.23 ± 0.00	0.19 ± 0.00	0.78 ± 0.00	0.14 ± 0.00	0.21 ± 0.00
Kaempferol	0.38 ± 0.00	0.27 ± 0.00	0.20 ± 0.00	0.33 ± 0.00	0.13 ± 0.00	0.15 ± 0.00	0.32 ± 0.00	0.31 ± 0.00

**Table 5 foods-08-00523-t005:** Phenolic acid composition of peels and seeds in different varieties of citrus.

Extract	Caffeic Acid	*p*-Coumaric Acid (Cis- and Trans-)	Ferulic Acid	Sinapinic Acid	Extraction Method	Ref.
Lemon seed (*Citrus Limon* Femminello Comune, Italy)	0.02	0.07	0.05	0.05	Alkaline hydrolysis of finely powdered peel/seed (bound phenolic compounds) Unit-mg/g DW	[34]
Sweet orange seed (*C. bergamia* Fantastico, Italy)	0.01	0.02	0.05	0.07		
Sour orange peel (*C. aurantium*, Italy)	0.23	0.19	1.58	0.95		
Bergamot peel (*C. sinensis* Biondo Comune, Italy)	0.01	0.07	0.04	0.03		
*Citrus unshiu Marc* peels (Zhejiang, China)	12.50	23.00	189.00	40.00	80% Methanol; Maceration extraction, 1 h, 40 °C; Unit-µg/g DW	[61]
*Citrus unshiu Marc* peels (Zhejiang, China)	31.70	63.20	763.50	132.39	80% Methanol; Maceration extraction, 8 h, 40 °C Unit-µg/g DW	
*Citrus unshiu Marc* peels (Zhejiang, China)	50.90	97.90	1187.60	218.20	80% Methanol; Ultrasonic Assisted Extraction; 40 °C, 1 h Unit-µg/g DW	
*Citrus unshiu Marc* peels (Zhejiang, China)	97.50	177.30	2226.80	219.80	80% Methanol; Ultrasonic Assisted Extraction; 15 °C, 1 h Unit-µg/g DW	
Orange peels (*Citrus sinensis,* Spain)	9.50	27.90	39.20	34.90	Solvent extraction; 95% EtOH, Reflux Unit-mg/g DW	[19]
Lemon peels (*Citrus limon,* Spain)	14.20	34.90	44.90	42.10		
Grapefruit peels (*Citrus paradisi,* Spain)	5.60	13.10	32.30	31.90		
Orange peels (Liucheng, *C. sinensis* (L.) Osbeck), Taiwan	12.60	229.10	45.30	44.90	Solvent extraction 0.1 g sample in 1 mL MeOH-DMSO (50:50, *v*/*v*) Stirring and centrifugation Unit-µg/g DW	[36]
Mandarin peels (Ponkan, *C. reticulata* Blanco), Taiwan	3.06	346.00	150.00	94.20		
Lemon peels (*C. limon* (L.) Bur), Taiwan	80.00	264.10	59.10	59.60		
Kumquat peels (*C. microcarpa*), Taiwan	17.30	41.70	52.70	49.50		
Pumelo peels (Peiyou; *C. grandis* Osbeck CV), Taiwan	27.50	241.00	32.20	29.20		
Pumelo peels (Wendun, *C. grandis* Osbeck), Taiwan	8.22	142.00	30.30	10.10		
Mandarin peels (Murcott, *C. reticulata* × *C. sinensis*), Taiwan	7.23	183.00	145.00	178.00		
Tonkan peels (*C. tankan* Hayata), Taiwan	7.30	319.00	139.00	162.00		
Satsuma Mandarin peels (*C. unshiu Marc*), China	143.70	299.70	2755.60	194.90	Hot water extraction Unit-µg/g DW	[98]

**Table 6 foods-08-00523-t006:** (**a**) Limonoid glucosides in various citrus fruit seeds extracted in hexane and estimated from reverse-phase HPLC; (**b**): composition of limonoid aglycons in various citrus fruit seeds [101]; (values are expressed in mg/g of dry seeds).

**(a) Seeds**	**Deacetylnomilinic Acid Glucoside**	**Nomilin Glucoside**	**Nomilinic Acid Glucoside**	**Obacunone Glucoside**	**Limonin Glucoside**	**Deacetylnomilin Glucoside**	**Total**
Fukuhara (*C. sinensis* Osbeck Hort.)	0.28	3.22	0.98	1.09	0.51	1.32	7.40
Hyuganatshu (*C. tamurana* Hort. Ex Tanaka)	0.42	1.10	0.76	0.65	Trace	0.37	3.31
Sanbokan (*C. sulkata* Hort. Ex Tanaka)	0.37	1.13	0.55	0.90	0.51	0.89	4.36
Shimamikan (*C. kinokuni* Hort. Ex Tanaka)	0.48	1.89	1.29	2.35	0.37	0.69	7.08
Grapefruit (*C. paradisi*)	0.75	2.01	0.89	0.86	1.48	0.68	6.67
Lemon (*C. limon*)	0.14	1.53	1.39	1.49	1.44	0.55	6.54
Valencia orange (*C. sinensis*)	0.13	4.48	0.98	1.06	0.59	1.69	8.94
Tangerine (*C. reticulata*)	1.69	0.42	0.96	0.45	0.90	0.93	5.36
**(b) Seeds**	**Limonin**	**Nomilin**	**Obacunone**	**Ichangin**	**Deacetylnomilin**		**Total**
Fukuhara (*C. sinensis* Osbeck Hort.)	9.77	3.88	0.37	Trace	2.14		15.92
Hyuganatshu (*C. tamurana* Hort. Ex Tanaka)	4.68	3.73	0.28	Trace	0.35		9.04
Sanbokan (*C. sulkata* Hort. Ex Tanaka)	3.95	1.02	0.30	0.16	1.36		6.79
Shimamikan (*C. kinokuni* Hort. Ex Tanaka)	7.85	2.01	0.28	0.16	2.01		12.31
Grapefruit (*C. paradisi*)	19.06	1.84	1.86	Trace	1.10		23.86
Lemon (*C. limon*)	8.95	3.03	0.58	Trace	Trace		12.56
Valencia orange (*C. sinensis*)	10.00	2.30	0.08	1.16	1.24		14.78
Tangerine (*C. reticulata*)	4.10	1.37	0.35	0.38	3.11		9.31

**Table 7 foods-08-00523-t007:** Composition of coumarins in some Columbian citrus fruits determined by HPLC-DAD (diode array detection) from methanolic extract of peels [108]; (values are expressed in µg/g of fresh weight).

Citrus Species	5-Geranyloxy-7 methoxycoumarin	Bergamottin	Bergapten + Isopimpinellin	Limettin	Oxypeucedanin Hydrate	Other Furanocoumarins
Tahitian lime (*C. latifolia*)	392 ± 19	349 ± 17	168 ± 80	183 ± 90	357 ± 19	217 ± 11
Key lime (*C. aurentifolia*)	352 ± 18	302 ± 15	128 ± 60	145 ± 70	256 ± 13	292 ± 15
Sweet orange (*C. sinensis*)	traces	traces	n.d.	n.d.	93 ± 50	n.d.
Mandarin lime (*C. limonia*)	traces	traces	44 ± 20	100 ± 50	478 ± 24	41 ± 2
Mandarin (*C. reticulata* var. Oneco)	traces	traces	48 ± 20	n.d.	569 ± 28	n.d.
Mandarin (*C. reticulata* var. Arrayana)	traces	traces	52 ± 30	n.d.	570 ± 29	n.d.

**Table 8 foods-08-00523-t008:** Composition of pigments determined from the peels of different citrus varieties grown in Taiwan [36].

Pigments	*C. reticulata* Blanco (Ponkan)	*C. tankan* Hayata (Tonkan)	*C. reticulata* × *C. sinensis* (Murcott)	*C. grandis* Osbeck (Wendun)	*C. grandis* Osbeck CV (Peiyou)	*C. microcarpa* (Kumquat)	*C. sinensis* (L.) Osbeck (Liucheng)	*C. limon* (L.) Bur (Lemon)
Total carotenoids (mg/g, dried base)	2.04 ± 0.04	1.42 ± 0.07	1.59 ± 0.01	0.04 ± 0.00	0.02 ± 0.00	0.74 ± 0.03	0.45 ± 0.01	0.11 ± 0.00
Lutein (µg/g, db)	7.75 ± 0.33	7.10 ± 0.25	13.30 ± 0.51	0.80 ± 0.04	1.03 ± 0.04	36.40 ± 1.56	29.30 ± 1.17	2.95 ± 0.12
Zeaxanthin (µg/g, dried base)	6.46 ± 0.29	11.60 ± 0.58	25.20 ± 0.99	0.51 ± 0.02	0.73 ± 0.04	36.40 ± 1.57	27.70 ± 1.21	0.81 ± 0.04
*β*-cryptoxanthin (µg/g, dried base)	30.50 ± 1.26	9.52 ± 0.43	16.90 ± 0.75	0.40 ± 0.02	0.52 ± 0.03	37.00 ± 1.45	0.76 ± 0.04	0.81 ± 0.04
*Β*-carotene (µg/g, dried base)	69.20 ± 2.67	36.90 ± 1.38	12.10 ± 0.51	0.96 ± 0.05	0.84 ± 0.04	2.79 ± 0.14	50.20 ± 2.28	10.30 ± 0.47

**Table 9 foods-08-00523-t009:** Composition of major fatty acids in citrus seeds ^a^ [21] and seed oils ^b^ [135] (in percentage by weight of total fatty acids).

Lipids	Grapefruit ^a^ (Marsh, Trinidad)	Lime ^a^ (Trinidad)	Sweet Orange ^a^ (Jamaica)	Tangerine ^ab^ (Dancy, Florida)	Shaddock ^a^ (India)	Orange ^b^	Grapefruit ^b^	Mandarin ^b^	Lemon ^b^	Lime ^b^
Saturated acids										
Palmitic acid	27.5	26.1	23.8	19.6	20.7	26.0–31.0	3.0–5.0	24.0–28.0	35.0–37.0	2.0–4.0
Stearic acid	2.9	9.6	8.3	5.2	15.3	26.0–36.0	1.0–4.0	18.0–25.0	32.0–41.0	3.0–6.0
Unsaturated acids										
Oleic acid	21.1	11.1	24.8	22.5	55.5	22.0–30.0	2.0–5.0	20.0–25.0	37.0–45.0	3.0–5.0
Linoleic acid	39.3	39.3	37.1	46.6	8.1	18.0–24.0	2.0–4.0	26.0–34.0	31.0–38.0	6.0–12.0
Linolenic acid	5.9	13.1	5.3	2.1	0.5	24.0–29.0	3.0–5.0	20.0–22.0	37.0–40.0	6.0–11.0

^ab^—percentage by weight of total mixed esters.

**Table 10 foods-08-00523-t010:** Pectin composition in the peels of different varieties of citrus fruits grown in Taiwan (mg/g, dried base) [36].

Citrus Variety	Total Pectin	Water-Soluble Pectin
*C. reticulata* Blanco (Ponkan)	37.3 ± 1.83	17.1 ± 0.79
*C. tankan* Hayata (Tonkan)	36.0 ± 1.46	14.6 ± 0.63
*C. reticulata* × *C. sinensis* (Murcott)	61.0 ± 2.41	26.5 ± 1.24
*C. grandis* Osbeck (Wendun)	86.4 ± 3.36	33.3 ± 1.46
*C. grandis* Osbeck cv (Peiyou)	81.9 ± 2.61	29.6 ± 1.09
*C. microcarpa* (Kumquat)	62.1 ± 2.36	27.5 ± 1.10
*C. sinensis* (L.) Osbeck (Liucheng)	43.7 ± 1.62	24.7 ± 1.21
*C. limon* (L.) Bur (Lemon)	65.2 ± 3.25	31.6 ± 1.44

**Table 11 foods-08-00523-t011:** Ascorbic acid composition in different parts of citrus waste (mg/100 g) [21].

Citrus Fruits	Flavedo	Albedo	Rag	Juice
Oranges (4 varieties), Florida	334	182	59	62
Oranges, Shamouti, Israel	236	123	42	41
Oranges, Italy	175–292	86–194	Rag and Juice 45–73	
Mandarins, Japan	Flavedo and Albedo 76–212		-	22–42
Mandarins, Punjab, India	80–206		-	13–30
Grapefruit (2 varieties), Florida	239, 147		47	36
Lemon standard variety	Flavedo and Albedo 128		-	37
Meyer Lemon	65		-	28

**Table 12 foods-08-00523-t012:** Recent reports on extraction of important value-added compounds from citrus wastes involving conventional and modern techniques.

Method/Technique	Matrix	Analyte/Extracted Molecule	Extraction Condition	Remarks	Ref.
Heat Treatment and Ultrasonication	Huyou peels (Changshanhuyou) *Citrus paradisi*	Phenolics (*p*-hydroxybenzoic acid, vanillic acid); Cinnamic acid (caffeic acid, *p*-coumaric acid, farulic acid, sinapic acid); chlorogenic acid (phenolic esters); Flavanone (Narirutin, Naringin, Hesperidin, Neohesperidin)	80% Methanol; Heating at 120–150 °C for 30–50 min.	Decline in total phenolic content on heating at high temperature and longer duration. Increase in the content of free phenolic acid fraction	[57]
Enzyme assisted extraction	Peel waste Lemon (*Citrus limon* cv. Meyer); Lemon (*Citrus limon* cv. Yenben); Grapefruit (*Citrusparadisi*); Mandarin (*Citrus reticulata* cv. Ellendale); Sweet orange (*Citrus sinensis* cv. Navel)	Total phenolic content	Liquid nitrogen from pulverization; Enzymes: Cellulase ^R^ MX, Cellulase ^R^ CL, Kleeras ^R^ AFP Enzymatic extraction at 50 °C	Total phenolic content (in mg Gallic Acid Equivalent/100 g peel) Grapefruit (90–162 mg) > Yenben Lemon > Mandarin > Orange Highest recovery obtained from Celluzyme MX at 1.5% *w*/*w* Recovery by enzyme −65.5% Recovery by solvent−87.9%	[64]
Steam Explosion and Simultaneous Saccharification and Fermentation (SSF)	Citrus peel waste, seeds and membrane	*d*-limonene and ethanol	*Saccharomyces cerevisiae* used for fermentation; Steam Explosion at 150–160 °C for 2.4 min extraction of *d*-limonene Enzyme SSF-Pectinase (PectinexUltraSP); Cellulase (Celluclast 1.5 L) Beta glucosidase (Novozyme 188) *S. cerevisiae* at 37 °C and pH 6.0	Steam explosion prior SSF removes 90% of *d*-limonene; Fermentation with initial pH of 6.0 produced greater amounts of ethanol; Ethanol production declined after 24h at *d*-limonene concentration of 0.33% (*v/v*) at initial stage and 0.14% (*v/v*) at final stage because of the antibacterial effect of the same. Steam explosion enables rupture of the cell wall and access of enzymes to all cellular components and removes the antimicrobial *d*-limonene.	[180]
	Peel waste Lemon (*Citrus limon* L.), Mandarin (*Citrus reticulata* L.)	*d*-limonene, galacturonic acid and bioethanol	*Saccharomyces cerevisiae* (CECT 1329) used for fermentation; Steam Explosion at 160 °C for 5 min; Enzymatic hydrolysis for 24 h at 45 °C with shaking at 150 rpm; Enzyme deactivation at 105 °C, 15 min. Alcoholic fermentation at 37 °C: (a) Hydrolysis and fermentation (HF); (b) SSF	Ethanol production 50–60 L/1000 kg fresh citrus peel waste; Significant antimicrobial effect on *S. cerevisiae* due to lemon essential oils at concentrations above 0.025%.	[181,182]
Ultrasound Assisted Extraction (UAE)	Citrus peel: Tahiti lime (*Citrus latifolia*); Sweet orange (*Citrus sinensis*); Oneco tangerine (*Citrus reticulata*)	Flavonoids: Hesperidin, Neohesperidin Nobiletin, Tangeretin	Peel: Deionized Water (1:10); flavonoids separated in water due to high solubility after UAE at 60 kHz for 30–90 min	Yield-Flavonoid fraction (in mg per gram peel) 40.25 ± 12.09 mg; Total phenolic content in flavonoid fractions (mg GAE per gram peel). Lime-74.80 ± 1.90 Orange-66.36 ± 0.75 Tangerine-58.68 ± 4.01	[183]
	Orange (*Citrus sinensis* L.)	Polyphenols (flavone glycosides)	80% Ethanol (Ethanol:DI Water/ 4:1, *v/v*) 40 °C, 60 min Sonication power 150 W	Extraction Yield: 10.9% per 100 g peel waste Total phenolic extraction 275.8 mg GAE/100 gm peel waste Flavonone concentration (mg/100 g peel waste): Naringine: 70.3 mg (by UAE) 50.9 mg (by solvent extraction, SE) Hesperidin-205.2 mg (by UAE) 144.7 mg (SE) The quantities are greater than SE No evidence for flavanone degradation 30–40% increase in extraction of total phenolic content in 60 min than SE; Energy saving and green method.	[56]
	Citrus peel (tangerine peels)	Carotenoids (*β*-carotenoids)	Ostrich oil; Sunflower oil Ultrasonic intensity 19 W/cm^2^ for 30 min	*β*-carotene extracted Sunflower oil- 75.741 mg/L Ostrich oil-88.110 mg/L	[184]
	Penggan peel (*Citrus reticulata*)	Hesperidin	Methanol 60 kHz, 30 W, 60 min, 40 °C	Yield: 50-55 mg/g peel Edge over Soxhlet extraction which works at 60 °C. UAE at 40 °C delivers better yield	[185]
	Citrus rind Nules Celmentine mandarin (*Citrus reticulata* Blanco)	Phenolic acids and flavanones	Ethanol: H_2_O::80: 20; (*v/v)*; Methanol:H_2_O:HCl: 70:29.5:0.5 (*v/v/v)*; DMSO:Methanol (50:50; *v/v)* at 35 °C, 10–30 min	Seven Phenolic acids and three flavanone glycosides extracted: Hydroxybenzoic acids (*p*-hydroxybenzoic acid, vanillic acid); hydroxycinnamic acids (chlorogenic acid, Caffeic acid, *p*-coumaric acid, ferulic acid, sinapic acid); **Flavanone glycosides**: (Nerirutin, hesperidin, didymin) Highest yield obtained was Hesperidin ~7500–32,000 µg/g dry wt.	[58]
Maceration- UAE	Kinnow (*Citrus reticulata* L.)	Polyphenols and flavanoids (ferulic acid and hesperidin)	80% Ethanol; 80% Methanol; 80% Ethyl acetate; Acetone Maceration: Sample: Solvent ratio 1:10; 1:15; 1:20 At 30 °C, 40 °C for 20 h; UAE- Sample: Solvent 1:10; 1:15; 1:20 at 35 °C, 45 °C, 55 °C for 40–70 min; Ultrasound Frequency fixed at 35 kHz	Eleven phenolic compounds, five phenolic acids and six flavonoids extracted: Maximum polyphenol yield with 80% Methanol using UAE = 32.48 mg GAE/g dry wt. Min. yield 80% with Ethyl acetate using maceration method- 8.64 mg GAE/g dry wt. Methanolic extract showed highest antioxidant activity for FRAP, highest scavenging activity for DPPH and superoxide anion radical.	[186]
(i) Hydrodistillation (HD) (ii) Cold Pressing Microwave (iii) Hydrodiffusion and Gravity (MHG)	Eureka lemon (*Citrus limon* L.), Villa Franca (*Citrus limon* L.), Lime (*Citrus aurentifolia* Chrism. Swing), Marsh Seedless (*Citrus paradisi* L.), Tarocco (*Citrus sinensis* L.), Valencia late (*Citrus sinensis* L.) Washington Navel (*Citrus sinensis* L.), Tangelo seminole (*Citrus paradisi* Macf.)	Essential oil	HD- 500 g peel in 3 L deionized water for 3 h; 3 kWh CP: 1 kg, Mechanical press MHG: 500 g; 500 W; 15 min; 0.2 kWh	MHG is simplified working mechanism, faster technique and requires lesser energy; Yields high-purity final products; Post treatment of the wastewater.	[44]
Supercritical water extraction	Citrus Pomace peels (*Citrus unshiu*)	Polymethoxy-lated flavones: Sinensetin, noniletin, tangeretin	Methanol and water at 200 °C, 1.4 MPa, 60 min	Max. phenolic content—2974.7 µM.	[187]
Conventional solvent extraction	Mandarin peels (*Citrus reticulata*)	Narirutin, Hesperidin	70% aqueous Acetone with 1 g dry peel in 50 mL solvent, 2 h at 40 °C	Narirutin: 15.3 mg/g extract; 5.93 mg/g peel Hesperidin: 80.9 mg/g extract; 31.42 mg/g peel	[188]
(i) Supercritical Fluid Extraction (SFE) (ii) Maceration (iii) Reflux (iv) Soxhlet Extraction	*Citrus paradisi* L.	Naringin	(i) Supercritical CO_2_, 15% Ethanol; 95 bar; 58.6 °C; (ii, iii) Ethanol:Water/70:30 where 20 g peel dissolved in 100 mL solvent; 3–30 h, 22–25 °C Evaporation at reduced pressure at 40 °C (iv) Ethanol 8 h, 60 °C	Naringin yield/ time of extraction (i) SFE: 14.4 ± 0.2/45 min (ii) Maceration: 11.1 ± 0.6/24 h (ii) Reflux: 13.5 ± 0.5/3 h (iii) Soxhlet: 15.2 ± 0.5/8 h	[189]
Step 1: (i) Hydrodistillation (HD) (ii) Instant Controlled Pressure Drop Technique (DIC) Step 2: (iii) UAE (iv) SE	Sweet orange peels (*Citrus sinensis*)	Step 1: Essential oils by HD and DIC; Step 2: Hesperidin and naringin by SE and UAE methods from the solid residue left after extraction of Essential oils by HD and DIC	(i) HD: 200g dried peel in 2 liters dist. H_2_O; 4 h; (ii) DIC: Saturated steam; 5 kPa to 1 MPa; 2 min; (iii) UAE: Ethanol: Water/4:1; 40 °C, 60 min, 25 kHz, 150 W; (iv) SE: Ethanol: Water/4:1; 40 °C, 60 min	Step 1: Essential oils from fresh dried peels (i) 1.97 mg/g dry matter; (ii) 16.57 mg/g dry matter Step 2: Hesperidin and naringin from solid residue after extraction of essential oils (iii) Hesperidin: 0.825 ± 1.6 × 10^−2^ g/g dry matter Naringin: 6.45 × 10^−2^ ± 2.3 × 10^−4^ g/g dry matter (iv) Hesperidin-0.64 ± 2.7 × 10^−2^ g/g dry matter Naringin-5.7×10^−2^ ± 1.6 × 10^−3^ g/g dry matter	[190]
Pulsed electric field (PEF) assisted pressing extraction	Sweet orange peels (*Citrus sinensis*)	Polyphenols, Flavonoids (naringin and hesperidin)	Distilled water during pressing step Step 1: PEF Square waveform pulses of a width of 3 µs with a frequency up to 300 Hz; Max. output voltage and current 30 kV, 200 A; Pulse frequency 1Hz; 20 pulses of 3 µs each was given for 60 µs followed by Pressing Step 2: Pressing for 30 min at 5 bars; 1–6 times	Total polyphenol extraction yield increased by 20%, 129%, 153%, 159% at electric field strength of 1, 3, 5, and 7 kV/cm of PEF, respectively, and the antioxidant property increased by 51%, 94%, 148% and 192%, respectively. PEF treated peels yield higher total polyphenolic content. Highest total polyphenolic content is 34.80 mg GAE/100 g of fresh peel.	[191]
Solvent Extraction	Lemon (*Citrus limon* cv. Meyer), Lemon (*Citrus limon* cv. Yenben), Grapefruit (*Citrus* × *paradisi*, unknown cultivar), mandarin (*Citrus reticulata* cv. Ellendale), Sweet orange (*Citrus sinensis* cv. Navel) peels	Phenolics	Ethanol, Methanol 3 h; 20-80 °C	Total yield in mg GAE/g fresh peel wt. Grapefruit (162 mg) > Mandarin (121 mg) > YenBen lemon (118 mg) > Orange (74 mg) > Meyer lemon (60 mg)	[50]
	Grapefruit *(Citrus paradisi* L., *Citrus aurantium*), Pumello (*Citrus grandis*)	Nootkatone (Sesquiterpene), Flavonoids (Narirutin, Naringin, Neohesperidin)	Sequiterpenes: n-pentane (1 g peel/4 mL), 200 µg lauric acid methyl ester; Samples homogenized 3 times in *n*-pentane; decanted and dried with anhydrous Na_2_SO_4_ followed by N_2_ at RT; Flavonoids: 60 mg dried peel/10 mL DMSO Extracts filtered through 0.45 µm nylon mesh	Grapefruit (mg/100 g fresh peel wt.) Narirutin: 1188 ± 220 mg (Immature fruit) 231 ± 63 (Mature fruit) Naringin: 12102 ± 2310 (Immature fruit) 2195 ± 339 (Mature fruit) Neohesperidin: 274 ± 35 (Immature fruit) 17 ± 9 (Mature fruit) Sesquiterpene: 5 ± 0.5 (Immature fruit) Not detected in mature fruits Pumello Narirutin: 12 ± 2 mg (Immature fruit) 10 ± 7 (Mature fruit) Naringin: 14775 ± 1892 (Immature fruit) 569 ± 65 (Mature fruit) Neohesperidin: 14 ± 3 (Immature fruit) 17 ± 3 (Mature fruit) Sesquiterpene: 0.9 ± 0.4 (Immature fruit) Not detected in mature fruits	[192]
	Orange peels (Baladi orange, Novel orange)	Phenols (tannic acid) and Flavanoids (rutin)	95% Ethanol 1:2 (*w/v)* (Peel/EtOH); Filtered and re-extracted 2 times; lipid removal by hexane	Total phenol (mg tannic acid/100 g fresh peel) Baladi orange: 591.69 mg Novel Orange: 591.77 mg Total flavonoid (mg rutin/100 mg fresh peel) Baladi orange: 80.93 mg Novel orange: 83.49 mg	[193]
(i) Cold pressing (ii) Hydrodistillation (HD) (iii) Microwave dry distillation (MD)	*Citrus limon* L.	Essential oils	DI Water for hydrodistillation (i) CP: 1 kg of whole lemon utilized for cold pressing for 1 h followed by centrifugation, dried over anhydrous sodium sulphate (ii) HD: 200 g peel in 2 L H_2_O distilled for 3 h (iii) MD: 200 g peel, microwave irradiation power 200 W for 30 min	(i) CP: Electricity consumed: 1 kWh Yield: 0.05% CO_2_ rejected: 800 g (ii) HD: Electricity consumed: 4.33 kWh Yield: 0.21% CO_2_ rejected: 3464 g(iii) Electricity consumed: 0.25 kWh Yield: 0.24% CO_2_ rejected: 200 g Microwave dry distillation is energy and time saving, requires no solvent and the extracted essential oils has higher amounts of oxygenated compounds	[128]
Water-based extraction (i) Soxhlet Extraction (SOE) (ii) Microwave method (iii) Hand Press Extraction (iv) Combined Microwave and Hand Press Extraction (MW-HP)	Navel Orange	Pectin	DI Water (i) SOE: Extraction in boiling water, 6 h, oven drying of the products at 100 °C, 48 h (ii) MW: 150 °C, 15 min, pH = 2.0 (iii) HP: Hand pressed contents were leached into water (iv) MW-HP: peels were microwaved prior HP	(i) SOE: Soxhlet method yielded twice as much pectin than microwave although the reproducibility factor of microwave method is 120 (ii) MW: A ratio of sample to solvent of 1:12.5 extracted highest amounts of pectin from albedo (iii) MW-HP: Combined microwave and hand pressing extraction yielded 12% more pectin from flavedo than hand pressing alone The aroma of the final yield by microwave is dark and bitter whereas that obtained from Soxhlet is light and sweeter	[194]
(i) Microwave Steam Distillation (MSD) (ii) Steam Distillation (SD)	Orange (*Citrus sinensis*), Lemon (*Citrus limon*), Mandarin (*Citrus reticulata*)	Essential oils; limonene	DI Water (i) Microwave power 135 W, 35 min(ii) Distillation for 45 min	Limonene extraction yield: SD/MSD Orange: 83.22%/80.97% Mandarin: 83.03%/84.39% Lemon: 65.29%/59.16%	[195]
	Orange peel (*Citrus aurentium* L.)		DI water (i) Microwave oven working at 800 W, 2.45 GHz, 140 min (ii) SD: Steam Distillation for 7 h	(i) MSD: Extracted essential oil contain 18 detectable components (ii) SD: Extracted essential oil contain 7 detectable components	[196]
(i) Hydrodistillation (HD) (ii) Gel Permeation Chromatography (GPC)	20-fold concentrates of pressed peel oils of Tangerine, Mid-season orange, Valencia orange	Pigments	(i) HD with deionized water, separation of tiny particles with Hexane; in rotary evaporator at 40 °C, 1 mm Hg. (ii) GPC using Tetrahydrofuran	Pigment extract (mg/kg cold press oil) Tangerine: 116 mg Mid-season orange: 95 mg Valencia orange: 116 mg Valencia peel extract: 350 mg	[123]

**Table 13 foods-08-00523-t013:** Separation, isolation, and purification of citrus phytochemicals from the crude extract obtained from different extraction procedures.

Citrus Species	Fruit Part/Extraction	Separation/Isolation Method/Determination	Column/Mobile Phase	Isolated Phytochemicals	Ref.
*C. sinensis* L. (Florida, USA)	Peel extract	1. Flash Chromatography-(UV spectrum = 254 nm) 2. Reverse-phase HPLC 3. Prep-HPLC (UV spectrum = 326 nm)	1. 330 g prepacked silica gel (particle size 35–60 µm) flash column/ethyl acetate (10–40%) and hexanes (90–60%); isopropanol (15%) and hexanes (85%) 2. Semi prep-HPLC: YMC HPLC column (75 × 30 mm i.d., 5–10 µm particle size) 3. Regis Whelk-O 1–450 g column/ 35% absolute ethanol and 65% hexanes	Nobiletin and 5,6,7,4′-tetramethoxyflavone	[215]
*C. sinensis* Florida, USA	Peel extract (cold-pressed oil)	1. Normal Phase Chromatography 2. Flash Chromatography (UV spectrum = 254 nm) 3. Reversed-phase analytical HPLC 4. Prep-HPLC 5. Semi-preparative reversed column HPLC 6. Supercritical Fluid Chromatography (UV spectrum = 220 nm) 7. Liquid chromatography (LC)–electron spray ionization mass spectrometry (ESI-MS)	1. Silica gel (60 Å, 32–63 µm) columns (330 g) 2. Silica gel (particle size 35–60 m) flash column/ethyl acetate and hexanes; 3. Octadecyl (C_18_) derivatized silica gel (4.6 mm × 50 mm, 5 µm) 4. Preparative HPLC (30 mm × 75 mm, 10µm, ODS-A) columns/35% acetonitrile and 65% water 5. C_18_ reverse-phase column, Xterra OBDTM (19 mm × 100 mm) 6. SFC chiral column (30 mm × 250 mm, 5 µm)/CO_2_ 7. Chromegabond WR C_18_, 3 µm, 120 Å; 30 mm × 3.2 mm/acetonitrile and H_2_O with 0.05% TFA, typical gradient of 10–90% acetonitrile	Nobiletin, tangeretin, 3,5,6,7,8,3’,4’-heptamethoxyflavone and 5,6,7,4’-tetramethoxyflavone	[222]
*C. sinensis*	Peel oil	1. Open column chromatography 2. Reversed-phase purification chromatography 3. analytical silica gel TLC 4. Analytical HPLC–MS	1. Silica gel prepacked-170 g column/hexane/ethyl acetate 1:3 to 2:1 (*v/v*) 2. RediSep reversed-phase C_18_ (43 g) column/water/methanol (80/20, *v/v*-60/40 (*v/v*)	3′,4′,3,5,6,7,8-heptamethoxyflavone	[229]
*C. sinensis*	Peel oil (Solvent extraction/Petroleum ether)	1. Isolation 2. Crystallization 3. Hydrolysis-Reflux	1. Carbon tetrachloride 2. Crystallization-Methanol 3. Hydrolysis-Reflux: Ethanol and Aqueous potassium hydroxide	Nobiletin	[230]
*C. sinensis* (Florida, USA)	Peel Molasses	1. Filtration and Ultrafiltration 2.Fractionation- 3. Size exclusion chromatography (SEC) 4. Analytical HPLC (PDA = 230 and 600 nm): LCMS-ESI-MS	1. Grade 161 glass fiber filter and Romicon model HF4 hollow fiber cartridge ultrafiltration system 2. Sepabeads SP70 column (1.5 cm × 29 cm/H_2_O, 50% acetone Bio-Gel P2 column (5 cm × 82 cm)/15% EthanolDE52 cellulose anion exchange column (2.5 cm × 25 cm) 3. P2 column (5 cm × 80 cm)/ 15% Ethanol 4. Alltech Alltima C_8_ 5 μm analytical column (4.6 mm × 100 mm)/10 mM phosphoric acid/acetonitrile (90:10, *v/v*): (Flavonoid glycosides were analyzed 2% formic acid/water/acetonitrile	3,5,7,10,14,16-hydroxycinnamates, **Polymethoxylated flavone**: Nobiletin, heptamethoxyflavone, and sinensetin, **Flavonoid glycosides**: Hesperidin and narirutin, narirutin 4′-glucoside, hesperetin trisaccharide, 6,8-di-*C*-glucosylapigenin, **Phenolic glycoside**: Coniferin, phlorin **Phenolic acids**: Ferulic acid, *p*-coumaric acid	[231]
*C. sinensis* Osbeck (Local fruit Oslo, Norway)	Peel extract	1. Open Column Chromatography; 2. Flash Chromatography; 3. Thin Layer Chromatography- UV irradiation (254 and 366 nm)	1. MPLC column (49 × 900 mm) filled with silica gel (40–63 µm)/Chloroform, ethyl acetate, acetone, methanol 2. Silica gel, 40–63 µm/hexane-ethyl acetate (9:1 or 1:1) 3. TLC (Si gel 60 F254, 0.2 mm thickness/g hexane-ethyl acetate (1:1); toluene-dioxane-acetic acid (90:25:4)	5-hydroxy6,7,3′,4′-tetramethoxyflavone, sinensetin (5,6,7,3′,4′-pentamethoxyflavone, 3), tetramethylscutellarein (5,6,7,4′-tetramethoxyflavone; 3,5,7,8,3′,4′-hexamethoxyflavone, tangeretin (5,6,7,8,4′-pentamethoxyflavone, 3,5,6,7,3′,4′-hexamethoxyflavone, nobiletin (5,6,7,8,3′,4′-hexamethoxyflavone, 3,5,6,7,8,3′,4′-heptamethoxyflavone; hesperidin (5,3′-dihydroxy-4′-methoxy-7-rutinosyloxyflavanone ferulic acid	[232]
*C. kinokuni* Hort. ex Tanaka (Shizouka, Japan)	Peel extract	1. Open Column Chromatography; 2. Preparatory-Thin Layer Chromatography	1.Silica gel/toluene, CH_2_Cl_2_, AcOEt, acetone, and MeOH 2. Si-gel TLC/acetone–CHCl_3_ (1:9, 1:19 or 1:29), acetone–benzene (2:8), acetone–hexane (3:7), AcOEt–benzene (1:1), AcOEt–hexane (1:1)	Scoparone, scopoletin, nobiletin, sinensetin, tangeretin, 5-hydroxy-6,7,8,49-tetramethoxyflavone, 5-hydroxy-6,7,39,49-tetramethoxyflavone, 5-demethylnobiletin, 6-demethoxynobiletin, 6-demethoxytangeretin, 5,6,7,49-tetramethoxyflavone, 3,5,6,7,39,49-hexamethoxyflavone, 39-hydroxy-5,6,7,8,49-pentamethoxyflavone, 3,5,6,7,8,39,49-heptamethoxyflavone, 7-hydroxy5,6,39,49-tetramethoxyflavone, 7-hydroxy-5,6,8,39,49-pentamethoxyflavone, 5,7,8,39,49-pentamethoxyflavanone, 5-O-demethylcitromitin, 3,4,39,49,59,69-hexamethoxy-29-hydroxychalcone, 29-hydroxy-4,49,59,69-tetramethoxychalcone, b-sitosterol, (2S)-5,6,7,8,49-pentamethoxyflavanone, (2S)-5,6,7,39,49-pentamethoxyflavanone, 29-hydroxy3,4,39,49,69-pentamethoxychalcone,	[233]
*C. reticulata* Blanco; (Tangerine, China)	Peel extract	1. High speed countercurrent chromatography; 2. HPLC-PDA = 270 nm)	1. Multilayer coil planet centrifuge- equipped with a polytetrafluoro-ethylene multilayer coil of 110 m × 1.6 mm, I.D./Two phase solvent system-*n*-hexane, ethyl acetate, methanol and water 2. Shim-pack VP-ODS column (250 mm × 4.6 mm, I.D.)/acetonitrile and water (50:50, *v*/*v*)	Nobiletin, 3,5,6,7,8,3,4-heptamethoxyflavone, tangeretin and 5-hydroxy-6,7,8,3,4-pentamethoxyflavone	[218]
*C. reticulata* Blanco cv. Ponkan	Peel extract	1. High speed countercurrent chromatography(Elite UV spectrum-200–330 nm) 2. Prep-HPLC 3. Analytical HPLC- PDA detector 4. Electrospray ionization mass spectrometry (ESI-MS)	1. 200 mL column with six-layer coils made of 5.0 mm i.d. polytetrafluoroethylene (PTFE) tubing: Separation system composed of a K-1800 Wellchrom pump: a 150 mL sample loop made of 3 mm i.d. PTFE tubing, the high-speed countercurrent chromatograph and a B-684 collector/*n*-hexane–ethyl acetate–methanol–water (1:1:1:1:1.5, *v*/*v*) 2. DS-BP-30 column (250 × 30 mm I.D./methanol–water (60:40, *v*/*v*) 3. C_18_ column (250 × 4.6 mm i.d., 5lm)/methanol and water	Isosinensetin, sinensetin, nobiletin and tetramethyl-*o*-scutellarein	[234]
*C. sunki* Hort. ex Tanaka (Jeju, S. Korea)	Peel extract (hot-water extraction)	1. Fractionation 2. HPLC (Flavonoids) (UV spectrum = 200–400 nm) 2. Semi-prep HPLC (Polymethoxy Flavone) UV-spectrum = 200–400 nm	1. *n*-hexane, chloroform, ethyl acetate, n-butanol: dissolved in EtOAc:methanol (MeOH) (1:1, *v*/*v*): filtration through 0.50-µm polytetrafluoroethylene (PTFE) 2. Sunfire™ C_18_ column (250 × 4.6 mm ID; 5 µm)/(A): acetonitrile (MeCN) containing 0.5% acetic acid (B) water containing 0.5% acetic acid 3. Symmetryprep™ C_18_ column (300 × 7.8 mm ID; 7 µm)/Methanol, Water	Isosinensetin, sinensetin, tetra-*O*-methylisosutellarein, 5,7,4′-trimethoxyflavone, nobiletin, tangeretin, 5-demethylnobiletin, 5-demethyltangeretin	[235]
*C. paradisi* (White grapefruit, Florida)	Peel oil (cold-pressed oils)	1. High speed countercurrent chromatography (UV spectrum = 310 nm) 2. HPLC with Diode Array Detection (HPLC-DAD)3. Gas Chromatography-Mass Spectrometry	1. Three preparative coils, connected in series (diameter of tubing 2.6 mm, total volume 850 mL)/Hexane–Ethyl acetate; Methanol–Water; hexane/Ethanol–Water 2. C_18_-Spherisorb ODS2 column (250 × 4 mm, particle size 5 µm) (A) acetonitrile, (B) water 3. DB-5 column (15 m × 0.25 mm i.d. fused silica capillary, 0.5 µm)/Helium gas	Meranzin hydrate, marmin, epoxybergamottin hydrate, auraptenol, Auraptenol meranzin, epoxyauraptene, and epoxybergamottin, nobiletin, tetra-omethylscutellarein (39), meranzin, isomeranzin, and heptamethoxyflavone, tangeritin, citropten, bergapten	[236]
*C. aurentifolia* Swingle (Mexico)	Peel oil (Cold-pressed oils)	1. High speed countercurrent chromatography (UV spectrum = 310 nm) 2. HPLC with Diode Array Detection (HPLC-DAD) 3. Gas Chromatography-Mass Spectrometry	1. Three preparative coils, connected in series (diameter of tubing 2.6 mm, total volume 850 mL)/Hexane-Ethyl acetate; Methanol-Water; hexane/Ethanol-Water 2. C_18_-Spherisorb ODS2 column (250 × 4 mm, particle size 5 µm) (A) acetonitrile, (B) water 3. DB-5 column (15 m × 0.25 mm i.d. fused silica capillary, 0.5 µm)/Helium gas	Herniarin, isopimpinellin, citropten, bergapten oxypeucedanin; bergamottin and, 5-geranyloxy-7-methoxycoumarin	[236]
*C. jambhiri* Lush (Lemon, Nagaland, India)	Peel extract (petroleum ether) Soxhlet extraction	Open Column Chromatography	Active alumina column/Benzene Washing with petroleum ether and crystallization with Methanol	Hesperidin, tangeretin, neohesperidin. 5-*O*-desmethyltangeretin	[237]
*C. hassaku* HORT. ex. TANAKA (Japan)	Peel extract (Ethanol reflux)	1. Open Column Chromatography; 2. Preparatory-Thin Layer Chromatography; 3. Preparatory-HPLC; UV-spectrum = 313 nm	1. Silica gel C-200 (Wako) 2. Wako-gel B-O (Wako) 3. ODS-120T column (7.8 mm i.d. × 30 cm)/Methanol–Water	4′,5,6,7,8-pentamethoxyflavone, 3’,4′,5,6,7,8-hexamethoxyflavone, 3,3’,4′,5,6,7,8-heptamethoxyflavone, 3′, 4’,5,6,7-pentamethoxyflavone, 4′,5,7-trimethoxyflavone, 3,3’,4′,5,7,8-hexamethoxyflavone, 4’,5,7,7.8-tetramethoxyflavone, 3′,4’,5,7,8-pentamethoxyflavone	[205]
*Citrus Sinensis.L* (sweet orange); India	Peels Solvent extraction (EtOH; Methanol–Dichloromethane–Water (MDW) (0.3:4:1, *v*/*v*); MeOH-H_2_O)	1. Reversed-phase HPLC 2. Thin layer chromatography 3. HPLC–ELSD; ESI-MS (Evaporative Light Scattering Detector (ELSD) coupled to electrospray ionization mass spectrometric (ESI-MS)	1. Adsorbosphere column–NH_2_, (250 × 4.6 mm column)/Mobile phase–acetonitrile–water 2. Cellulose MN 300 G/Mobile phase- *N*–butanol–acetone–pyridine–water (10:10:5:5, *v/v/*) 3. Atlantis dc-18 column (50 x 4.6mm—5μm)/ Mobile phase (A) 0.10% formic acid in HPLC grade deionized water; (B) Methanol	Fructose, galactose, glucose, arabinose and xylose	[238]
Sour and sweet oranges, umbilical orange, novel orange, lime, lemon, pink and white grapefruit, aeglemarmelos, bergamot, sour and sweet tangerines and clementines (Iran)	Extraction of juice; Centrifugation	Reversed-phase chromatography UPLC–MS (λ = 254 nm)	(150 mm × 4.6 mm i.d., 5 μm particle ZORBAX Eclipse XDB-C_18_)/Mobile phase—2% acetonitrile and 50 mM phosphate solution (dissolve 6.8 g potassium dihydrogen phosphate in 900 mL water (pH = 2.8)	Ascorbic acid	[239]

**Table 14 foods-08-00523-t014:** Identification, determination, and analysis of citrus phytochemicals by chromatography assisted with spectroscopic techniques.

Citrus Fruit Part	Extraction/Solvent	Separation and Isolation/Column/Detection	Mobile Phase Technique	Phytochemicals Detected	Ref.
Peel and Pulp (*limón de Pica* (Pica Lemon, *C. aurantifolia* (Christm) Swingle var. Pica), *limón Sutil* (*C. aurantifolia* (Christm) Swingle) var. sutil; Chile)	Maceration; filtration and evaporation/MWC	Purospher star—C_18_ (250 × 5 mm) HPLC-UV and ToF–ESI–MS/MS	(A) 10% HCOOH in H_2_O, (B) CH_3_CN	Citric acid: Ppe, Spe, Spu **Phenolics**: Quercetin: Ppu, Spe Diosmetin: Ppu Naringin: Ppe, Ppu, Spe Hesperidin: Ppe, Ppu, Spe, Spu Eriodictyol: Ppu, Spe Apigenin: Spe, Ppe Luteolin: Ppe, Ppu, Spe, Spu Isoquercetin: Spe, Ppu Lucenin: Ppe, Ppu, Spe, Spe Rutin: Ppu, Spe Neodiosmin: Ppe Diosmin: Spe, Spu Eriocitrin: Ppe, Ppu, Spe (Spe: Sutil lemon peel Spu: Sutil Lemon pulp Ppe-Pica lemon peel Ppu: Pica lemon pulp)	[250]
Peel (*C. aurentifolia;* Italy)	Maceration/MWH	(a) Phenomenex Luna C_18_, 250 × 4.60 mm (b) HP-5 MS capillary column (30 m length, 0.25 mm i.d., 0.25 μm film thickness) (a) HPLC-UV-Vis, 280 nm (b) GC–MS	HPLC (A) 0.1% HCOOH in H_2_O, (B) MeOH GC–MS Carrier gas-He, N_2_	HPLC **Phenolics** rutin, apigenin, quercetin, kaempferol, nobiletin, tangeretin and hesperidin GC–MS **Terpenes**-Limonene, linalool and linalyl acetate, *β*-pinene **Fatty acids**: Palmitic acid, methyl palmitate	[251]
Seeds (Bitter orange, *C. aurantium*; Tunisia)	Grinding, maceration, filtration and evaporation/MeOH	Hypersil ODS C_18_ 250 × 4.6 mm RP–HPLC– UV-Vis, 280 nm	(A) CH_3_CN, (B) 0.2% H_2_SO_4_ in H_2_O	**Flavonoids**: Epigallocatechin, Naringin, Hesperidin, Neohesperidin, Resorcinol, Catechin, Rutin, Kaempherol **Phenolic acids**: Gallic acid, Syringic acid, Rosemarinic acid, *p*-Coumaric acid, *trans*-2-Hydroxicinnamic acid	[252]
Whole (Sour orange, *C. aurentium* Linn. cv Xiaohongc-heng; China)	Drying, UB30, centrifugation/MDS	Diamonsil C_18_ 250 × 4.6 mm FGs, PMFs and phenolic acids HPLC–PDA 210 to 400 nm Limonoids UV—210 nm Synaphrine UV-vis—225 nm	FGs, PMFs and phenolic acids (A) MeOH, (B) 4% AcOH (*v/v*) Limonoids MeOH/Acetonitrile/PBS (containing 0.03 M potassium dihydrogen phosphate, pH 3.5) 10:40:39 by Volume Synaphrine MeOH/H_2_O/SDS (70:30:0.1) by volume	**Flavonoids**: Narirutin, Naringin, Hesperidin, Nobiletin, Tangeretin **Phenolic acids**: Caffeic, *p*-Coumaric, Ferulic, Sinapic, Protocatechuic, *p*-Hydroxybenzoic, Vanillic **Limonoids**: Limonin, Nomilin, **Alkaloid**: Synephrine	[253]
Epicarpand whole fruit(*C. grandis* ‘Tomentosa’; China)	Drying, UB30, centrifugation/MeOH	Phenomenex Kinetex column (2.1 mm × 100 mm, 2.6 µmUFLC–DAD–Q-TOF-MS/MSDAD—190–400 nmFlavonoids: Band I (300–380 nm)Band II (240–290 nm){Flavones(310–350 nm) Flavonols (350–330 nm) Flavanones (300-330)} Coumarins: UV—270–320 nm	(A) MeOH, (B) 0.1% HCOOH in H_2_O (*v/v*)	**Flavonoids**: Eriocitrin, Neoeriocitrin, Narirutin, Naringin, Hesperidin, Isorhoifolin, Rhoifolin, Neodiosmin, Poncirin, Melitidin, Naringenin, Apigenin, Kaempferol **Phenolic acids**: Protocatechuic acid, Veratric acid, Caffeic acid, 3-coumaric acid, **Coumarins**: Bergaptol, Meranzin hydrate, Oxypeucedanin, Bergapten, 5-Hydroxyisomeranzin, Isomeranzin, 7-Hydroxycoumarin, Epoxyaurapten, Imperatorin, Osthol, Isoimperatorin, Epoxybergamottin, Bergamottin **Fatty acid**: Palmitic acid **Limonoids**: Nomilin, Limonin, Ichangin, Obacunone, Nomilinic acid, Isoobacunoic acid, Deacetylnomilinic acid **Sesquiterpene**: Nootkatone	[254]
Flavedo (Pummelo cultivars, *C. grandis* L. Osbeck; China	UB30, centrifugation, evaporation/MeOH	Zorbax SB C_18_ 250 × 4 mm HPLC–MS/MS	(A) H_2_O/AcOH (99:1, *v/v*), (B) CH_3_CN/AcOH (99:1, *v/v*)	**Flavonoids**: Neoeriocitrin, Naringin, Neohesperidin, Acetyl naringin, Meltidin, Rhoifolin, Diosmin, Vicenin-2	[255]
Peel and Pulp (*C. grandis* ‘Tomentosa’; China)	Drying, UB30, centrifugation/MeOH	Acquity UPLC BEH C_18_ 100 × 2.1 mm UPLC–PDA Flavanones: 283 nm Polymethoxylated flavones: 330 nm	(A) 0.2% AcOH in H_2_O, (B) MeOH	Eriocitrin, Narirutin, Naringin, Hesperidin, Neohesperidin, Naringenin, Nobiletin, Tangeretin	[256]
Whole fruit (Pumelo *C. grandis* (L.) Osbeck cv Foyou; China)	Drying, UB30, centrifugation/MDS	Diamonsil C_18_ 250 × 4.6 mm Flavonoids and phenolic acids HPLC–PDA 210 to 400 nm Limonoids UV-210 nm Synaphrine UV-vis-225 nm	Flavonoids and phenolic acids (A) MeOH,(B) 4% Acetic acid Limonoids MeOH/Acetonitrile/PBS (containing 0.03 M potassium dihydrogen phosphate, pH 3.5) 10:40:39 by Volume Synaphrine MeOH/H_2_O/SDS (70:30:0.1) by volume	**Flavonoids**: Naringin, Tangeretin, **Phenolic acids**: Caffeic, *p*-Coumaric, Ferulic, Sinapic, Protocatechuic, *p*-Hydroxybenzoic, Vanillic **Limonoids**: Limonin	[253]
Whole fruit (*C. grandis* ‘Tomentosa’; China)	Drying, UB30, centrifugation/MeOH	Agilent Zorbax SB-C_18_ (4.6 × 50 mm^2^, 1.8 μm) RRLC–ESI–QTOF/MS RRLC–DAD	RRLC-ESI-MS/MS Analysis (A) 0.1% HCOOH in H_2_O (B) CH_3_CN RRLC-DAD Analysis (A) 0.1% aqueous formic acid (B) Methanol	**Flavone C-glycosides**: Vicenin-2, apigenin-8-C-glucoside-*O*-arabinoside apigenin-8-C-glucoside-*O*-rhamnoside apigenin-8-C-glucoside **Flavone C-glycosides**: luteolin-7-*O*-rutinoside **Flavanone *O*-glycosides**: Eriocitrin, Narirutin, Naringin, Melitidin **Coumarins**: Meranzin hydrate, Isoimperatorin, Meranzin, Marmin, Bergaptan, Isomeranzin, Imperatorin **Flavanone aglycone**: Naringenin **Limonoids**: Limonin, Isoobacunoic acid, Nomilin, Obacunone	[257]
Peel and Pulp (*C. limon*–Lemon; *limón Genova* (*Citrus* x *limon* (L.) Burm) var. Genova; Chile)	Maceration, filtration, evaporation/MWC	Purospher star—C_18_ 250 × 5 mm HPLC-UV and ToF–ESI–MS/MS	(A) 10% HCOOH in H_2_O, (B) CH_3_CN	Citric acid **Flavonoids in both peel and pulp**: Hesperidin, Naringin, Eriocitrin, Diosmin, Isohorifolin **Peel**: Rutin, Eriodictyol, Apigenin, Luteolin	[250]
Whole fruit (*C. limon* (L.) Burm.f. cv Eureka; China)	Drying, UB30, centrifugation/MDS	Diamonsil C_18_ 250 × 4.6 HPLC Flavonoids and phenolic acids PDA—210 to 400 nm LimonoidsUV—210 nm Synaphrine UV-vis—225 nm	Flavonoids and phenolic acids (A) MeOH, (B) 4% AcOH (*v/v*) Limonoids MeOH/Acetonitrile/PBS (containing 0.03 M potassium dihydrogen phosphate, pH 3.5) 10:40:39 by Volume Synaphrine MeOH/H_2_O/SDS (70:30:0.1) by volume	**Flavonoids** Narirutin, Hesperidin, Nobiletin, **Phenolic acids**: Caffeic, *p*-Coumaric, Ferulic, Sinapic, Protocatechuic, *p*-Hydroxybenzoic, Vanillic **Limonoids**: Limonin, Nomilin	[253]
Peel (*C. reticulata*–Mandarin; Slovenia)	Conventional extraction, EtOH, Acetone	Chromsep SS C_18_ 250 × 4.6 mm, 5 μm UV-DAD, 282–330 nm	(A) MeOH, (B) 2% AcOH in H_2_O (*v*/*v*)	**Flavanone**: Hesperidin, Narirutin, Didymin **PMFs**: Tangeretin, Nobiletin	[258]
Peel Mandarin (*C. reticulata* Blanco; China)	Lyophilization, maceration, centrifugation/MeOH	Zorbax SB-C_18_, 250 × 4.6 mm UV-DAD, 283–367 nm	(A) 0.1% HCOOH in H_2_O, (B) MeOH	**Flavanone**: Eriocitrin, Taxifolin, Narirutin, Naringin, Hesperidin, Neohesperidin, Eridictyol, Didymin, Poncirin, Naringenin, **Flavone**: Rhoifolin, Quercitin, Luteolin, Diosmetin, Sinensetin, Nobiletin, Tangeretin, **Flavonol**: Kaempferol **Phenolic acid**: Protocatechuic acid, *p*-Hydroxybenzoic acid, Vanillic acid, Caffeic acid, *p*-Coumaric acid, Ferulic acid, Sinapic acid, Chlorogenic acid	[259]
Peel (*C. reticulata* ‘Chachi’; Guang-chenpi; China)	Pressurized liquid extraction (PLE-Dionex ASE 300™)/MeOH, EtOH UAE (Aq. EtOH, 30 min, 40 °C); SE (MeOH, 80 °C) Heat-reflux extraction (HRE-MeOH-60 min 80 °C)	ZORBAX SB-Aq column (4.6 × 150 mm,5 μm)/ESI+, Capillary (350°C, 4kV) LC-DAD–ESI/MS	(A) Water containing 0.5% formic acid, (B) Methanol	**Flavanone**: Hesperidin **PMFs**: Tangeretin, Nobiletin	[201]
Pulp (*C. reticulata;* Mauritius)	Freeze drying, vortexing-maceration, centrifugation/MeOH	Waters Spherisorb ODS-2 150 × 4.6 mm UV-DAD, 280–330 nm Flavanone glycosides 280 nm Flavone and Flavonol glycosides 330 nm	(A) H_2_O-CH_3_CN (90:10, *v/v*), (B) CH_3_CN	**Flavanone glycosides**: Poncirin, Dydimin, Hesperidin, Neocitrin, Narirutin **Flavone and Flavonol glycosides**: Rhoifolin, Rutin, Isorhoifolin	[260]
Pulp (mandarin (*C. reticulata* Blanco.; China)	Drying, UB30, centrifugation/MeOH, DMSO	Flavonoids: Zorbax SB—C_18_, 250 × 4.6 mm Phenolic acid: Diamonsil C_18_, 250 mm × 4.6 mm UV-DAD Flavonoids: 283–367 nm (Flavanones: 283 nm; Flavones: 330 nm; Flavonols: 367 nm) Phenolic acid: 260–360 nm	Flavonoids: (A) 0.1% HCOOH in H_2_O (B) MeOH Phenolic acid: (A) 4% acetic acid, (B) methanol (20:80, *v/v*)	**Flavonone**: Eriocitrin, Narirutin, Taxifolin, Naringin, Hesperidin, Neohesperidin, Didymin, Poncirin, **Flavone**: Sinensetin, Nobiletin Rhoifolin, Quercitrin **Phenolic acids**: Protocatechuic acid, p-Hydroxybenzoic acid, Vanillic acid, Caffeic acid, Ferulic, *p*-Coumaric acid, Sinapic acid, Chlorogenic acid	[261]
Seeds (Mandarin (*C. reticulata Blanco*; Tunisia)	Grinding, maceration, filtration and evaporation/MeOH	Hypersil ODS C_18_, 250 × 4.6 mm RP–HPLC UV-Vis, 280 nm	(A) CH_3_CN, (B) 0.2% H_2_SO_4_ in H_2_O	**FGs and PMFs** Epigallocatechin, Catechin, Naringin, Hesperidin, Quercetin, **Phenolic acids**: Gallic acid, Caffeic acid, Chologenic acid, Ferulic acid, *trans*-2-Hydroxicinnamic acid	[252]
Pulp (with segment membrane, juice sacs) (Sweet orange (*C. sinensis*; China)	Freeze drying, maceration, centrifugation/EAA	Agilent Eclipse XDB—C_18_ 150 mm × 2.1 mm, 5 µm HPLC–ESI–MS–DART DART (*m*/*z*-50-800)	LC elution for Positive mode: (A) 0.1% HCOOH (B) MeOHFor Negative mode (A) 1mM NH_4_F in H_2_O, (B) MeOH	**Flavanones**: Hesperidin, neohesperidin, naringin, liquiritigenine, poncirin 4′-hydroxy-3,6-dimethoxy-6′,6″-dimethylpyrano, vicenin-2, gardenin, sinensetin, jaceosidin, nobiletin, **Flavonol**: quercetin 3, 5-*o*-di hexoside **Alkaloids**: Bchaconine, Isonaamine A, Phosphatidylcholine, Subaphyllin, Stachydrine, Calystegine A3, Indole-3-acetamide, Ethanolamine and Caffeine **Limonoids**: Limonin, Nomilin, Ichangin Obacunone **Coumarin**: Neoacrimarine-I, Citrusarin A **Amino acid**: tyrosine, proline, *N*-Methyl-proline Sugars: Erythrose, Xylose **Organic acids**: citric acid, malic acid, quinic acid, aconitic acid, glyceric acid, furoic acid, maleic acid, myristic acid, stearic acid, Succinic acid, citramalic acid, myristic acid, linoleic acid, glucosyringic acid, fusidic acid, nomilinic acid	[262]
Peels Wild orange (*C. sinensis* [L.] Osbeck) ‘Hong Anliu’; Colombia	Drying, ultrasonication/Water	Hypersil BDS (C_8_) 250 × 4.6 mm, 5 µm HPLC–DAD ESI–MS [M^+^Na]^+^ (m/z 633)	(A) 0.1% HCOOH in H_2_O, (B) CH_3_CN, 75% A and 25% B	**Flavanones**: Hesperidin, Neohesperidin **Flavones**: Diosmin, Tangeretin, Hesperetin	[183]
Peels (Navel sweet orange (*Citrus sinensis;* Greece)	Soxhlet extraction/MeOH-EtOAc	2DTLC; TLC-UV Folin Ciocalteu test	80:20:40, EtOAc:CH_3_COOH: H_2_O and 15% CH_3_COOH	Total phenolic content	[263]
Peels (Orange peels (*C. sinensis*; France)	Instant controlled pressure drop DIC technology and Hydro-distillation; UAE and Solvent extraction	GC–MS: Nonpolar column was HP5MS™ (30 m × 0.25 mm × 0.25 µm film thickness Polar column was a Stabilwax consisting of Carbowax-PEG (60 m × 0.25 mm × 0.25 lm film thickness HPLC: Purospher Star RP C_18_ column (250 × 4mm, 5 µm) with a RP18 guard column (4 × 4 mm I.D.; 5 µm GC–MS and GC–FID (Essential oils) HPLC (Flavonoids)	GC–MS Career gas: Helium 0.5% CH_3_COOH and 100% CH_3_CN	Essential oils fractions: **Monoterpenes**: *α*-Pinene, Camphene, *β*-Myrcene, Limonene, *δ*-3-Carene, Terpinolene **Oxygenated monoterpenes**: Cis-Sabinene hydrate, Linalool, Camphor, Terpin-4-ol, *α*-Terpineol **Sesquiterpenes**: *ε*-Caryophyllene, *α*-Humulene, *α*-Muurolene, *δ*-Cardinene **Oxygenated Sesquiterpenes**: *δ*-Germacrene **Flavonoids**: Hd, Ni	[190]
Whole fruit (Sweet orange; *C. sinensis*; China)	Drying, UB30, centrifugation/MDS	Diamonsil C_18_ 250 × 4.6 mm FGs, PMFs and phenolic acids HPLC–PDA 210 to 400 nm Limonoids UV—210 nm Synaphrine UV-vis—225 nm	FGs, PMFs and phenolic acids (A) MeOH, (B) 4% AcOH (*v/v*) Limonoids MeOH/Acetonitrile/PBS (containing 0.03 M potassium dihydrogen phosphate, pH 3.5) 10:40:39 by Volume Synaphrine MeOH/H_2_O/SDS (70:30:0.1) by volume	**FGs and PMFs**: Narirutin, Hesperidin, Nobiletin, Tangeretin **Phenolic acids**: Caffeic, *p*-Coumaric, Ferulic, Sinapic, Protocatechuic, *p*-Hydroxybenzoic, Vanillic **Limonoids**: Limonin, Nomilin **Alkaloid**: Synaphrine	[253]
Peels (Satsuma mandarin (*C. unshiu* Marc.; China)	Ultrasound assisted extraction (UAE)/80% MeOH	RP C_18_ column(250 × 4.6 mm, 5 µm) HPLC) coupled with a photodiode array (PDA) detector	4% (*v/v*) acetic acid in water/100% methanol (80:20, *v/v*)	**FGs and PMFs**: Nr, Hd **Phenolic acids**: Caffeic, *p*-Coumaric, Ferulic, Sinapic, Protocatechuic, *p*-Hydroxybenzoic, Vanillic	[264]
Peels (*C. unshiu*; S. Korea)	Solvent extraction/EtOH	SunFire C_18_ column 250 × 4.6 mm, 5 µm HPLC UV-DAD, 280 nm	(A) MeOH, (B) 0.5% Acetic acid in H_2_O	Hesperidin, Narirutin, Naringin	[265]
Peels (*C. unshiu*; S. Korea)	Subcritical water extraction	HPLC (Zorbax Eclipse Plus C_18_ column (4.6 × 100 mm, 3.5-μm) LC–MS/MS (Zorbax Eclipse XDB C_18_ column (4.6 × 50 mm, 1.8 μm) HPLC, LC–MS/MS (ESI+ capillary (350 °C, 5.5 kV, N_2_)	HPLC (A) Aq. CH_3_COOH (0.6%); (B) MeOHLC MS/MS (distilled water containing 5 mM HCOONH_4_–MeOH–CH_3_COOH (29.4:70:0.6, *v/v/v*)	Hesperidin, Naringin	[266]
Whole fruit (Weizhang Satsuma or Owari Satsuma Mandarin; *C. unshiu* Marc. cv Owari; China)	UAE/80% MeOH: dimethylsulphoxide:water (4:5:1, *v*/*v*/*v*)	Diamonsil C_18_, 250 mm × 4.6 mm HPLC-UV–DAD, 200–400 nm Limonoids: 210 nm Synaphrine- 225 nm	Flavonoids (A) methanol and (B) 4% acetic acid, *v/v* Limonoids: MeOH: Acetonitrile: PBS (10:40:39. *v/v/v*) Synaphrine: MeOH:Water:SDS (70:30:0.1)	**Flavonoids**: Narirutin, Hesperidin, Nobiletin, Tangeretin **Phenolic acids**: Caffeic, *p*-Coumaric, Ferulic, Sinapic, Protocatechuic, *p*-Hydroxybenzoic, Vanillic **Limonoids**: Nomilin, Limonin **Alkaloid**: Synephrine	[267]
Whole fruit (Weizhang Satsuma or Owari Satsuma Mandarin; *C. unshiu* Marc. cv Owari; China)	Drying, UB30, centrifugation/MDS	Diamonsil C_18_ 250 × 4.6 mm FGs, PMFs and phenolic acids HPLC–PDA 210 to 400 nm Limonoids UV-210 nm Synaphrine UV-vis—225 nm	FGs, PMFs and phenolic acids (A) MeOH,(B) 4% AcOH (*v*/*v*) Limonoids MeOH/Acetonitrile/PBS (containing 0.03 M potassium dihydrogen phosphate, pH 3.5) 10:40:39 by Volume Synaphrine MeOH/H_2_O/SDS (70:30:0.1) by volume	**FGs and PMFs**: Narirutin, Hesperidin, Nobiletin, Tangeretin **Phenolic acids**: Caffeic, *p*-Coumaric, Ferulic, Sinapic, Protocatechuic, *p*-Hydroxybenzoic, Vanillic **Limonoids**: Limonin, Nomilin **Alkaloid**: Synaphrine	[253]
Peel and Pulp Jaffas weetie (Oroblanco, pummelo-grapefruit hybrid (*Citrus grandis* ×*paradisi*) and white grapefruit; Israel	Lyophilization, Solvent extraction (MeOH); Alkali/Acid hydrolysis	Spherisorb 5 ODS column (250 × 4.6 mm)	(A) 5 mM citric acid + 5 mM sodium dihydrogen Orthophosphate + 0.3 mM caprylic acid (adjusted to pH 2.0 by phosphoric acid) and (B) 80% (*v*/*v*) methanol	**Phenolic acids**: Gallic acid, Protocatechuic acid, *p*-hydroxybenzoic acid, Vanillic acid, *p*-coumaric acid, Ferulic acid, Sinapic acid	[49]

Solvents: MWC: Methanol, water, and HCl; MWH: Methanol, water, and *n*-hexane; MOH: Methanol; MDS: Methanol and dimethylsulfoxide; WEA: Water, ethanol, and acetone; EAA: Ethanol and ammonium acetate; AcOH: Acetic acid. Detection: HPLCMS: High-performance liquid chromatography coupled with ESI–MS/MS; UV-Vis: Ultraviolet and visible detector; UV-DAD: Ultraviolet diode array detector; UPLC–PDA: Ultra-performance liquid chromatography with photodiode array detector. UFLC–DAD–Q-TOF-MS/MS: Ultra-fast liquid chromatography coupled with diode-array detection and quadrupole/time-of-flight mass spectrometry; RRLC–ESI–QTOF/MS: Rapid-resolution liquid chromatography coupled to electrospray ionization quadrupole time-of-flight mass spectrometry; Q-TOF, Quadrupole time of-flight; ESI: Electrospray ionization; MS: Mass spectrometry; RRLC: Rapid-resolution liquid chromatography; DART: Direct analysis in real time; PMF: Polymethoxylated flavones.

**Table 15 foods-08-00523-t015:** Identification of the chemical constituents in citrus phytochemicals by UV-visible and mass spectroscopy [34,254,257,268,269,270,271,272].

Analyte Identification	Formula	[M + H]^+^	λ_max_ (nm)
Flavone-*C*-glycoside			
Vicenin2	C_27_H_30_O_15_	595.2	270
Apigenin-8-*C*-glucoside	C_21_H_20_O_10_	433.1	2870, 334
Flavone-*O*-glycoside			
Luteolin-7-*O*-rutinoside	C_27_H_30_O_15_	595.2	255, 267
Rhiofolin	C_27_H_30_O_14_	579.2	330
Flavone aglycones			
Apigenin	C_15_H_10_O_5_	269.0	209, 270, 324
Luteolin	C_15_H_10_O_6_	285.0	276, 326
Flavanone-*O*-glycoside			
Eriocitrin	C_27_H_32_O_15_	597.2	283
Narirutin	C_27_H_32_O_14_	581.2	283
Naringin	C_27_H_32_O_14_	581.2	283, 330
Melitidin	C_33_H_40_O_18_	725.2	280
Hesperidin	C_28_H_34_O_15_	609.0	285, 330
Flavanone aglycones			
Naringenin	C_15_H_12_O_5_	273.1	283
Hesperetin	C_16_H_14_O_6_	303.1	280,
Eriodictyol	C_15_H_12_O_6_	287.0	287, 324
Polymethoxylated Flavones			
Tangeretin	C_20_H_20_O_7_	373.1	335, 415
Nobiletin	C_21_H_22_O_8_	403.1	331
Sinensetin	C_20_H_20_O_7_	373.1	328
Flavonols			
Rutin	C_27_H_30_O_16_	609.0/611.0	254, 354
Quercetin	C_15_H_10_O_7_	301.0	254, 354
Phenolic acids			
*p*-hydroxybenzoic acid	C_7_H_6_O_3_	137.0	285
Protocatechuic acid	C_7_H_6_O_4_	153.0	254, 286
Ferulic acid	C_10_H_10_O_4_	193.0	283, 306
*p*-coumaric acid	C_9_H_8_O_3_	163.0	269, 308, 298
*o*-coumaric acid	C_9_H_8_O_3_	163.0	277
Caffeic acid	C_9_H_8_O_4_	181.0	295, 323
Alkaloids			
Synaphrine	C_9_H_13_NO_2_	168.0	225
Coumarins			
Meranzin hydrate	C_15_H_18_O_5_	279.1	330
Meranzin	C_15_H_16_O_4_	261.1	330
Marmin	C_19_H_24_O_5_	333.2	320
Bergaptan	C_12_H_8_O_4_	217.0	320
Isomaranzin	C_15_H_16_O_4_	261.1	320
Limonoids			
Limonin	C_26_H_30_O_8_	471.2	210, 277
Isoobacunoic acid	C_26_H_32_O_8_	473.2	210
Nomilin	C_28_H_34_O_9_	515.2	210
Obacunone	C_26_H_30_O_7_	455.2	210
Carotenoids			
Phytoene	C_40_H_64_	545.8	274, 286, 300
Phytofluene	C_40_H_62_	543.4	332, 348, 368
*α*-carotene	C_40_H_56_	537.4	418, 446, 472
*β*-carotene	C_40_H_56_	537.4	422, 450, 478
*β*-cryptoxanthin	C_40_H_56_O	553.3	422, 450, 470
Zeaxanthin	C_40_H_56_O_2_	569.4	423, 450, 474
Lutein	C_40_H_56_O_2_	551.7	420, 442, 472
Organic acids			
Malic acid	C_4_H_6_O_5_	133.0	213
Citric acid	C_6_H_8_O_7_	191.0	209, 295
Ascorbic acid	C_6_H_8_O_6_	175.0	289
Fatty acids			
Palmitic acid	C_16_H_32_O_2_	257.2	-
Stearic acid	C_18_H_36_O_2_	283.3	-
Oleic acid	C_18_H_34_O_2_	281.2	
Linoleic acid	C_18_H_32_O_2_	279.2	-
Linolenic acid	C_18_H_30_O_2_	277.2	-

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
