# Peer review of "Modern Extraction and Purification Techniques for Obtaining High Purity Food-Grade Bioactive Compounds and Value-Added Co-Products from Citrus Wastes"

_foods, 2019, doi:10.3390/foods8110523_

Round 1

Reviewer 1 Report

The essentially is written very well, I have only minor editorial comments.

In the introduction, the font size is different in lines 34-44.

In lines 86 are too many spaces for  Figure 2.

In lines 291 reference number is missing

In lines 332 there is an error in the author's name - should be Sanfélix-Gimeno,

In lines 422 no needed space after Figure 12

In Table 8 value at Zeaxanthin is shifted

In lines 544 the% sign is at the beginning of a line with no value, the same in lines 564

In lines 645 not a good description - “in the Ref.” the same in lines 944

In lines 856-858 different font color

In lines 856 Malgorzata and Marek these are first names and not names, their names are Dajs and Henczak

In lines 1007 no capital letter is needed

In Table 14 sometimes are different font color

In lines 1362 are too many spaces after [206,303].

In lines 1985 the dot is missing and there are too many spaces

Author Response

Authors’ response to Reviewers' comments #1

In the introduction, the font size is different in lines 34-44.

Authors’ response: The required correction has been incorporated and highlighted in the revised manuscript.

In lines 86 are too many spaces for Figure 2.

Authors’ response: The unnecessary spaces have been deleted and the sentence has been correctly arranged and highlighted in the revised manuscript.

In lines 291 reference number is missing.

Authors’ response: The reference hasbeen added and the portion is highlighted in the revised manuscript.

In lines 332 there is an error in the author's name - should be Sanfélix-Gimeno,

Authors’ response: The error has been corrected and the portion is highlighted in the revised manuscript.

In lines 422 no needed space after Figure 12

Authors’ response: Figure 12 has been transferred to the Supplementary Information. The required correction has been incorporated in the revised manuscript and highlighted.

In Table 8 value at Zeaxanthin is shifted.

Authors’ response: The required modification has been incorporated in the revised manuscript and the portion is highlighted.

In lines 544 the% sign is at the beginning of a line with no value, the same in lines 564.

Authors’ response: The required correction has been incorporated in the manuscript.

In lines 645 not a good description - “in the Ref.” the same in lines 944.

Authors’ response: The sentences have been modified in the revised manuscript and the portions have been highlighted for reviewer’s perusal.

In lines 856-858 different font color.

Authors’ response: The abnormal portion has been corrected and does not appear in the revised manuscript.

In lines 856 Malgorzata and Marek these are first names and not names, their names are Dajs and Henczak

Authors’ response: The required corrections in the names have been incorporated in the revised manuscript and the portion has been highlighted.

In lines 1007 no capital letter is needed

Authors’ response: The required modification has been incorporated in the manuscript.

In Table 14 sometimes are different font color

Authors’ response: The errors have been removed in the revised manuscript.

In lines 1362 are too many spaces after [206,303].

Authors’ response: The required modification has been incorporated in the revised manuscript and the portion has been highlighted.

In lines 1985 the dot is missing and there are too many spaces

Authors’ response: All the references have been thoroughly revised and mistakes have been corrected at required places.

We sincerely thank to the reviewers for their valuable suggestions to help us bring this manuscript to its best possible form. To the best of our efforts, we have revised the manuscript to its best possible format.

Reviewer 2 Report

The manuscript submitted by Mahato et al. addresses “Modern extraction and purification techniques for obtaining high purity food grade bioactive compounds and value-added co-products from citrus wastes”. In a first analysis, the subject of the review is of great interest, tend in account that one of the hot topics is the valorization of waste food. Although the relevance of the addressed subject, the manuscript requires to be revised, detailed comments and suggestions are shown below. It is recommended not to accept it until major revisions will be considered.

The authors should take care with the formation of the document, for example in the lines 35-43 and line 541 the letters are bigger than the usual. Table 10, are also not well formatted, check please.

The quality of the figures must be improved, please.

The Figures 6, 7, 10, 11, 12, 13, 14, 16, 20, 21, 22, 23, 24 and 25 should move to supporting information. In the article put figures simpler, easy to read, i.e., figures with essential information only.

The figures in supporting information should have a different numeration of the article. For example Figure S1.

The Figure in the supporting information number 3 do not have subtitle.

Please, standardize the number of decimal places, for example check table 1.

Line 216/217 (“The method showed better performance in terms of higher yields than conventional techniques and confirmed by different antioxidant assay systems.”), please add the values obtained for both techniques  for an easier interpretation, please.

Line 275/276 (“The solvent ratios can be adjusted according to the sample type, peel waste or pulp waste, depending upon the water content in the sample”), comment the best ratios for which type of sample, please.

Line 292-294 (“In this method, the orange peel waste is first pre-treated with calcium  hydroxide, hydrochloric acid, etc. followed by extraction of the compounds by solvent extraction  method.”), why did Curto et al. pre-treatment? The authors can explain, please.

Please, write um paragraph about yours perspectives/opinion in the future for this area.

Author Response

Authors’ response to Reviewers' comments #2

The authors should take care with the formation of the document, for example in the lines 35-43 and line 541 the letters are bigger than the usual. Table 10, are also not well formatted, check please.

Authors’ Response: The required corrections have been incorporated in the revised manuscript and the portions have been highlighted.

The quality of the figures must be improved, please.

Authors’ Response: All the figures in the manuscript have been improved up to the resolution of 600 dpi and 1000 pixels width/height, as per Journal’s requirement.

The Figures 6, 7, 10, 11, 12, 13, 14, 16, 20, 21, 22, 23, 24 and 25 should move to supporting information. In the article put figures simpler, easy to read, i.e., figures with essential information only.

 Authors’ response: The Figures 6, 7, 10, 11, 12, 13, 14, 16, 20, 21, 22, 23, 24 and 25 have been transferred to the Supplementary Information File. The main manuscript now contains fewer number of figures (12 Figures), simpler than previous file, and to the best of our knowledge, retains essential information only.

The figures in supporting information should have a different numeration of the article. For example, Figure S1.

Authors’ response: The suggestion has been incorporated in the revised manuscript and the portions are highlighted. The figures in Supplementary information are now designated as Figure S-1, S-2, and so on. The table numbers have also been modified as Table-ST-1, ST-2, and so on.

The Figure in the supporting information number 3 do not have subtitle.

Authors’ response: The Figure S-3 has been provided with a subtitle in the revised supporting Information file.

Please, standardize the number of decimal places, for example check table 1.

Authors’ response: The required corrections have been incorporated in the revised manuscript. All the tables have been thoroughly revised in the manuscript.

Line 216/217 (“The method showed better performance in terms of higher yields than conventional techniques and confirmed by different antioxidant assay systems.”), please add the values obtained for both techniques for an easier interpretation, please.

Authors’ response: The sophisticated techniques have an edge over conventional techniques of extraction of biomolecules from citrus wastes in many respects. The tables in the manuscript have been meticulously constructed to interpret this concept. The comparison of quantitative values reported for the amounts of different bioactive compounds obtained by different conventional and sophisticated techniques have been elaborately recorded in the Table 12.

Line 275/276 (“The solvent ratios can be adjusted according to the sample type, peel waste or pulp waste, depending upon the water content in the sample”), comment the best ratios for which type of sample, please.

Authors’ response: The solvent ratios employed for the extraction of biomolecules from dried and wet samples are different. Table 3 shows different solvent types and percentage by volume used for extracting total phenolic content from the different citrus species. According to the different research reports published so far, it is inferred that no fixed ratio can be termed as ‘best ratio’ which can furnish best yields in every possible sample structure or method of extraction process and the best ratio has to be optimized by hit and trial method during every individual case. For example, with a solvent 80 % aqueous ethanol used for the extraction of total phenolic content from orange peel and pulp by solvent extraction method at 70ºC for 3h yield 123 and 179 mg GAE/100 g of total phenolic content from pulp and peel, respectively from orange waste, whereas, the same combination yield 98 and 61 mg GAE/100 g of TPC from lemon peels and pulp, respectively, and 105 and 170 mg GAE/100 g of TPC from mandarin peel and pulp, respectively. Similar observation is found with other solvent composition and extraction conditions. For obtaining different bioactive molecules, different solvents are utilized. For best result, a gradient ratio with respect to the polarity of the solvent in a step wise methodology is suggested for systematic research. The concept has been elaborated in terms of extraction, isolation, purification and determination of bioactive compounds has been meticulously summarized in Tables 12-14.

Line 292-294 (“In this method, the orange peel waste is first pre-treated with calcium  hydroxide, hydrochloric acid, etc. followed by extraction of the compounds by solvent extraction  method.”), why did Curto et al. pre-treatment? The authors can explain, please.

Authors’ response: Alkaline treatment or pre-treatment with calcium hydroxide induces the insolubilization of pectin present in the complex mixture of citrus waste under extraction process. The insoluble pectin is now easy to filter out leaving behind flavonoids in the solution. The pectin otherwise remains as a hydrosoluble entity in the solution and hampers the process of crystallization of flavonoids and separation of the same. Furthermore, liming facilitates isomerization of the flavonoids and solubilizes the derived chalcones. The filtered liquid is then acidified with hydrochloric acid to facilitate the inverse reaction and separation of soluble flavonoids.

Please, write um paragraph about yours perspectives/opinion in the future for this area.

Authors’ response: The following paragraph has been added to the revised manuscript:

The ‘Citrus’ itself is a complete profitable industry. With continuously increasing production and utilization of citrus fruits and various citrus products across the globe and anticipation of huge progress in the future years, it has now become crucial to address on this topic. With integrated research and active collaboration between different scientific and engineering fields, it is possible to achieve maximum utilization of citrus waste and explore large scale applications. However, there remain several lacunae in the following areas. (a) elaborated profiling of chemical constituents of citrus waste, with an aim of detecting as well as extracting higher amounts of bioactive compounds and value added products, (b) design of process engineering and chemical plants that can adapt to a variety of sources and facilitate production, (c) development of efficient microbial strains for converting effectively the citrus wastes into high value products using either classic mutagenesis or metabolic engineering (d) information sharing between breeders/ cultivators and citrus processing units/industries, waste management plants, and collaboration of industrial/academic researchers (e) policy on systematic management of citrus waste integrated with modern scientific techniques, (g) design of academic curricula/programs in educational centers and universities and construct workshops to increase awareness, work skills and efficiency, (h) efficient transport facilities. The exploitation of citrus bioactive molecules in food and neutraceutics, dietary supplements and by-products in food and beverages and pharmaceutics indicates promising field of future research. Some phytochemicals, such as d-limonene, nomilin and nitrates, which have adverse effect on the processing methods, must be excluded by efficient quality control systems in food processing units. Detailed investigations on their bioactivity, toxicology, stability, and interactions with other food ingredients or packaging materials should be carried out with carefully assessment in vitro and in vivo. This requires active collaboration and interdisciplinary research of food technologists, food chemists, nutritionists and toxicologists to as to develop efficient methods of processing of citrus fruits as well as their waste, innovating user friendly apparatus and equipment to yield products in small as well as large scales and standardize information and database on this area of research to inculcate interests among young generation, or in other words, efforts to provide education to shape the future.

We sincerely thank to the reviewers for their valuable suggestions to help us bring this manuscript to its best possible form. To the best of our efforts, we have revised the manuscript to its best possible format.

Round 2

Reviewer 2 Report

The authors made the requested modifications.